# Van der Waals polarity-engineered 3D integration of 2D complementary logic

Yimeng Guo[1,2,19], Jiangxu Li[1,19], Xuepeng Zhan[3,19], Chunwen Wang[4,19], Min Li[5,6,19], Biao Zhang[7,8], Zirui Wang[9], Yueyang Liu[10], Kaining Yang[11,12], Hai Wang[3], Wanying Li[1], Pingfan Gu[13,14], Zhaoping Luo[1], Yingjia Liu[1,2], Peitao Liu[1], Bo Chen[3], Kenji Watanabe[15], Takashi Taniguchi[16], Xing-Qiu Chen[1], Chengbing Qin[12,17], Jiezhi Chen[3], Dongming Sun[1], Jing Zhang[11,12], Runsheng Wang[9], Jianpeng Liu[5,6,18], Yu Ye[13,14,18], Xiuyan Li[1,18✉], Yanglong Hou[7,8✉], Wu Zhou[4✉], Hanwen Wang[18✉] & Zheng Han[11,12,18✉]

Vertical three-dimensional integration of two-dimensional (2D) semiconductors holds great promise, as it offers the possibility to scale up logic layers in the $z$ axis[1–3]. Indeed, vertical complementary field-effect transistors (CFETs) built with such mixed-dimensional heterostructures[4,5], as well as hetero-2D layers with different carrier types[6–8], have been demonstrated recently. However, so far, the lack of a controllable doping scheme (especially p-doped $WSe_2$ (refs. 9–17) and $MoS_2$ (refs. 11,18–28)) in 2D semiconductors, preferably in a stable and non-destructive manner, has greatly impeded the bottom-up scaling of complementary logic circuitries. Here we show that, by bringing transition metal dichalcogenides, such as $MoS_2$, atop a van der Waals (vdW) antiferromagnetic insulator chromium oxychloride (CrOCl), the carrier polarity in $MoS_2$ can be readily reconfigured from n- to p-type via strong vdW interfacial coupling. The consequential band alignment yields transistors with room-temperature hole mobilities up to approximately 425 $cm^2 V^{-1} s^{-1}$, on/off ratios reaching $10^6$ and air-stable performance for over one year. Based on this approach, vertically constructed complementary logic, including inverters with 6 vdW layers, NANDs with 14 vdW layers and SRAMs with 14 vdW layers, are further demonstrated. Our findings of polarity-engineered p- and n-type 2D semiconductor channels with and without vdW intercalation are robust and universal to various materials and thus may throw light on future three-dimensional vertically integrated circuits based on 2D logic gates.

Among the plenitude of advantageous properties, the capability of $z$-dimensional stacking—in principle of an unlimited number of layers—is believed to be one of the most fascinating perspectives of semiconducting van der Waals (vdW) nanoelectronics. This method of bottom-up three-dimensional (3D) vdW integrability may provide an alternative approach to continue the scaling of transistors in the so-called post-Moore's-law age, as the silicon technology is approaching its physical limit for further shrinking of the lateral size of transistors[29,30]. Indeed, for decades, from the very first planar field-effect transistor (FET), to FinFET and to the most advanced gate-all-around FET, the scaling of Si semiconductors has followed an in-plane strategy,

as illustrated in Fig. 1a, while achieving 3D integrability remained extremely challenging. Although 3D interconnection of electrodes has been widely implemented in modern silicon integrated circuits, the essential logic gates are yet confined to only the surface of the silicon substrate, which cannot be arranged into multi-layers. Other attempts of face-to-face bonding of two chips require alignments with ultra-high precision, and the gain of room in the $z$ dimension is not that satisfactory[31,32]. Meanwhile, multilayered 3D flash memory (3D NAND) consists of orthogonally crossed junctions (where floating gate memories are formed) between horizontal and vertical bit and word lines, which, however, do not meet the need for free-design of circuitry[33].

[1]Shenyang National Laboratory for Materials Science, Institute of Metal Research, Chinese Academy of Sciences, Shenyang, China. [2]School of Materials Science and Engineering, University of Science and Technology of China, Anhui, China. [3]School of Information Science and Engineering (ISE), Shandong University, Qingdao, People's Republic of China. [4]School of Physical Sciences and CAS Key Laboratory of Vacuum Physics, University of Chinese Academy of Sciences, Beijing, People's Republic of China. [5]School of Physical Science and Technology, ShanghaiTech University, Shanghai, China. [6]ShanghaiTech Laboratory for Topological Physics, ShanghaiTech University, Shanghai, China. [7]School of Materials, Shenzhen Campus of Sun Yat-Sen University, Shenzhen, China. [8]School of Materials Science and Engineering, Beijing Key Laboratory for Magnetoelectric Materials and Devices, Peking University, Beijing, China. [9]School of Integrated Circuits, Peking University, Beijing, China. [10]State Key Laboratory of Superlattices and Microstructures, Institute of Semiconductors, Chinese Academy of Sciences Beijing, Beijing, China. [11]State Key Laboratory of Quantum Optics and Quantum Optics Devices, Institute of Optoelectronics, Shanxi University, Taiyuan, China. [12]Collaborative Innovation Center of Extreme Optics, Shanxi University, Taiyuan, China. [13]Collaborative Innovation Center of Quantum Matter, Beijing, China. [14]State Key Lab for Mesoscopic Physics and Frontiers Science Center for Nano-Optoelectronics, School of Physics, Peking University, Beijing, China. [15]Research Center for Functional Materials, National Institute for Materials Science, Tsukuba, Japan. [16]International Center for Materials Nanoarchitectonics, National Institute for Materials Science, Tsukuba, Japan. [17]State Key Laboratory of Quantum Optics and Quantum Optics Devices, Institute of Laser Spectroscopy, Shanxi University, Taiyuan, China. [18]Liaoning Academy of Materials, Shenyang, China. [19]These authors contributed equally: Yimeng Guo, Jiangxu Li, Xuepeng Zhan, Chunwen Wang, Min Li. ✉e-mail: xyli@imr.ac.cn; hou@pku.edu.cn; wuzhou@ucas.ac.cn; hwwang@lam.ln.cn; vitto.han@gmail.com

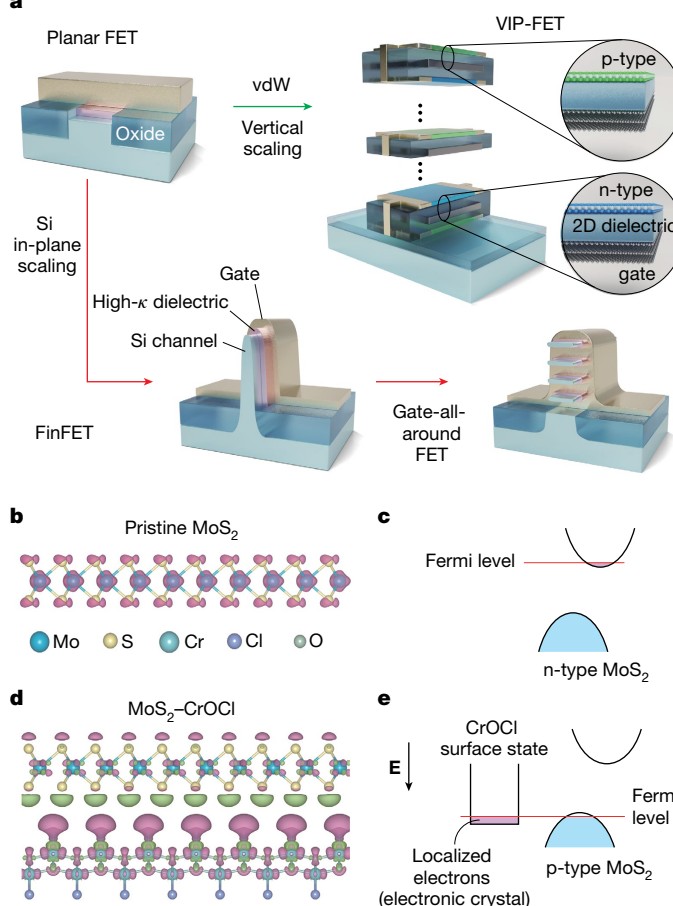

**Fig. 1 | Vertical scaling versus in-plane scaling of semiconducting circuits. a**, Schematic illustration of the routes of scaling in Si and vdW technologies. **b**, Molecular structure of a pristine $MoS_2$ layer. The difference in charge density distributions of slight electron doping with respect to a neutral layer for an isosurface of 10 µe $Bohr^{-3}$ above the Fermi level is superimposed on its molecular model. **c**, Schematic diagram of the n-type $MoS_2$ in the conventional pristine state, with the Fermi level (red solid line) positioned in the vicinity of the conduction band minimum (CBM) in $MoS_2$. **d**, Same plot of differential charge density distributions as **b**, but in a $MoS_2$–CrOCl heterostructure. In **b** and **d**, the charge carrier types of electrons and holes are marked in pink and green, respectively. The atom-symbols used in **b** and **d** are illustrated in the bottom part of **b**. Clear n-type behaviour on electron doping can be seen in **b**, while in the $MoS_2$–CrOCl heterostructure case in **d**, most of the doped electrons are transferred to the CrOCl side. **e**, Schematic band alignment diagram of the $MoS_2$–CrOCl heterostructure under finite vertical negative electric field (corresponding to the negative bottom gate voltage in our experimental configurations), indicating the realization of a p-type semiconducting $MoS_2$ with the Fermi level (red solid line) positioned in the vicinity of the valence band maximum (VBM) in $MoS_2$.

Recently, stacking up vdW semiconductors into 3D vertical circuitry has been a grail pursued with continuous efforts[34–36]. The sizable band-gap and dangling-bond-free surfaces, together with high carrier mobilities and excellent electrostatic control at the ultimate scale (less than 1 nm), make two-dimensional (2D) semiconductors ideal candidates for vertical 3D integration[1–3,37]. The prediction is that advanced monolithic 3D integrated circuits constructed with speedy layer-to-layer signal transmission and efficient heat dissipation will provide much higher integration density[38]. However, application-wise, 3D integrated circuits of 2D semiconductors have largely been restricted due to the difficulty in obtaining controllable doping of n- and p-type polarities, which is fundamental for complementary logic[34]. To date, a limited number of examples have been realized in vertical complementary field-effect

transistors (CFETs) constructed using 2D semiconductors, such as mixed-dimensional heterostructures[4,5] and hetero-2D layers with different carrier polarities[6–8], among which a maximum of two vertical layers of complementary logic has been demonstrated. Indeed, while n-type 2D semiconductors are advancing rapidly in terms of electrical performance[39,40], only a handful of p-doping strategies are known for 2D semiconductors such as $WSe_2$ (refs. 9–17) and $MoS_2$ (refs. 11,18–28), using methods including chemical dopants, contact engineering, or oxide coating. Notice that these doping methods may suffer from inhomogeneities or degradation of carrier mobility, and few of them are physically capable of enabling multilayered vertical assemblies of 3D complementary logic.

In this work, we devise a simple and non-destructive doping method to reconfigure, in a controllable manner, the carrier polarity of 2D semiconductors through vdW interfacial coupling. We found that, unlike the usually manifested n-type nature, few-layered transition metal dichalcogenides (TMDs) (including $MoS_2$, $WSe_2$ and $MoSe_2$) interfaced with few-layered CrOCl systematically turn into p-type and display excellent air stability. Density functional theory (DFT) calculations suggest that this interfacial-coupling-induced polarity inversion is a result of charge transfer from TMDs to CrOCl, followed by subtle $e$–$e$ interactions in the surface state of CrOCl, which should be a universal effect at the interface between TMDs and layered insulators with high work function and large-enough effective mass in its surface band. Taking $MoS_2$ as an example, thanks to the atomically clean interface, the $MoS_2$–CrOCl hybrid exhibits a maximum room-temperature hole mobility reaching approximately 425 $cm^2$ $V^{-1}$ $s^{-1}$, with on/off ratio exceeding $10^6$. Further, we construct n- and p-doped logic units by selectively stacking modules of vdW gate, dielectric and semiconducting layers, with and without interfacial coupling layers, defined as vertical inversely polarizable field-effect transistors (VIP-FETs). Our doping strategy can therefore be employed to fabricate self-complemented logic devices vertically, throwing light on the vertical scaling route (Fig. 1a) towards advanced 3D integration of semiconducting circuits.

## Modelling of vdW polarity-engineered $MoS_2$

To enable the vertical 3D integration of 2D semiconductors, controlling the n- and p-type polarities is crucial. Therefore, it is essential to identify an effective approach to achieve p-doped 2D semiconductors without compromising the carrier mobility. Recent research shows that CrOCl is one of the candidates for engineering the interfacial coupling in 2D electronic gas systems such as graphene, which gives rise to exotic quantum ground states[41,42]. The interfacial coupling between TMDs and CrOCl, however, remains unexplored so far. We first consider theoretically a model system of CrOCl coupled to TMD, using $MoS_2$ as an example. By calculating the charge density difference between the slightly electron-doped and neutral $MoS_2$ layer, as shown in Fig. 1b, the pristine state of $MoS_2$ (and most of the TMDs) exhibits n-type behaviour, with the Fermi level close to the conduction band minimum (CBM), illustrated in Fig. 1c. To determine their work functions, we performed DFT calculations of 10-layer $MoS_2$ and 5-layer CrOCl, and the CBM of $MoS_2$ is estimated to be about 0.465 eV above the CBM of CrOCl in the absence of vertical electric fields. Therefore, if $MoS_2$ is doped by additional electrons, we would expect these extra charges to transfer from the CBM of $MoS_2$ to CrOCl. This is more explicitly elucidated by calculating the charge density difference between the electron-doped case and the charge-neutral case at the $MoS_2$–CrOCl interface, as shown in Fig. 1d. Clearly, the doped electron carriers are concentrated at the CrOCl side at the interface, leaving some holes on the $MoS_2$ side.

To further elucidate the experimental observations, we carried out DFT calculations of a three-$MoS_2$-layer + three-CrOCl-layer heterostructure with a supercell of about 200 atoms (more details can be found in Methods). Note that, as experimentally CrOCl still acts as a

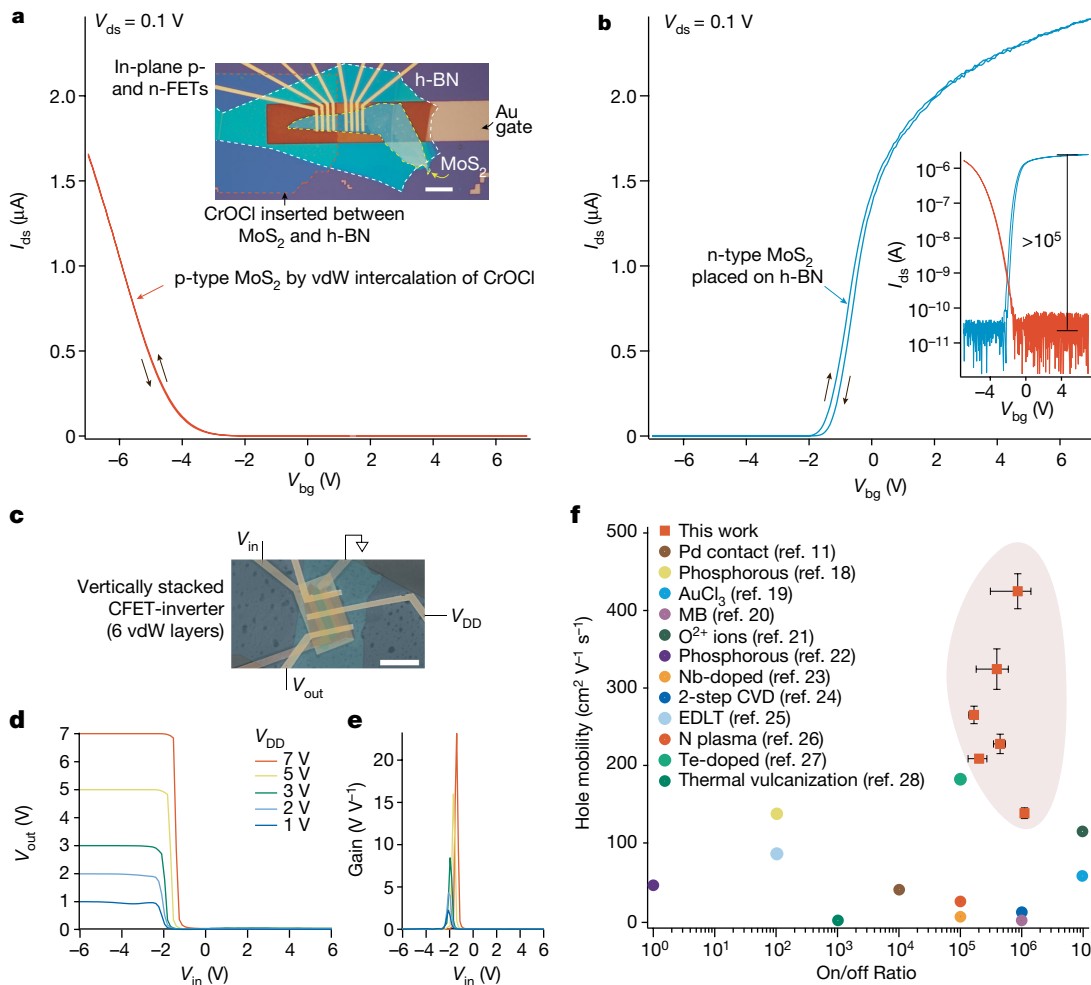

**Fig. 2 | Electrical performance of MoS₂–CrOCl complementary FETs.**
**a,b**, Source–drain current $I_{ds}$ as a function of back gate voltage $V_{bg}$, measured for a typical MoS₂–CrOCl FET (**a**) (orange line) and a conventional pristine MoS₂ FET placed on h-BN (**b**) (blue line). Trace and retrace (as indicated by the solid black arrows) are recorded in **a** and **b**, exhibiting negligible gate hysteresis. Measurements in **a** and **b** are carried at $V_{ds}$ = 0.1 V and at room temperature. The inset in **a** shows an optical micrograph of MoS₂ FETs made from the same MoS₂ flake, but with different polarities when placed on CrOCl or h-BN. Each constituent layer is highlighted by coloured dashed lines. The inset in **b** shows the same data as in **a** and **b**, but plotted on a log scale. **c**, False-coloured SEM image of a vertically stacked MoS₂ complementary logic inverter. **d**, Output voltage $V_{out}$ as a function of input voltage $V_{in}$ at various supply voltages $V_{DD}$. **e**, The gain for each of the curves in **d**. **f**, Performance of state-of-the-art p-type MoS₂ FETs in the parameter space of on/off ratio and hole mobility at room temperature. Data points of this work (solid red squares) are discussed in more detail in Supplementary Fig. 15. In **f**, the error bar for the on/off ratio is defined as $1/\delta I_{off}$, with $\delta I_{off}$ being the standard deviation of $I_{off}$ in each device, while maximum $I_{on}$ is fixed. The error bar for the hole mobility is defined as the standard deviation of the $\gamma \times dI_{ds}/dV_g$ at the vicinity of the maximum, where $\gamma$ is a coefficient obtained from the sample, written as $\frac{L}{WCV_{ds}}$. $L$, $W$, $C$ and $V_{ds}$ are respectively the length, width, the gate capacitance and the source–drain voltage for the measured device. Scale bars, 10 µm (**a**,**c**).

gate dielectric and hence no free carriers can be found in it, the transfer of electrons (via tunnelling from MoS₂ to the Cr 3$d$ orbitals, as suggested by DFT calculations) into the surface states of CrOCl has to be in a localized manner. Indeed, by considering the band structures of MoS₂, CrOCl and MoS₂–CrOCl heterostructures separately (Supplementary Figs. 1–3 in Supplementary Note 1, and also see Methods), we found that the mechanism here in our system is more than a trivial charge transfer, but rather is further followed by a combination of $e$–$e$ interaction (which drives the charges in the CrOCl surface state into insulator) and self-adjustment of band alignments (Fig. 1e and Supplementary Fig. 4). This is fundamentally different from conventional doping strategies for such TMD semiconductors.

## Characterizations of CrOCl-interfaced MoS₂

We now construct MoS₂–CrOCl vdW heterostructures by stacking few-layer MoS₂ onto CrOCl through a standard dry-transfer method[43]. An optical micrograph of a typical sample is shown in the inset of Fig. 2a,

where the left part is MoS₂–CrOCl FET and the right part is a control sample of conventional MoS₂ FET, with each constituent layer highlighted by red and yellow dashed lines. Here, few-layered hexagonal boron nitride (h-BN), highlighted by a white dashed line in the inset of Fig. 2a, is employed as the gate dielectric for the MoS₂–CrOCl and MoS₂ FETs. Electrodes of Cr (5 nm) and Au (50 nm) are fabricated via standard lithography and thermal evaporation (fabrication details are available in Methods). Indeed, as shown in Fig. 2a, CrOCl-interfaced MoS₂ FET manifests typical p-type semiconducting behaviour (as expected from our simulations) with source–drain current $I_{ds}$ above 1 µA at a source–drain voltage $V_{ds}$ = 0.1 V. The on-state current of p-type MoS₂ FET is comparable to that of its counterpart of an n-type MoS₂ placed on h-BN, with field-effect curve shown in Fig. 2b. The n-type nature of the latter is often attributed to its electron-donating sulfur vacancies and considerable Fermi level pinning effect at the metal-electrodes–MoS₂ interface[44,45]. Additional characterizations of p-type MoS₂–CrOCl FETs can be found in Supplementary Figs. 5 and 6. Output characteristics of a typical MoS₂–CrOCl FET with SiO₂ serving

as gate dielectric are also shown in Supplementary Fig. 7. In general, an on/off ratio exceeding $10^5$ is observed in the MoS$_2$–CrOCl FETs (inset in Fig. 2b), whose transfer curves show negligible hysteresis, indicating the high quality of the MoS$_2$–CrOCl interfaces. It also excludes the trivial scenario of defect-induced effects (control experiment of mild-plasma-treated CrOCl surface can be found in Extended data Fig. 1, where huge hysteresis is seen in the field-effect curves of MoS$_2$ placed on it).

Evidence for such a p-type doping characteristic in CrOCl-interfaced MoS$_2$ FETs is further supported by Kelvin probe force microscopy (KPFM) measurements, where an obvious reduction of surface potential is seen in MoS$_2$ in the CrOCl-interfaced region, as shown in Supplementary Fig. 8. From our KPFM experiment, the work function for CrOCl is much larger than that of MoS$_2$, leading to a significantly downward band alignment. This is consistent with the DFT results described in the previous section. It is noticed that, compared with few-layer MoS$_2$, the frequency of the Raman $A_{1g}$ and $E^1_{2g}$ modes in the MoS$_2$–CrOCl region are blue-shifted (more discussions can be seen in Supplementary Fig. 9), which might be related to the change of Fermi level[46,47], echoing the observed p-type doping effect. Detailed electrical measurements are further conducted on FETs fabricated based on other CrOCl-interfaced TMD 2D semiconductors, including MoSe$_2$ and WSe$_2$ (see Supplementary Figs. 10 and 11). It is seen that, using Cr–Au as electrodes, most of the TMD–CrOCl heterostructures become p-doped, speaking to the universality of our doping strategy. Their electrical performance can be further improved by fine tuning the fabrication procedures, which will be discussed in the next sections.

## MoS$_2$-based vertical CFETs

In the following, we demonstrate the feasibility of MoS$_2$-based self-complemented circuits, where the combination of n- and p-type transistors enables logic function with high noise immunity and low static power consumption. To start with, a conventional planar inverter, namely, a 'NOT' logic gate that outputs a voltage representing the opposite logic level to its input[12,48], is constructed by using laterally adjacent n- and p-type transistors of MoS$_2$, as indicated by the optical image in Supplementary Fig. 12a. Electrical performance of the inverter is shown in Supplementary Fig. 12b, where standard inverter action is observed for switching between logic '1' (close to the supply voltage $V_{DD}$) and logic '0' (close to 0 V).

The key achievement of this study is the obtaining of vertically stackable and polarity-invertible high mobility 2D FETs, which is of significance in increasing the integration density by scaling in the $z$ dimension. For instance, vertically assembled 2D semiconducting inverters that consist of n- and p-type transistors vertically assembled with each other, can improve the integration level by 42–50% compared with those conventional logic circuits with planar architecture using bulk Si semiconductors[4,49]. As illustrated by the drawing in Extended data Fig. 2, using the described controllable p-doping strategy, a 3D-integrated 2D inverter with both n- and p-type transistors vertically stacked using modules of vdW gate, dielectric and MoS$_2$, with and without CrOCl coupling layers, can be realized, which we previously defined as VIP-FETs and is also a typical vertical nanoarchitecture of CFET.

False-colour scanning electronic microscope (SEM) image of such a typical 3D-integrated 2D inverter with 6 vdW layers is shown in Fig. 2c, with its circuit diagram being similar to that depicted for conventional lateral inverter in Extended data Fig. 2d. Here, a gate electrode (vertically shared by the upper p-channel and lower n-channel) serves as the input voltage ($V_{in}$) terminal, and the n-FET is grounded while a supply voltage $V_{DD}$ is applied to the p-FET. The transfer curves, showing output voltage $V_{out}$ versus $V_{in}$, of the vertically stacked 2D CFET inverter with various $V_{DD}$, are plotted in Fig. 2d. Clear signal

inversion is observed: the ON-states in p-FET and n-FET compete with each other, yielding a $V_{out}$ switched from $V_{DD}$ (logic '1') to ground (logic '0') at different ranges of the input voltage $V_{in}$. The sharp transition between the logic states can be characterized by the voltage gain (defined as |d$V_{out}$/d$V_{in}$|), which is a crucial metrics presenting the sensitivity of $V_{out}$ to the change in $V_{in}$. As shown in Fig. 2e, the resulted voltage gain is approximately 23 for $V_{DD} = 7$ V and approximately 2 for $V_{DD} = 1$ V.

We evaluate the noise margins of the MoS$_2$-based vertical CFETs by using the expressions NM$_H$ = $V_{OH}$ − $V_{IH}$ and NM$_L$ = $V_{IL}$ − $V_{OL}$, where NM$_H$ and NM$_L$ represent the high- and low-state noise margins, respectively, and $V_{IL}$ and $V_{IH}$ are the input voltages at which the slope of the voltage transfer curve is −1, whereas $V_{OH}$ and $V_{OL}$ are the corresponding output voltages (Supplementary Fig. 13). The calculated total noise margin of the CFET is around 83% at $V_{DD} = 3$ V. The CFET devices can operate at low power consumption, as shown in Extended data Fig. 3, where the peak power consumption is 518 pW for $V_{DD} = 1$ V. In addition, the dynamic inverting performance was also investigated by applying an a.c. voltage on the input terminal. Supplementary Fig. 14 displays the input voltage sequence and the corresponding output signals, clearly showing the output states are opposite to the input signals. It is worth mentioning that, as depicted in Extended data Fig. 4, the electrical property of the device shows minimal degradation within 12 months at room temperature in air, highlighting the long-term stability of our vdW interfacial doping strategy. Among those MoS$_2$–CrOCl FETs tested, the best performance exhibits a high on/off ratio reaching $10^6$ and hole mobility of approximately 425 cm$^2$ V$^{-1}$ s$^{-1}$ at room temperature (see Supplementary Fig. 15), indicating that our doping method gives better device performance compared with previous reports[11,18–28], as shown in Fig. 2f.

## 3D-integrated 2D logic over 10 vdW layers

In the following, by vertically integrating these VIP-FETs described in the previous text, we benchmark the bottom-up scaling of complementary logic circuitry with a true 3D architecture up to 14 vdW-stacked layers. We take a four-transistor SRAM (4T-SRAM) and a four-transistor NAND (4T-NAND) logic as examples. The schematic images of these devices can be arranged in a 3D vertical manner, as shown in Fig. 3a,b. It consists in building blocks of two n-type and two p-type FETs. When realized by the fabrication of VIP-FETs as devised in this work, a NAND logic can be further illustrated in the schematics in Fig. 3c, where the MoS$_2$ layer (blue), with and without interfacial coupling layer of CrOCl (pink), are the p- and n-type semiconducting channels, with few-layered h-BN (not shown) serving as gate dielectric and few-layered graphene (black) as gate electrode. Inputs A and B, supply voltage $V_{DD}$ and the ground electrode of the NAND logic are labelled correspondingly in Fig. 3b,c. The detailed fabrication process of the final devices can be found in Extended data Fig. 5.

Figure 3d displays the bright-field scanning transmission electron microscopy image of the cross-section of a typical vertically integrated NAND gate based on MoS$_2$ VIP-FETs. It is clearly seen that all the functional layers illustrated in Fig. 3c are realized by vdW layers, with each FET spaced by an h-BN dielectric spacer. A total of 14 vdW layers are included in the NAND logic device, as labelled in Fig. 3d. Zoomed-in STEM images in Fig. 3e,f further illustrate atomically sharp interfaces between the MoS$_2$, CrOCl and h-BN layers. Notice that the distance between MoS$_2$ and CrOCl is about 0.2 nm, which is taken as a reference for our DFT calculations. The corresponding electron energy loss spectroscopy (EELS) mappings (Fig. 3g) also confirm a clean vdW heterostructure with high-quality interfaces.

We achieved the milestone of vertical integration of more than ten layers of 2D complementary FETs and demonstrated such a 3D-integration of 2D semiconductors at the device level. Figure 4a shows the logic truth table of such a typical 14-vdW-layer 3D NAND

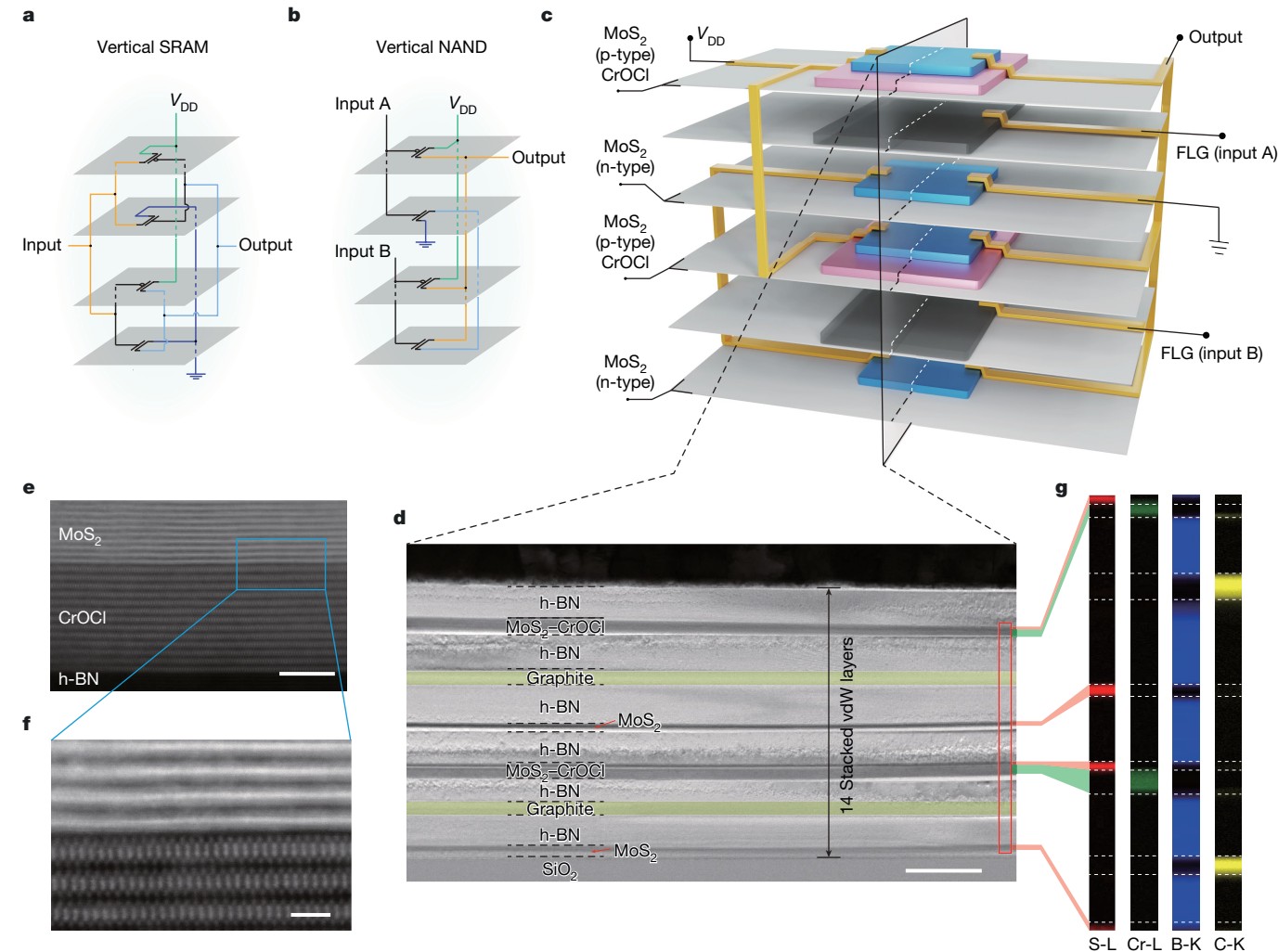

**Fig. 3 | 3D-integrated logic gates with more than ten vdW layers.** **a**,**b**, Illustrative schematic images of logic gates arranged in a 3D manner, leading to a SRAM (**a**) and a NAND gate (**b**). **c**, Artist rendering of the 3D-integrated NAND gate based on $MoS_2$ VIP-FETs, which consists of building blocks of four transistors, with two n-type and two p-type FETs. **d**, Bright-field scanning transmission electron microscopy image of the cross-section of a typical 3D-integrated NAND gate, with the stacking order for the 14 constituent vdW flakes indicated at each layer. The two graphite gate layers are rendered with false colour in yellow, for better visibility. **e**, High resolution high-angle annular dark-field STEM image of a typical $MoS_2$–CrOCl interface encapsulated between two h-BN layers. **f**, Zoomed-in image of the blue-boxed area in **e**. **g**, EELS mappings of the boxed area in **d**, which highlights the distribution of S, Cr, B and C in each layer. The EELS maps are elongated along the horizontal direction for better visualization. Scale bars, 100 nm (**d**), 5 nm (**e**), 1 nm (**f**).

gate operation. Figure 4b exhibits a dynamic performance for the device, where two different input voltages, $V_{in-A}$ and $V_{in-B}$, are fed with a rectangle wave in a time sequence but phase shifted, and the supply voltage $V_{DD}$ and ground are fixed at +3 V and −3 V, respectively, during the measurements. The device outputs a logic state '0' only if both input states are '1', firmly demonstrating the functionality of a NAND gate (more details on the performance of the 3D NAND gate are provided in Supplementary Fig. 16).

## Outlook of the VIP-FETs

Before going further in the demonstration of 3D logic, we discuss in the Methods further details (Supplementary Figs. 17–21 in Supplementary Note 2) of the interfacial-coupling-induced p-doping effect and the improvement of their electrical performance. The vertically free arrangement of CFETs demonstrated in our work can in principle be expanded into any 3D-integrated circuitries. For example, by re-wiring the vertically piled up four transistors, functionalities of SRAM with 14 vdW layers can also be realized, as shown in Fig. 4c,d (more details can be found in Extended data Figs. 6 and 7). To show the universality, statistics of field-effect curves of p-type $MoS_2$ and $MoSe_2$ are shown in Fig. 4e. Characterizations of output performance of a typical device and the effects of channel length are illustrated in Fig. 4f,g (see Methods).

The interfacial-coupling-induced p-doping and the resulting VIP-FETs are key techniques invented in this study and are conceptually suitable for the vertical scaling of future 2D semiconducting complementary metal–oxide–semiconductor circuits. To visualize the envisioned picture, we compare side-by-side the 4T-SRAM devices based on the planar-FET, CFET and VIP-FET architectures, shown in the SEM images in Fig. 4h–j, with their technical illustrations in Fig. 4k–m, respectively. Clearly, their footprint sizes are sequentially decreasing from 4 unit areas to 1 unit area, with a z-dimensional accumulation from 3 vdW to 14 vdW layers. We can thus expect future 3D integration of 2D VIP-FETs as shown in Fig. 4n. And this makes the most distinct feature of our technique, though there are very interesting alternative ways of p-doping TMDs[9–28]. Nevertheless, we emphasize that there are yet further technical challenges that need to be addressed in the longer-term perspective for the 3D-integration of 2D semiconductors in future nanoelectronics at a large scale application. (We have

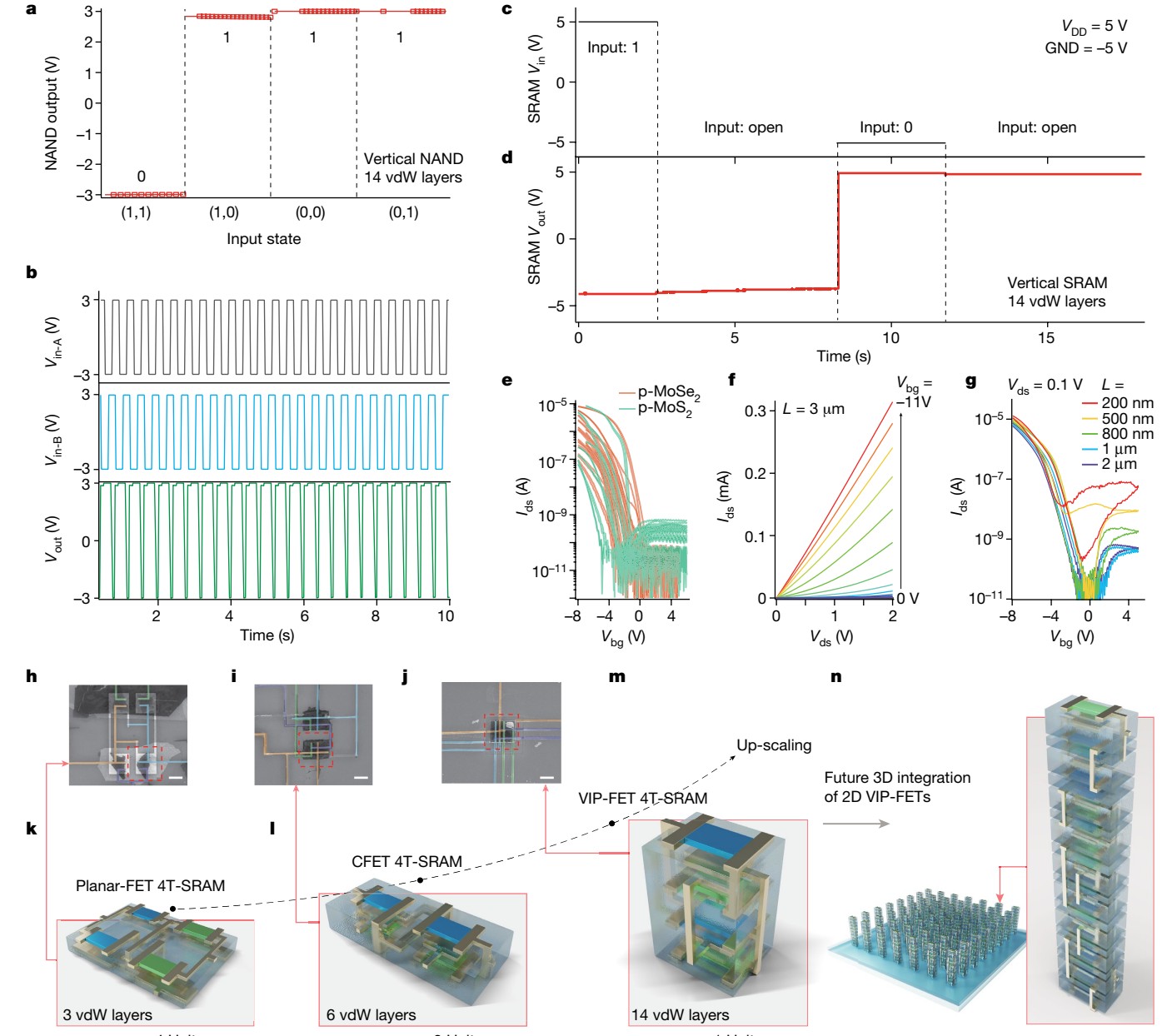

**Fig. 4 | Towards future 3D integration of 2D semiconducting complementary logic. a**, Logic operation of $V_{in}$–$V_{out}$ characteristics of a typical 3D-NAND device with 14 vertically integrated vdW layers as shown in Fig. 3c. A $V_{DD}$ of 3 V, GND of −3 V and $V_{in}$ = ±3 V for inputs A and B were applied during measurements. **b**, Dynamic NAND performance of the same sample as in **a** tested at a pulse period of 400 ms. **c**,**d**, show the input waveforms (**c**) and output level (**d**) of a typical 3D-SRAM device with 14 vertically integrated vdW layers. **e**, Field-effect curves of p-type MoSe₂ (red curves, with the statistics of 12 devices) and p-type MoS₂ (green curves, with the statistics of 8 devices) induced by coupling of the CrOCl substrates. $V_{ds}$ = 0.1 V is used in the measurements. **f**, Output performance of a typical p-type MoSe₂–CrOCl transistor, with a maximum $I_{ds}$ reaching 0.3 mA at $V_{ds}$ = 2 V. The channel length and width of the tested device are 3 µm and 5 µm, respectively. **g**, Logarithmic plot of transfer curves of typical p-type MoSe₂–CrOCl transistors, measured using the transfer length method (TLM). A $V_{ds}$ of 0.1 V was used in the measurement. **h**–**j**, False-colour SEM images of typical 4T-SRAM devices based on the planar-FET (**h**), CFET (**i**) and VIP-FET (**j**) architectures, with distinctive colour-coding for their $V_{DD}$, GND, $V_{in}$ and $V_{out}$ electrodes in green, purple, yellow and blue, respectively. **k**–**m**, art illustrations of the devices pictured in **h** (**k**), **i** (**l**) and **j** (**m**). Devices in **i**–**j** are plasma-etch patterned into square areas for visual clarity. **n**, An outlook of future 3D integration of 2D VIP-FETs, based on the technology described in this work. Scale bars, 10 µm (**h**–**j**).

become aware that recent efforts on wafer-scale vdW vertical CFETs have been reported during our submission[2,3]). These challenges include heat dissipation, large-area growth of p-type film (MoS₂–CrOCl heterostructure for example) with high uniformity suitable for industrial production, as well as the possible interlayer parasitic capacitance. We here briefly discuss the possibility of chemical-vapour-deposition growth of large scale CrOCl thin films in Supplementary Note 3 (Supplementary Figs. 22–26), and we also performed technology computer-aided design simulations of the parasitic capacitance due to the interlayer overlapping of electrodes, shown in Supplementary Note 4 (Supplementary Figs. 27–30).

Rather than pursue increased in-plane scaling of Si-based semiconducting devices, we took the route of vdW vertical scaling and have demonstrated its feasibility. We report a facile and stable p-type doping strategy for 2D semiconductors in order to have access to complementary FETs that are compatible with vertical integration. Our studies show

that by stacking TMDs ($MoS_2$, $WSe_2$ and $MoSe_2$) onto a vdW insulator CrOCl, the dominant carrier type can be effectively modulated from electrons to holes. First-principles calculations further reveal that such behaviour may originate from the strong vdW interfacial coupling. It indicates a cooperative effect of gate-tunable band alignment, charge transfer and $e-e$ interactions, which could essentially be different from conventional p-doping strategies for semiconducting TMDs. It is worthwhile noting that a similar mechanism already leads to a number of exotic quantum electronic states reported previously in graphene–CrOCl systems[41,42]. FETs fabricated based on this approach exhibit excellent electrical properties with on/off ratios reaching $10^6$, and the extracted room-temperature hole mobility reaches $425\ cm^2\ V^{-1}\ s^{-1}$ in $MoS_2$ with outstanding long-term air stability. Furthermore, based on our doping method, advanced 3D logic circuits, such as vertically constructed inverters with 6 vdW layers, NANDs with 14 vdW layers and SRAMs with 14 vdW layers, are implemented, confirming that our vdW interfacial-coupling-induced p-type doping may be a potent strategy for the design of future vertical scaling to realize ultra-high 3D-integration of advanced logic circuits.

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

## Methods

### Sample fabrication

The vdW few-layers of $MoS_2$ and CrOCl were obtained by mechanically exfoliating high-quality bulk crystals. The vertical assembly of vdW-layered compounds was fabricated using the dry-transfer method. Electron-beam lithography was done using a Zeiss Sigma 300 SEM with a Raith Elphy Quantum graphic writer. Input gates, as well as contacting electrodes, were fabricated with an e-beam or a thermal evaporator, with typical thicknesses of Cr and Au of approximately 5 nm and 50 nm, respectively.

### STEM characterizations

STEM characterizations were carried out on a monochromated Nion U-HERMES 100 microscope equipped with an alpha-type monochromator and a fifth-order aberration corrector and operated at 60 kV. Cross sections of as-prepared devices were made using a focused ion beam tool, Thermo Scientific Helios G4 CX DualBeam system cut at 30 kV and milled with the voltage gradually decreasing from 30 kV to 5 kV to minimize sample damage.

### Morphology tests

A Bruker Dimension Icon AFM was used for thickness and morphology tests, as well as KPFM characterizations. Optical images were collected by a Nikon LV-ND100 microscope.

### Electrical measurements

The high precision of current measurements of the devices were provided by measurement using a Cascade M150 probe station at room temperature, with an Angilent B1500A Semiconductor Device Parameter Analyzer. For the Dynamic NAND performance measurement in Fig. 4, a pulse train is adopted by using the waveform generator fast measurement unit.

### Density functional theory calculations

The first-principles calculations based on DFT were carried out with Vienna Ab initio Simulation Package with a projector augmented wave method[50,51]. The plane-wave energy cutoff was set to be 600 eV, the generalized gradient approximation by Perdew, Burke and Ernzerhof (PBE) was taken as the exchange-correlation potential[52]. As Cr is a transition metal element with localized 3$d$ orbitals, the so-called fully localized limit of the spin-polarized DFT + $U$ functional was adopted, as suggested by Liechtenstein and co-workers[53]. The on-site Hubbard $U$ = 3.0 eV parameter was used in the calculations, and the magnetic configurations of CrOCl is an antiferromagnetic order as shown in Supplementary Fig. 1f. This leads to a gap of 2.12 eV for CrOCl. For the work-function calculations, ten layers of $MoS_2$, five layers of CrOCl and a three + three-layer heterostructure were used, and the crystal structure was fully relaxed until the residual forces on the atoms were less than 0.01 eV Å$^{-1}$. To avoid any artificial interactions, 15 Å vacuum layers are added. In the calculation of heterostructure, a $7 \times 1$ supercell for the CrOCl and a $5\sqrt{3} \times 1$ supercell for $MoS_2$ were adopted. To construct such commensurate supercells, the lattice constant of CrOCl has been slightly compressed by 0.3% in one direction (changed from 3.2 Å to 3.188 Å) and has been slightly expanded by 1.5% in the other direction (changed from 3.88 Å to 3.94 Å). $MoS_2$ remains unstrained. A $12 \times 12 \times 1$ Γ-centred **k**-grid mesh for layered structures and a $6 \times 1 \times 1$ mesh for heterostructure were used. The DFT + D2 type of vdW correction was adopted to properly describe the interlayer interactions[54,55].

We note that although Heyd–Scuseria–Ernzerhof hybrid functional[56] calculation gives a gap of approximately 3 eV for CrOCl, which is more consistent with optical measurement[57], the calculation yields a highly over-estimated gap of 1.55 eV for $MoS_2$ (compared with experimental gap 1.29 eV (ref. 58)). Therefore, we adopt a PBE functional for $MoS_2$ and the DFT + $U$ method for CrOCl. When $U$ = 3.0 eV, the CBM of CrOCl is lower than that of $MoS_2$ (calculated using DFT + $U$ and PBE) by 0.465 eV based on work-function calculations, while the CBM of CrOCl is 0.37 eV lower than that of $MoS_2$ based on hybrid-functional-based work-function calculations. Therefore, the CBM energy positions obtained from our calculations are quite consistent with those obtained from hybrid functional calculations. The underestimated gap of CrOCl is attributed to the higher VBM energy position, which is always far (approximately 1–2 eV) below the chemical potential anyway, thus is irrelevant with regard to the mechanism discussed in this work.

Based on the above, we then calculated the band structures of $MoS_2$–CrOCl heterostructure without electric field (Supplementary Fig. 3a), as well as those of $MoS_2$ slabs and CrOCl slabs under electric fields (Supplementary Fig. 3b). The detailed evolution behaviour of the band edges of CrOCl and $MoS_2$ as a function of vertical electric field in these scenarios can be seen in Supplementary Table 1. When a negative electric field of approximately 0.15 V nm$^{-1}$ is applied (corresponding to the situation of negative bottom gate in our setup), the CBM of CrOCl is slightly lowered in energy, by 0.16 eV compared to the case without electric field, while the VBM of $MoS_2$ is dramatically increased in energy, by approximately 0.25 eV, such that it is only about 0.074 eV above the CBM of CrOCl. Meanwhile, the electron carriers that are transferred to the surface CBM of CrOCl are expected to be frozen to form an electronic crystal state driven by $e$–$e$ interactions, owing to the large effective mass and small carrier density. And the chemical potential of the heterostructure would be lowered and getting closer to the VBM of $MoS_2$ as illustrated in Fig. 1e, which thus can be easily tuned to be $p$-type upon further increasing the negative gate voltage. Therefore, it is very likely the subtle interplay among the gate-tunable band alignment, charge transfer and $e$–$e$ interactions that results in a chemical potential resident in the vicinity of $MoS_2$ VBM and eventually leads to the effective gate tuning from $n$-type to $p$-type carriers. Such a mechanism is believed to be universal and applicable to a wide range of 2D semiconducting materials (M. L. and J .Liu, manuscript in preparation).

### Electrical performance of the p-FETs

We note that, as shown in Supplementary Fig. 17, in the scenario of TMD–CrOCl, the thinnest working p-type doped $MoSe_2$ layer was found to be around 3.2 nm (about four layers). This might be due to the fact that thinner TMDs have larger band gaps with lower VBMs, which do not fulfil the required Fermi level down-shift as calculated by our DFT results, as also illustrated in Supplementary Fig. 4. We did find that other replacements, such as few-layered $Cr_2Ge_2Te_6$, can effectively dope monolayered $MoS_2$ into a p-type FET, as shown in Supplementary Fig. 19. In this case, even the on-state threshold voltage $V_{th}$ can be tuned to different values as compared to the value in TMD–CrOCl devices. Further, TMD–CrOCl FETs with different gate dielectric materials have also been fabricated, showing remarkable p-type field-effect behaviour, as shown in Supplementary Fig. 20. Our investigations also reveal that using a delicate annealing process with Ti–Au electrodes in an Ar–$H_2$ (30:4 sccm) mixture, the output currents of both p-type $MoS_2$–CrOCl and $MoSe_2$–CrOCl can be significantly improved (Supplementary Fig. 21) compared to the typical Cr–Au contacted device shown in Fig. 2a.

### Characterization of contact resistance

It is important to have an estimation of temperature dependence of the hole mobility of these TMD–CrOCl systems. Taking CrOCl-coupled few-layer $MoS_2$ and $MoSe_2$, for example, we found a significant suppression of the hole mobility upon cooling below 200 K (Extended data Fig. 8). This suggests that, although the $I$–$V$ curves are rather linear (Fig. 4f) in the $V_{ds}$ range of ±0.1 V at room temperature, there is still a tiny contact barrier, which is detrimental in low-temperature performance. TLM measurements show that this barrier is detrimental when the channel is approaching sub-200 nm, as illustrated in Fig. 4g.

Nevertheless, it is the best performance among TMD-based p-type transistors, to our knowledge. The contact resistance was determined to be 8.8 KΩ μm using the TLM, as shown in Extended data Fig. 9. Notice that to obtain ultrascaled sub-50-nm channel lengths, a higher precision lithography tool may be needed. In the scenario of very small samples, manual alignment will also pose limitations. Larger-sized films would be preferable for potential practical applications.

## Data availability

The data that support the findings of this study are available via Zenodo at https://doi.org/10.5281/zenodo.10262243 (ref. 59).

## Code availability

The codes used in theoretical simulations and calculations are available from the corresponding authors upon reasonable request.

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

**Acknowledgements** We thank X. Jiang and S. Wei for discussions. This work is supported by the National Key R&D Program of China (grant nos. 2022YFA1203903, 2023YFF1500600, 2019YFA0307800 and 2022YFA1203902) and the National Natural Science Foundation of China (NSFC) (grant nos. 92265203, 12104462, 12250007, 12034011, U23A6004, 92263203 and 52188101). Z.H. acknowledges the support of the Fund for Shanxi "1331 Project" Key Subjects Construction. J.Z. acknowledges the Innovation Program for Quantum Science and Technology (grant no. 2021ZD0302003) and the XPLORER Prize. This research benefited from resources and support from the Electron Microscopy Center at the University of Chinese Academy of Sciences. K.W. and T.T. acknowledge support from the JSPS KAKENHI (grant nos. 20H00354 and 23H02052) and World Premier International Research Center Initiative (WPI), MEXT, Japan. W.Z. acknowledges support from Beijing Outstanding Young Scientist Program (BJJWZYJH01201914430039) and CAS Project for Young Scientists in Basic Research (YSBR-003). J. Liu acknowledges the support from the Science and Technology Commission of the Shanghai Municipality (grant no. 21JC1405100).

**Author contributions** Z.H., Hanwen Wang, X.L. and Y.H. conceived the experiment and supervised the overall project. Y.G. and Hanwen Wang carried out device fabrication and electrical measurements. X.Z., B.C. and J.C. contributed to electrical measurement of NANDs. D.S. supported sample fabrication. P.G. and Y.Y. performed synthesis of bulk CrOCl crystals. Z.L., X.L., C.W. and W.Z. carried out STEM characterizations. K.W. and T.T. provided high quality h-BN bulk crystals. Hai Wang, W.L., K.Y. and Yingjia Liu helped in sample fabrication. Z.H., Y.G. and Hanwen Wang analysed the experimental data. M.L. and J. Liu performed first-principles calculations, as well as modelling, while Yueyang Liu, P.L., J. Li and X.-Q.C. contributed to early versions of DFT calculations. Z.W. and R.W. performed the technology computer-aided design modelling and simulations. B.Z. and Y.H. carried out chemical-vapour-deposition growth of CrOCl thin films. C.Q. and J.Z. contributed to Raman measurements and participated in discussions. The manuscript was written by Hanwen Wang and Z.H. with discussion and inputs from all authors.

**Competing interests** The authors declare no competing interests.

**Additional information**
**Correspondence and requests for materials** should be addressed to Xiuyan Li, Yanglong Hou, Wu Zhou, Hanwen Wang or Zheng Han.

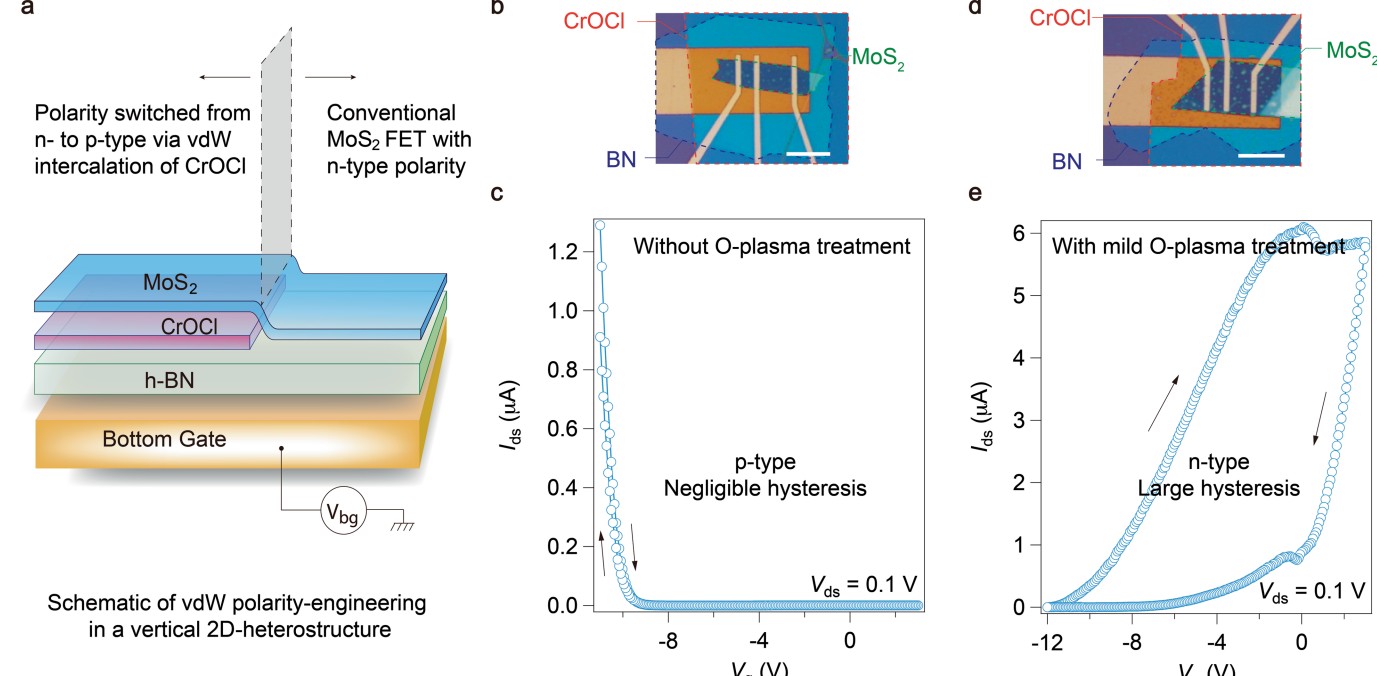

**Extended Data Fig. 1 | VdW polarity-engineering of MoS$_2$ on CrOCl.**
(a) Schematic drawing of the vdW polarity-engineering in a vertical 2D heterostructure. Here the structure is illustrated into two parts: left, MoS$_2$ semiconducting channel is placed on a few-layered CrOCl that is intercalated between MoS$_2$ and the supporting h-BN; and right, MoS$_2$ semiconducting channel is in contact directly with the top surface of h-BN. The drawing corresponds to the scenario of a typical experimental device as shown in the inset of Fig. 1a in the main text. The left part scenario will be further discussed here. (b) Optical micrograph of a typical MoS$_2$/CrOCl device with a pristine surface of CrOCl (i.e., freshly exfoliated and without O-plasma treatment). (c) Field effect curve of the CrOCl-supported MoS$_2$ device, measured at $V_{ds} = 0.1\,$V at room temperature. The device exhibits a typical p-type behavior, with negligible hysteresis (trace and re-trace are recorded as indicated by the black solid arrows). In this case, the vdW polarity-engineering works perfectly as the conventional n-type behavior of a MoS$_2$-FET is switched into p-type by simply a touch of few-layered vdW insulator (due to strong interfacial couplings). It is noteworthy that, as shown in (d), when the surface of CrOCl flake is treated by a mild O-plasma (50 Watt, 180 sccm flow rate of oxygen, with an Aluminum etching tunnel in a Yamato-PR 500 plasma reactor) for 30 s before stacking MoS$_2$ on top of it, the transfer curve of the resulted device is then very hysteric and is of n-type (instead of p-type), even the macro-scale structures look the same for both devices in (b) and (d). It thus speaks highly the importance of a clean/pristine interface, which plays a key role in the vdW polarity-engineering process. Scale bar in (b) and (d) are 10 µm.

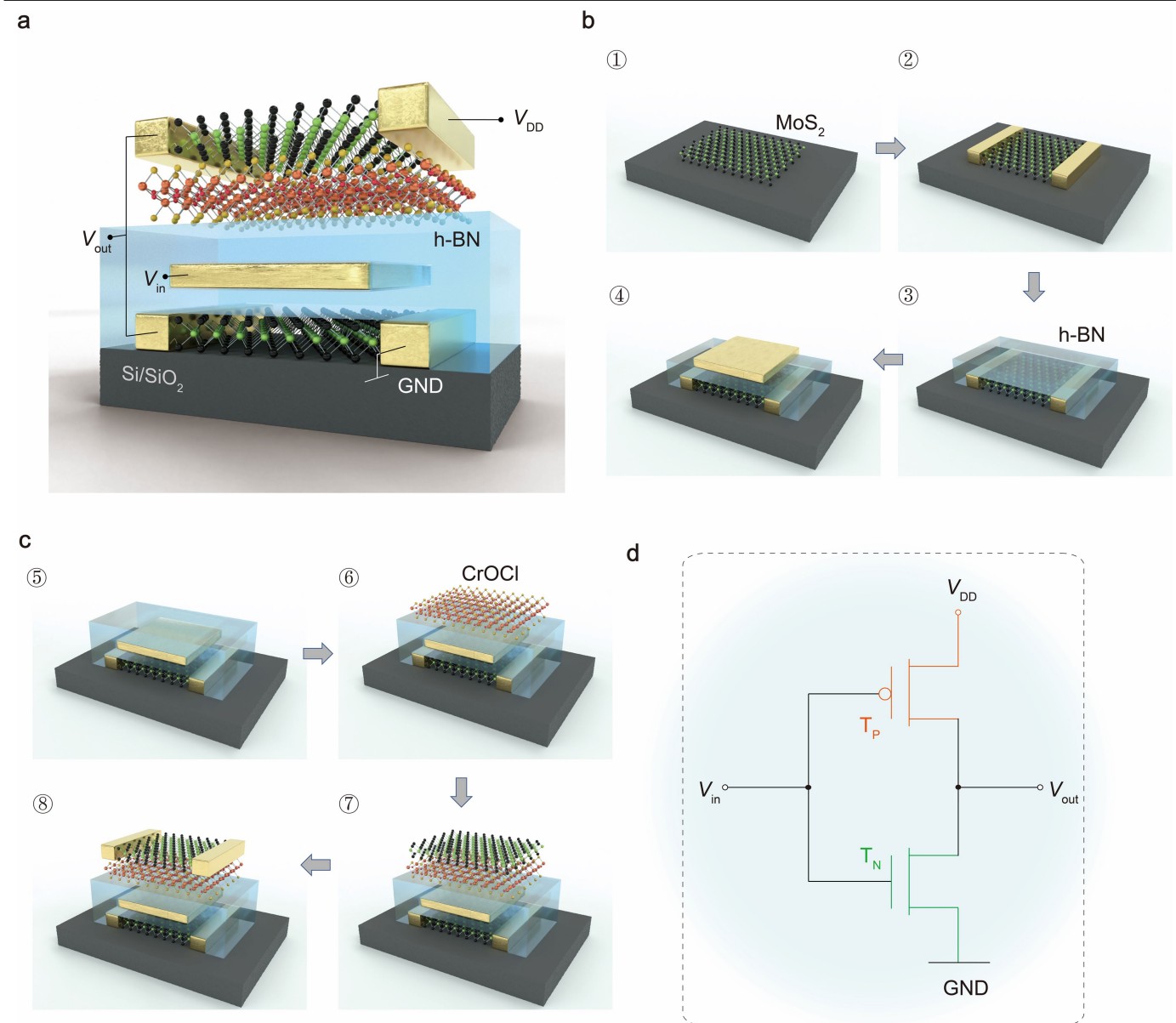

**Extended Data Fig. 2 | Six-vdW-layer vertically assembled inverter.**
(a) Art view of the MoS$_2$-CFET inverter using the vdW interfacial doping scheme described in this work. The CFET consists of 2 FETs with n- and p-type polarity for the MoS$_2$ channel without (the bottom one) and with (the top one) CrOCl intercalation, respectively. (b) Illustration of the fabrication process of the n-type MoS$_2$-FET part of the CFET: 1) A flake of few-layered MoS$_2$ was exfoliated onto a SiO$_2$/Si substrate; 2) Metal electrodes served as source and drain were deposited by standard e-beam lithography followed by evaporation of Cr (5 nm)/Au (30 nm); 3) A spacing/dielectric layer of few-layered h-BN was transferred on top of the MoS$_2$ channel by a polypropylene carbonate stamp; 4) Top-gate electrode was evaporated. (c) Illustration of the fabrication process of the p-type MoS$_2$-FET part of the CFET: 5) Following step 4, another spacing/dielectric layer of few-layered h-BN was transferred by a polypropylene carbonate (PPC) stamp; 6-7) few-layered MoS$_2$ interfaced with few-layered CrOCl as a p-channel were transferred by a polydimethylsiloxane (PDMS) stamp; 8) Metal electrodes were evaporated to form the source and drain in the top semiconducting channel. (d) The schematic diagram of the CFET inverter.

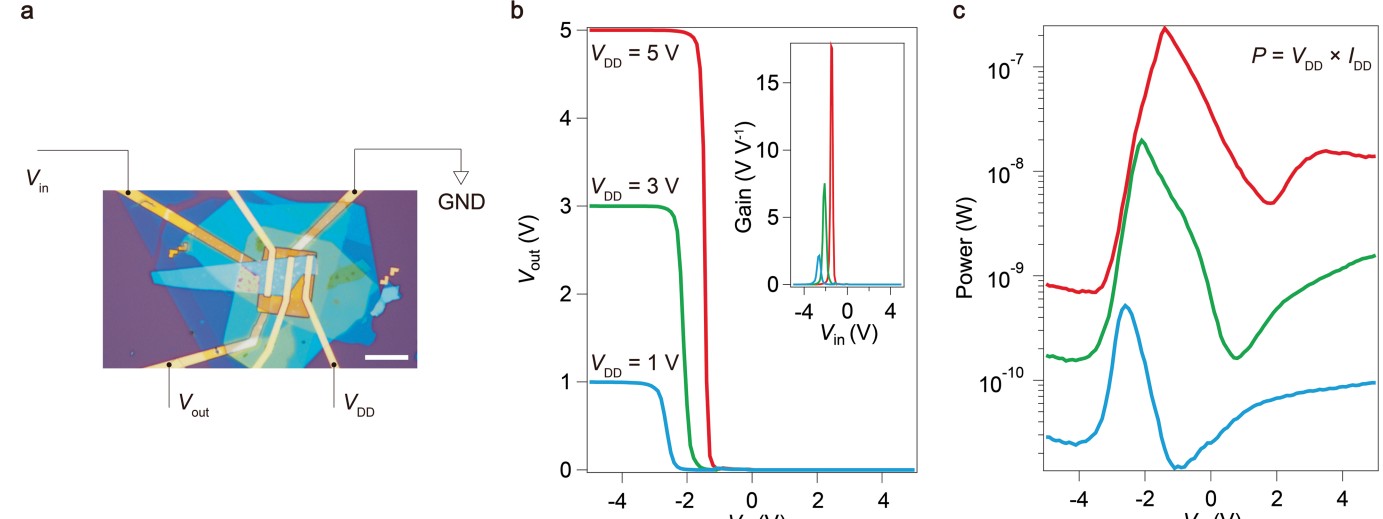

**Extended Data Fig. 3 | Power consumption of a typical MoS₂ CFET inverter.**
(a) Optical micrograph image of a typical tested CFET device (sample #S1b), constructed with the same structure as the illustration in Extended Data Fig. 2a. Scale bar is 10 μm. (b) Voltage transfer characteristics of the CFET inverter in (a) at different $V_{DD}$. The corresponding voltage gains for each voltage transfer curve are shown in the inset in (b) using the same color code. (c) Power consumption (defined as $V_{DD} \times I_{DD}$) of the CFET are shown as a function of $V_{in}$ for different $V_{DD}$. For example, the peak power consumption is found to be ~ 518 pW for $V_{DD} = 1$ V. Same color codes for the data are used in (b) and (c).

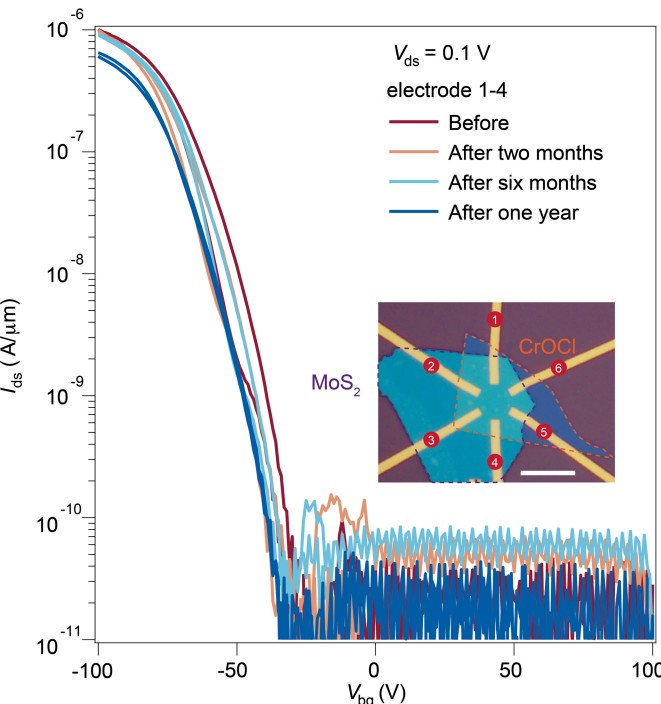

**Extended Data Fig. 4 | Air stability of the p-doped devices.** To examine the air stability of the as prepared p-FETs of MoS$_2$/CrOCl heterostructure, field effect curves of a typical device before (red line), after two months (yellow line), after six months (light blue line), and after 12 months (dark blue line) exposed to air, are shown. It is seen that the performance of the device exhibits negligible changes over a time duration of one year.

**a**

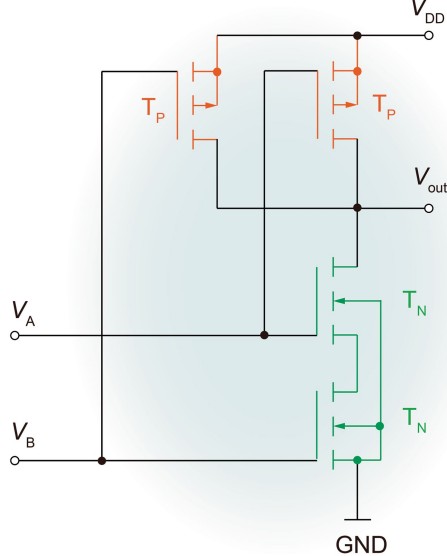

**b**

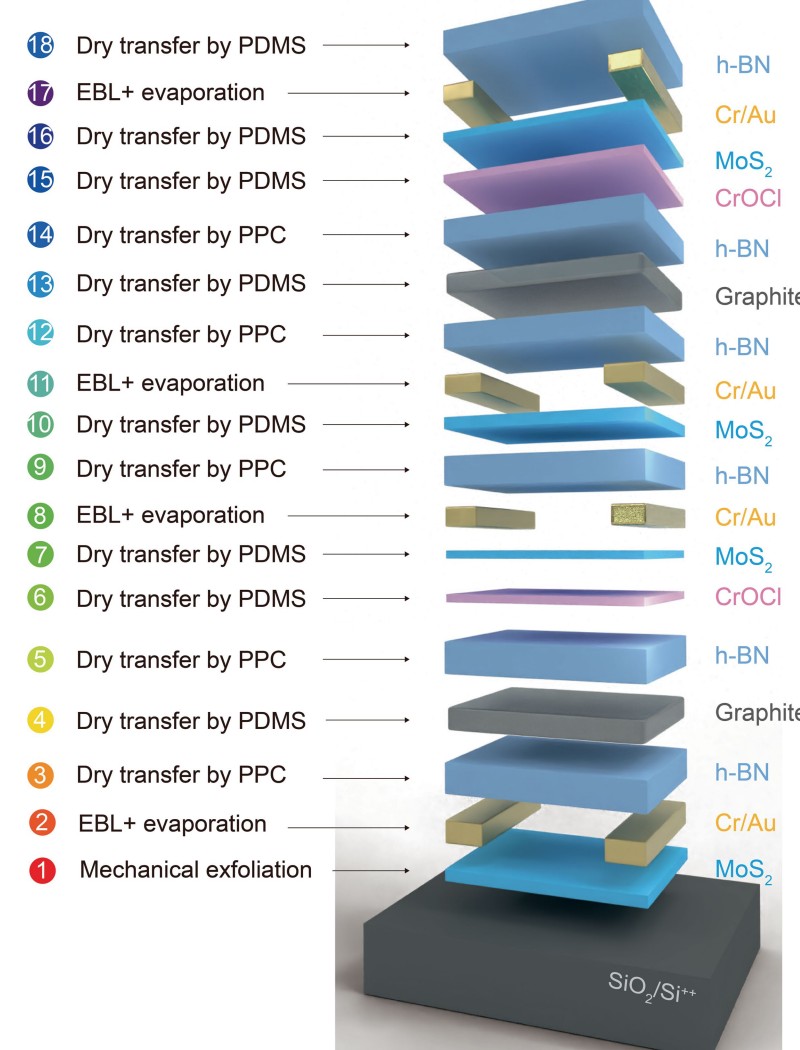

18 Dry transfer by PDMS → h-BN
17 EBL+ evaporation → Cr/Au
16 Dry transfer by PDMS → MoS$_2$
15 Dry transfer by PDMS → CrOCl
14 Dry transfer by PPC → h-BN
13 Dry transfer by PDMS → Graphite
12 Dry transfer by PPC → h-BN
11 EBL+ evaporation → Cr/Au
10 Dry transfer by PDMS → MoS$_2$
9 Dry transfer by PPC → h-BN
8 EBL+ evaporation → Cr/Au
7 Dry transfer by PDMS → MoS$_2$
6 Dry transfer by PDMS → CrOCl
5 Dry transfer by PPC → h-BN
4 Dry transfer by PDMS → Graphite
3 Dry transfer by PPC → h-BN
2 EBL+ evaporation → Cr/Au
1 Mechanical exfoliation → MoS$_2$

SiO$_2$/Si$^{++}$

**Extended Data Fig. 5 | VIP-FET based 3D NAND logic made of over-10-layer vdW heterostructure.** (a) The schematic diagram of the conventional planar 4-transistor (4T) NAND logic circuit (whose 3D inter-connected version is illustrated in Fig. 3b-c in the main text). The 2 n-type and 2 p-type transistors are labelled as $T_N$ and $T_P$, respectively. It is seen that in a conventional planar scheme, the NAND logic takes up 4 unit areas, while the stacked-up version by 3D integration will shrink the total effective channels into 1 single unit area. The vdW polarity-enginnered VIP-FET thus is particularly suitable for this vertical integration approach. As shown in (b), the cartoon illustration of the detailed vdW layers with a vertical stacking sequence (14 vdW layers in total, with 2 p-FETs and 2 n-FETs) is given, in order to achieve the same function as the planar NAND in (a) but rather fabricated in a 3D manner. Such 3D NAND structure with 14 vdW layers is further realized by interconnecting the inter-layer electrodes according to the architecture illustrated in Fig. 3c in the main text. Notice that $T_N$ and $T_P$ in the vertical 3D NAND in (b) are realized by MoS$_2$ and MoS$_2$/CrOCl, respectively.

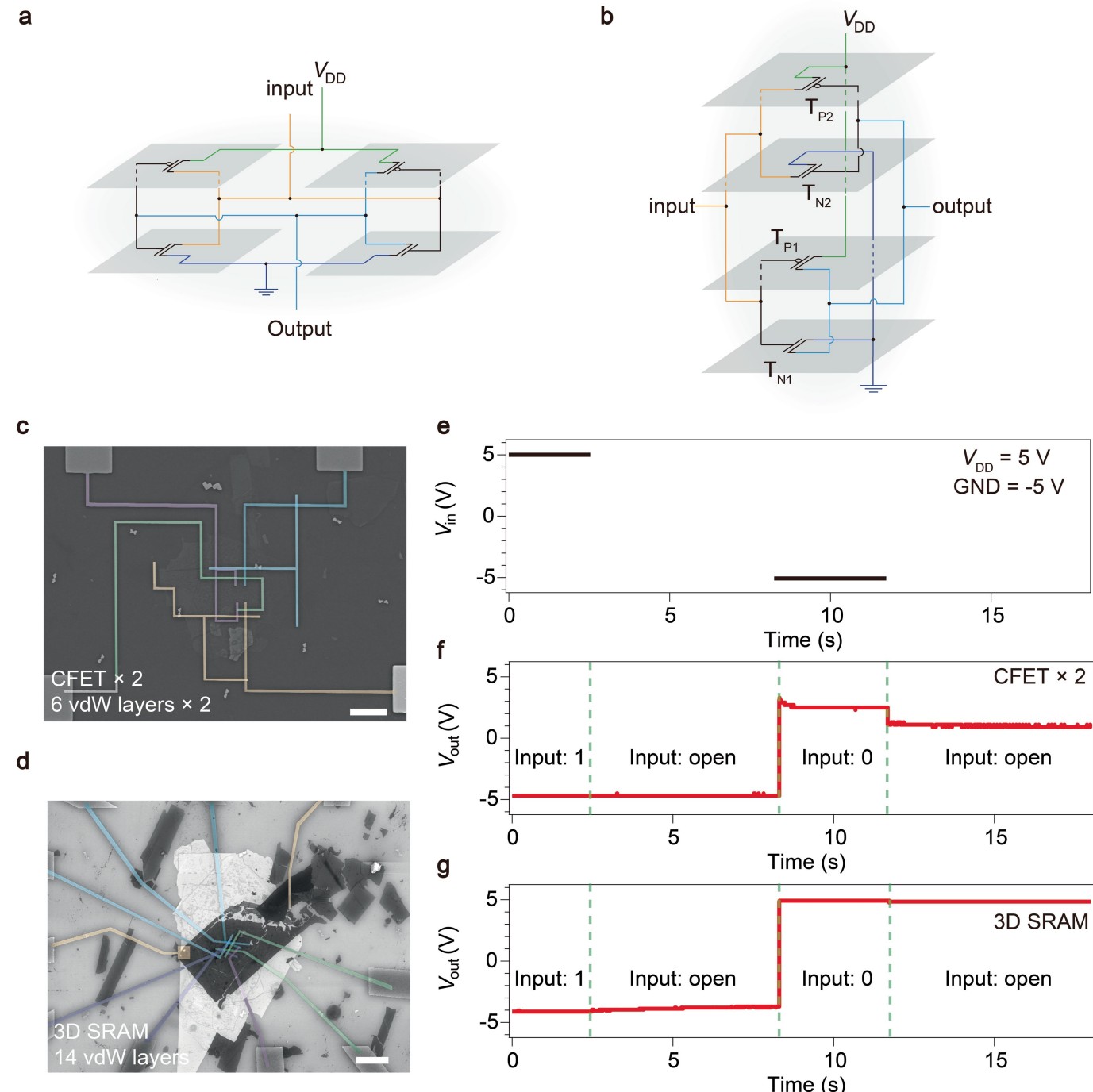

**Extended Data Fig. 6 | 3D integrated 4T-SRAM.** As another example of 3D logic using the vdW polarity-engineering method described in this work, we demonstrate the 3D vertical 4T-SRAM here. (a) Schematic diagram of a 4T-SRAM constructed using 2 sets of vertical CFETs with 6 vdW layers in each set. (b) Schematic diagram of a 4T-SRAM constructed by vertically stacking all the 4 complementary transistors within a single unit area (illustrated by the gray shadowed area). Notice that here the diagram of vertical 4T-SRAM is a duplicate of Fig. 3a in the main text, in order to have a side-by-side comparison with the 2-sets CFET scenario in (a). (c) and (d) are false-colored SEM images of the real devices using the design in (a) and (b), respectively. Scale bars are 10 μm. (e) The input wave form of the SRAM with "0" and "1" levels as triggers to operate the write and erase function for the SRAM device. (f) and (g) are output waveforms of the SRAM for typical devices as shown in (c) and (d), respectively. Data in (g) is repeated from Fig. 4d in the main text.

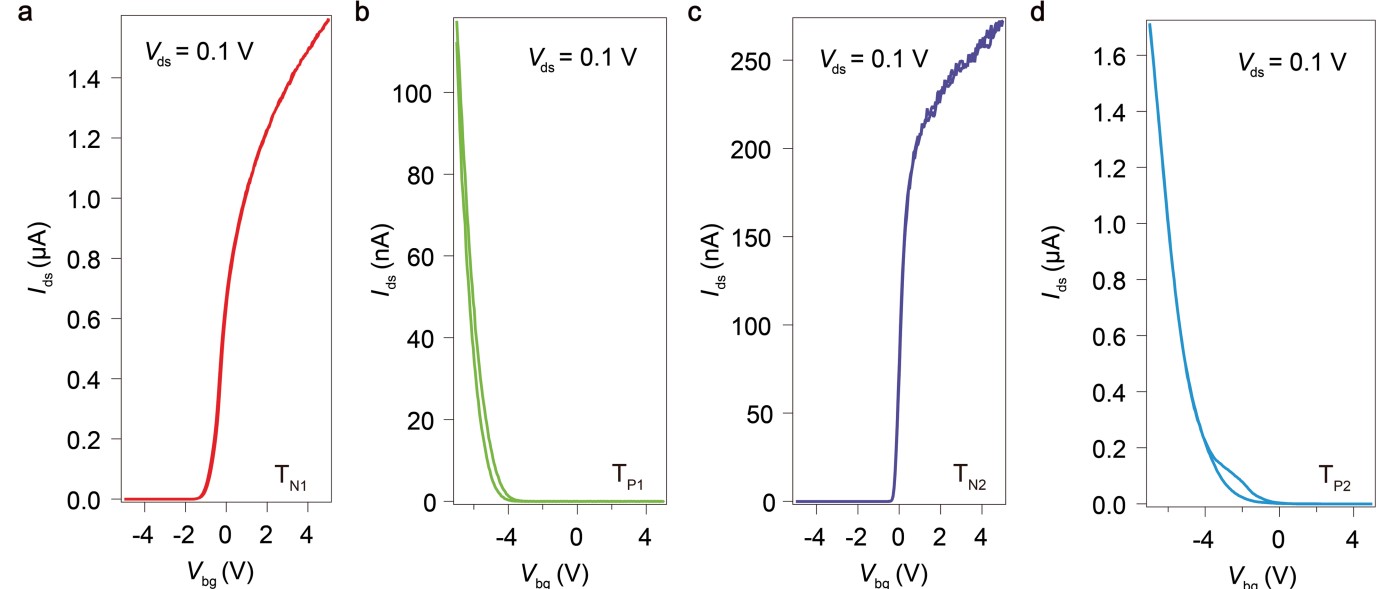

**Extended Data Fig. 7 | Field effect curves of each transistor in the vertical 4T-SRAM.** For a typical vertical 4T-SRAM device as shown in Fig. 4j in the main text (or in Extended Data Fig. 6d), we characterized each transistors that are inter-connected for the test of SRAM. The labels (i.e., $T_{Ni}$, i = 1, 2; and $T_{Pj}$, j = 1, 2) of each transistor follows the convention indicated in Extended Data Fig. 6b.

Field effect curves of them are shown respectively: (a) the bottom $MoS_2$ n-FET $T_{N1}$, (b) the bottom $CrOCl/MoS_2$ p-FET $T_{P1}$, (c) the top $MoS_2$ n-FET $T_{N2}$, and (d) the top $CrOCl/MoS_2$ p-FET $T_{P2}$. All data obtained at room temperature with a source-drain voltage $V_{ds} = 0.1$ V.

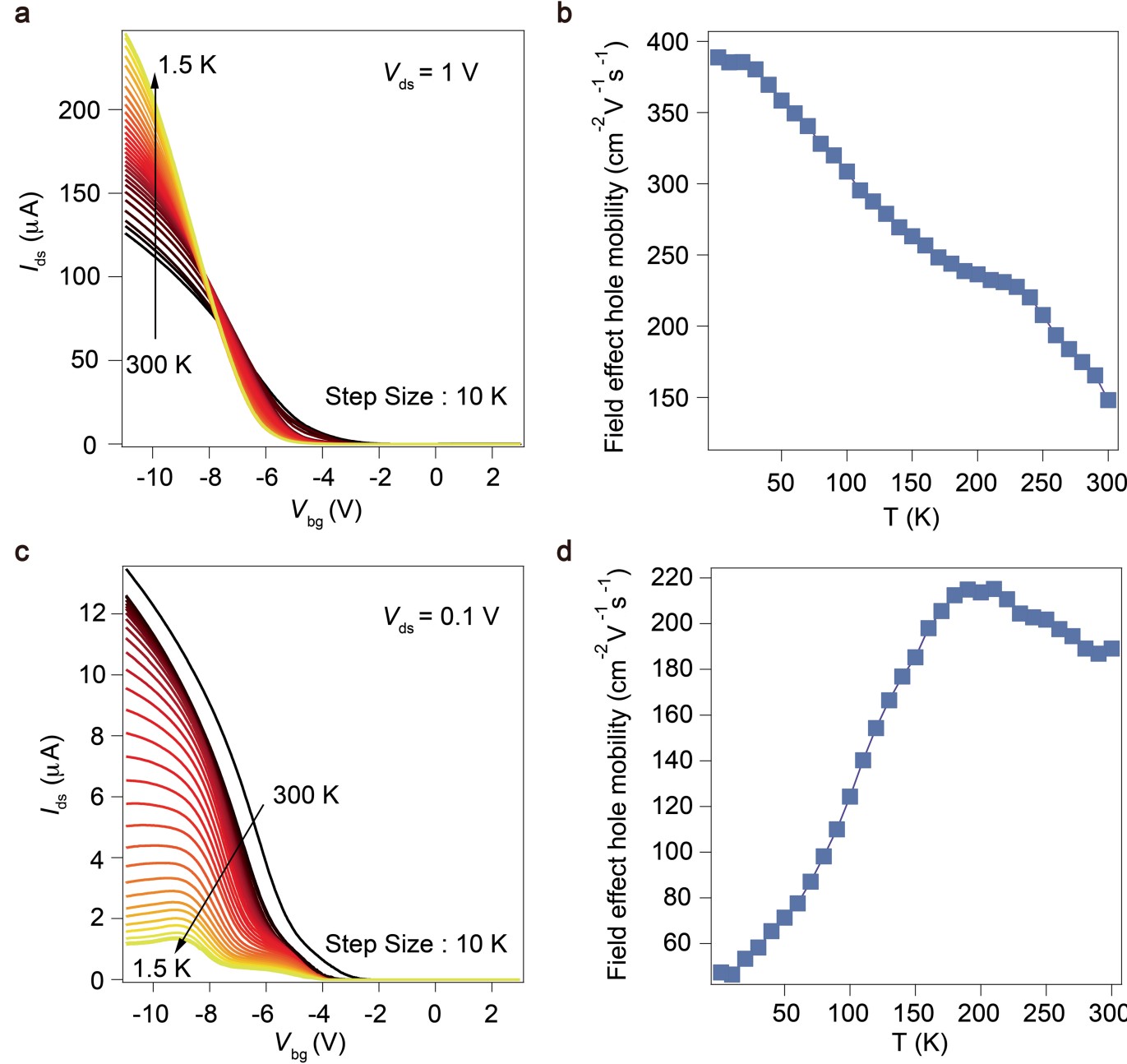

**Extended Data Fig. 8 | Transfer characteristics of a typical MoS$_2$/CrOCl p-FET at different temperatures.** Transfer characteristics measured at (a) $V_{ds}$ = 1 V and (c) $V_{ds}$ = 0.1 V, with their corresponding extracted field effect mobility (both estimated at a fixed gate voltage of $V_{bg}$ = −10 V) as a function of temperature shown in (b) and (d), respectively. We have to admit that the contact resistance of these p-FETs has not yet reached a well-defined Ohmic contact, and there seem to still exist a contact barrier. As can be seen that while the device at high bias ($V_{ds}$ = 1 V) show metallic behavior (higher conductivity at lower temperatures, at large-enough negative gate voltages) with increasing hole mobility upon cooling (Extended Data Fig. 8a-b), its hole mobility at low bias ($V_{ds}$ = 0.1 V) significantly decreases at temperatures below 200 K (Extended Data Fig. 8c-d). There is still room for further optimization in the future development of Ohmic contacts at low temperatures for the p-FETs devices based on the technique reported in this work.

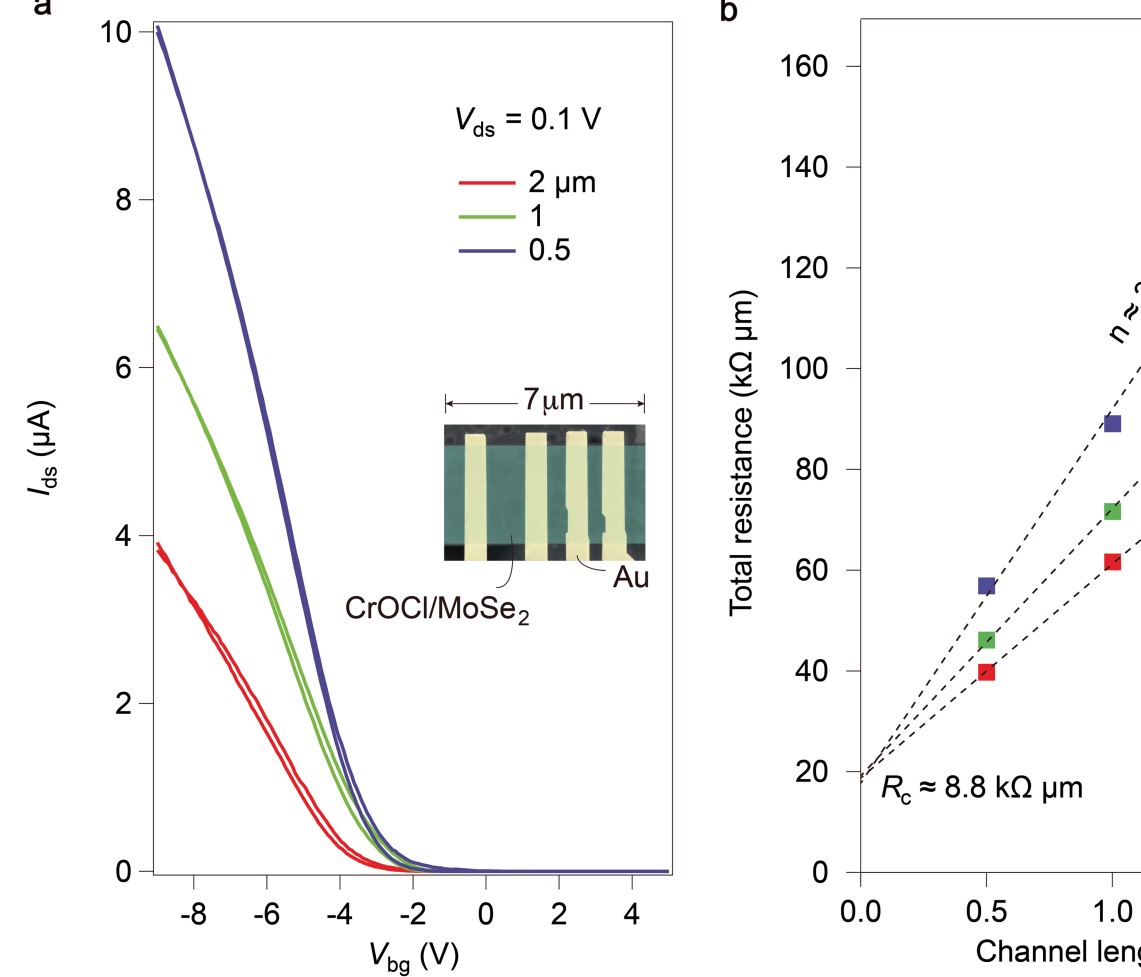

**Extended Data Fig. 9 | Estimation of contact resistance $R_c$ in a typical MoSe$_2$/CrOCl device.** We performed contact resistance examinations in vdW polarity-engineered MoSe$_2$ p-FET at room temperature using the TLM method. (a) shows the transfer curves of a typical TLM structure with the channel length $L_c$ ranging from 2 μm to 500 nm. Inset shows the false-colored SEM image of the device. (b) The contact resistance can be estimated by reading the intercept with the *y*-axis extrapolated from the two-terminal resistance of different channel length at carrier densities of $5.0 \times 10^{12}$ cm$^{-2}$ (red solid square), $4.5 \times 10^{12}$ cm$^{-2}$ (green solid square), and $3.7 \times 10^{12}$ cm$^{-2}$ (blue solid square), respectively. $R_c$ in the current case is therefore estimated to be about 8.8 kΩμm.