## [Peer Review File · Nature]

Manuscript Title: Van der Waals polarity-engineered 3D integration of 2D complementary logics

Reviewer Comments & Author Rebuttals

Reviewer Reports on the Initial Version:

Referees' comments:

Referee #1 (Remarks to the Author):

I am impressed by the two points delivered by this manuscript. Firstly, the authors found a novel route to p-dope the 2D transitional dichalcogenides (TMDs), mainly MoS₂ which is conventionally filled with n-type carriers. It is a very neat doping strategy using another 2D insulator with a proper interfacial charge transfer, which can avoid the sacrifice of carrier mobility in the channel (in many cases such as molecular dressing, ion bombardment, they can also dope 2D semiconductors but end up with a much-degraded quality). Second point is that the authors modularized their n- and p-doped channels and managed to show a vertically integrated complementary logic circuits, up to a maximum of 14 layers of vdW building blocks. The demonstration of the free-design of CFETs in a 3D-interconnected manner is intriguing, and definitely an important demonstration so far in the 2D device community. Based on the above, I would like to congratulate the authors on the nice results which deserve to be published in first-tier journals such as Nature. The manuscript indicates the very well planned and nicely organized experiments. The paper is clearly structured and well written. I would like to draw attention to only a few points and hope the authors can answer following questions before my full recommendation to publish.

1, The first and most important question is: can the device demonstrated be scaled up to wafer-scale? In the 2D case, the film grown by CVD, ALD, or MOCVD has become the preferred material for device investigations in the industrial community. The 2D device community has also adopted this trend in recent years.

2, Regarding the DFT part. I think there is a significant room to strengthen up the calculations/simulations in this manuscript. By making a very simplified model of monolayer CrOCl and monolayer MoS₂, they claimed a band alignment so that electron charge transfer happens from MoS₂ to CrOCl, which eventually leads to a p-type MoS₂. I understand that it's not realistic to model a 30 nm heterostructure by DFT but at least a few layers are possible for computing. In addition, I don't understand why it gives a gap of only 0.5 eV in CrOCl, which is quite problematic and should be re-examined.

3, What other candidate materials can we consider to play the same role as that of CrOCl? It is important to elaborate or at least comment on this issue.

3, The authors claimed a hole mobility as high as 400 cm²/V/s, as far as I know, that is the highest value reported. How is this value calculated? How about the temperature dependence? Recently, it was reported that n-type MoS₂ can manifest an ON state current exceeding 100 μ A/ μ m under a bias

voltage of a few hundred mV. I am interested to know the comparison of the performances of the p-type FETs in this work with the state-of-the-art results for n-type FETs based on MoS₂.

4, The fact that stand-alone devices can be stacked up to 14 layers might already be the limit. Can the authors comment on the possibility that such 3D interconnected circuits can be fabricated in a wafer-scale in the future?

5, Is there any chance that the authors could demonstrate more complicated structures using the same technique of 3D integration of 2D semiconductors, such as SRAM and ring oscillators? Adding schematics and different 3D architectures to an outlook figure will make it more impactful.

Referee #2 (Remarks to the Author):

In the manuscript, "Toward Skyscrapers of 2D Semiconducting Complementary Logics" by Y. Guo et al., the authors have explored a new charge transfer p-doping technique for the fabrication of p-type 2D transistors, which ultimately yields vertically integrated NAND gates. The manuscript is suggestive of a new approach to designing vertically-stacked 2D complementary logic transistors. Despite the promising foundation of the study, the reviewer concludes that the manuscript lacks sufficient novelty and merit to justify publication in Nature. Detailed comments are provided below.

1. The presented charge transfer p-type doping method is not a novel contribution. Previous literature [1,2,3] has documented numerous such doping schemes that employ the interface of a 2D-TMD with a high work function material, achieving comparable, if not superior, doping concentrations. Certain p-2D-FET transistors crafted through such methods have exhibited higher performance metrics than the one presented in this manuscript.

2. Moreover, the p-type doping scheme explored in this study is not selective, thereby doping the entire source-channel-drain region. This is disadvantageous to electrostatic channel control as doping strength escalates. It is speculated that the weak electrostatic gate control at reduced channel thicknesses, with more effective doping, could be why the manuscript does not report the p-2D-FET performance with thinner layers (<4 channel layers). It is crucial to note that the primary advantage of 2D-TMDs for FET design is their superb gate electrostatic control facilitated by their atomic body thickness. Hence, this process must be displayed for relevant thinner body thicknesses, where 2D-FETs surpass Si (and/or Ge) counterparts.

3. In addition to the claimed novel p-type doping scheme, the paper illustrates the creation of stacked complementary transistor logic. However, this demonstration does not address the most significant challenges in vertical (3D) integration of 2D transistor logic. Primarily, the transistors developed in this paper are transfer-based, rather than directly grown, which is critical for a functioning 3D integration process flow. Furthermore, n- and p-2D-FETs lack a consistent gate topology (either top/bottom-gated; while top-gate preferred) crucial for equivalent, matched transistor performance.

4. Although the FET mobility and ON/OFF ratio are commendable, the ON-current is significantly low (~1-5 μA) for practical applications. This deficiency results from poor contacts, a large Schottky barrier height, and a lack of expansive doping schemes.

5. Lastly, the demonstrated NAND gate operates with input voltages ranging from -4V to 4V, but generates an output voltage of only 0V to 5V. This discrepancy impedes circuit design as one stage cannot power the subsequent stage.

References:

[1]: C.-S. Pang, T. YT Hung, A. Khosravi, R. Addou, Q. Wang, M. J. Kim, R. M. Wallace, Z. Chen, *Advanced Electronic Materials* 6, 1901304 (2020).

[2]: M. Yamamoto, S. Nakaharai, K. Ueno, K. Tsukagoshi, *Nano Letters* 16, 2720-2727 (2016).

[3]: P. Zhao, D. Kiriya, A. Azcatl, C. Zhang, M. Tosun, Y.-S. Liu, M. Hettick et al., *ACS Nano* 8, 10808-10814 (2014).

Referee #3 (Remarks to the Author):

Vertical 3D integration of 2D semiconductors is a promising technology that could overcome the limitations of silicon-based transistors. The authors have made a significant contribution to the field of 2D semiconductor integration by demonstrating a robust and universal method for controlling the carrier polarity of 2D semiconductors. The results are impressive, with transistors achieving room-temperature hole mobilities up to $\sim 425 \text{ cm}^2 \text{V}^{-1} \text{ s}^{-1}$, and on/off ratios exceeding 106. The devices are also air-stable for over six months, which is a significant advantage. The authors have also demonstrated the feasibility of vertically constructing inverters and NANDs with 6 and 14 vdW layers, respectively. These findings are important for the future of 3D integrated circuits, and I believe that this work will be of interest to a wide audience. However, there are still some important points that need to be clarified in order to be published in Nature.

Below are some comments.

1. Metal electrodes are formed in multiple layers to create a three-dimensional device structure. The thickness of the h-BN used as the insulator layer is 27.3 nm, but the metal electrodes are formed vertically, creating unwanted parasitic capacitors, and the capacitance will be high. A clear explanation and supporting data are required for this part.

2. The Raman spectra of MoS₂, MoS₂/CrOCl, and CrOCl are shown in Supplementary Figure 7. In the Raman spectrum of MoS₂/CrOCl, the author only describes the peak shift of the A_{1g} mode compared to MoS₂, but it appears that the E_{2g} mode is also shifted in the same direction and by a similar distance. The authors need to clarify this phenomenon.

3. The purpose and concept of this research are excellent. However, unless the area of the two-dimensional semiconductor is fundamentally large, it is difficult to replace the Si semiconductor. Authors should suggest ways to overcome these limitations.

Reply to referee #1:

General Comment. *I am impressed by the two points delivered by this manuscript. Firstly, the authors found a novel route to p-dope the 2D transitional dichalcogenides (TMDs), mainly MoS₂ which is conventionally filled with n-type carriers. It is a very neat doping strategy using another 2D insulator with a proper interfacial charge transfer, which can avoid the sacrifice of carrier mobility in the channel (in many cases such as molecular dressing, ion bombardment, they can also dope 2D semiconductors but end up with a much-degraded quality). Second point is that the authors modularized their n- and p-doped channels and managed to show a vertically integrated complementary logic circuits, up to a maximum of 14 layers of vdW building blocks. The demonstration of the free-design of CFETs in a 3D-interconnected manner is intriguing, and definitely an important demonstration so far in the 2D device community. Based on the above, I would like to congratulate the authors on the nice results which deserve to be published in first-tier journals such as Nature. The manuscript indicates the very well planned and nicely organized experiments. The paper is clearly structured and well written. I would like to draw attention to only a few points and hope the authors can answer following questions before my full recommendation to publish.*

Response:

We are very grateful for Reviewer #1 for her/his positive comments. Her/his comments are very helpful in terms of improving our manuscript. We present here answers to her/his questions in a point-to-point manner in the following, with related modified parts highlighted in blue color in the main text and/or Supplementary Information in the revised version of our manuscript.

References/figures appearing in this Response-to-Referees file will be re-indexed if any of them are added in the revised main text of our manuscript.

Comment 1. *The first and most important question is: can the device demonstrated be scaled up to wafer-scale? In the 2D case, the film grown by CVD, ALD, or MOCVD has become the preferred material for device investigations in the industrial community. The 2D device community has also adopted this trend in recent years.*

Response:

We thank the referee a lot for raising this question.

Indeed, large scale production of wafer scale 2D materials have been the pursuit for device investigations in the industrial community. In recent years, many reports also show the potential of wafer scale 2D semiconductors, especially the n-type TMDs.

As suggested by Referee #1, we fully agree that, whether the p-type TMD/CrOCl device demonstrated in this work can be scaled up to wafer-scale, is an important issue that should be discussed in our work. In the revised manuscript, we added a paragraph in the section of **Outlook of the VIP-FETs**, to emphasize this grand challenge, quoted below:

“These challenges include the heat dissipation, large- area growth of p-type film (MoS₂/CrOCl heterostructure for example) with high uniformity suitable for industrial production, as well as the possible interlayer parasitic capacitance. We here briefly discuss the possibility of chemical vapor deposition (CVD) growth of large scale CrOCl thin films in Supplementary Note 3 (Supplementary Figures 27-31), and we also performed Technology Computer-Aided Design (TCAD) simulations of the parasitic capacitance due to the interlayer overlapping of electrodes, shown in Supplementary Note 4 (Supplementary Figures 32-35)”.

And, we also added additional data in Supplementary Note 3, and Supplementary Figures 27-31 therein, to show the attempt of large-scale growth of CrOCl thin layers as a starting point of our future work toward large area TMD/CrOCl by CVD methods. We quote it below:

“Supplementary Note 3. Possibility of large-scale production of CrOCl thin films

One of the challenges for future applications based on the VIP-FETs described in this work is the scaling up of these devices. Apparently, large scale production of CrOCl thin layers can be a starting point. In this supplementary note, we discuss the possibility of chemical vapor deposition (CVD) growth of CrOCl thin films.

2D CrOCl nanoflakes were synthesized by an ambient pressure CVD method. The growth was conducted in a 1-m length, 1-inch outer diameter quartz tube heated by a one-zone furnace (MTI, OTF-1200X-III-C). The reaction principle for synthesizing CrOCl involves the hydrolysis of chromium trichloride. The specific reaction equation is as follows: $\text{CrCl}_3 + \text{H}_2\text{O} \rightarrow \text{CrOCl} + 2\text{HCl}$.

According to this reaction principle, we have devised the following experimental procedure: using chromium trichloride powder as the source of chromium and utilizing air as the source of water. In the initial stages, we employ compressed air as the source of air. Different quantities of CrCl₃ powder (Alfa, 99.99%) was used as the solid precursor which was placed under a stacked mica substrate in a quartz boat. Prior to the growth process, the furnace was purged by the flowing of 300 sccm high-purity Ar gas for 5 min. Then, 50-200 sccm Ar and 1-10 sccm Air was introduced into the CVD system. The furnace was heated to 600-800 °C and growth time was set at 10 min. However, regardless of temperature and gas flow rate adjustments, the resulting outcome remains chromium oxide nanoflakes, shown in Supplementary Figure 27. The results indicate that the water content flow still exceeds our expectations.

In order to reduce the water content to a level conducive for the growth of CrOCl during the experimental process, compressed air was no longer utilized. Instead, the air inlet was slightly opened to allow a minimal amount of atmospheric air to enter the reaction system. Furthermore, to achieve thinner nanoflake thickness, a spatial confinement approach for growth was employed as depicted in Supplementary Figure 28.

The specific experimental procedure was as follows: 50 mg of CrCl₃ powder was positioned before a stacked mica substrate in a quartz boat. Prior to the growth process, the furnace was purged by flowing 300 sccm of high-purity Ar gas for 5 minutes. Subsequently, 200 sccm of Ar was introduced into the CVD system with an incomplete seal of the quartz tube to permit a small amount of air to enter. The furnace was then heated to 780°C for 78 minutes, with the growth time set at 10 minutes. Once the heating process was concluded, the furnace was allowed to naturally cool down. The CrOCl nanoflakes would be obtained between the two mica pieces. An optical image of the synthesized CrOCl nanoflakes is depicted in Supplementary Figure 29.

The nanoflakes exhibited a rhombic morphology. To characterize the phase of the nanoflakes, Raman and XRD analyses were conducted. As shown in Supplementary Figure 30 (a), the Raman peaks at 209.5 cm⁻¹, 418.5 cm⁻¹, and 460.5 cm⁻¹ corresponds to the A_g mode of CrOCl. The XRD patterns are shown in Supplementary Figure 30 (b). The sample demonstrated a strong preferred orientation, with the intense peaks in the pattern corresponding to the (001) diffraction plane of CrOCl. In summary, CrOCl nanoflakes were successfully synthesized.

It is thus believed that wafer-scale growth of CrOCl thin films, followed by either further transfer or growth steps of TMDs (and by repeatedly doing so) could lead to larger scale or higher number of stacked layers of VIP-FET arrays, which will be our future studies.

Supplementary Figure 27. Optical images of the obtained chromium oxide (Cr₂O₃).

Supplementary Figure 28. Schematic illustration of the CVD setup for growing CrOCl nanoflakes.

Supplementary Figure 29. Optical images of the obtained CrOCl nanoflakes. (a) Optical micrograph of the obtained CrOCl nanoflakes. (b) AFM height map of the selected area in (a). (c) Height profile along the red solid line in (b).

Supplementary Figure 30. Raman (a) and XRD (b) of the obtained CrOCl nanoflakes.

Supplementary Figure 31. Optical images of the obtained CrOCl nanoflakes with different coverage rates. It is believed that wafer-scale growth of CrOCl thin films, followed by either further transfer or growth steps of TMDs (and by repeatedly doing so) could lead to larger scale or higher number of stacked layers of VIP-FET arrays, which will be our future studies.”

Comment 2. *Regarding the DFT part. I think there is a significant room to strengthen up the calculations/simulations in this manuscript. By making a very simplified model of monolayer CrOCl and monolayer MoS₂, they claimed a band alignment so that electron charge transfer happens from MoS₂ to CrOCl, which eventually leads to a p-type MoS₂. I understand that it's not realistic to model a 30 nm heterostructure by DFT but at least a few layers are possible for computing. In addition, I don't understand why it gives a gap of only 0.5 eV in CrOCl, which is quite problematic and should be re-examined.*

Response:

We appreciate the great point by the referee very much.

We fully agree that the DFT calculations in the previous submission was way too simplified, which may not address the physical origin of the experimentally observed phenomena.

As suggested by the Referee, a few layers are used for computing in the new submission. And the gap of CrOCl has been re-examined. In the revised manuscript, we performed new calculations, using 10-layer MoS₂ and 5-layer CrOCl to determine the work function and band structures of each material, and constructed 3 MoS₂ layers + 3 CrOCl layers heterostructure with a super-cell of about 200 atoms, in order to consider the interfacial coupling between the MoS₂/CrOCl interface.

Our new DFT calculations together with electronic correlation corrections suggest that this interfacial coupling induced polarity inverse is a result of charge transfer from TMDs to CrOCl, followed by subtle *e-e* interactions in the surface state of CrOCl, which should be a universal effect at the interface between TMDs and layered insulators with high work function (WF) and large-enough effective mass in its surface band.

We have re-written the whole part of DFT calculation in the main text, in the left column in page 3, highlighted in blue. We also quote it as below:

“We first consider theoretically a model system of CrOCl coupled to TMD, using MoS₂ as an example. By calculating the charge density difference between the slightly electron-doped and neutral MoS₂ layer, as shown in Fig. 1b, the pristine state of MoS₂ (and most of the TMDs) exhibits n-type behavior, with the Fermi level close to the conduction band, illustrated in Fig. 1c. In order to determine their work functions, we performed DFT calculations of 10-layer MoS₂ and 5-layer CrOCl, and the conduction band minimum (CBM) of MoS₂ is estimated to be about 0.5 eV above the CBM of CrOCl. Therefore, if MoS₂ is initially electron doped, we would expect charge transfer from the CBM of MoS₂ to CrOCl. This is more explicitly elucidated by calculating the charge density difference between the electron-doped case and charge neutral case at the MoS₂-CrOCl interface, as shown in Fig. 1d. Clearly, the doped electron carriers are concentrated at the CrOCl side at the interface, leaving some holes on the MoS₂ side.

To further elucidate the experimental observations, we carried out DFT calculations of 3 MoS₂ layers + 3 CrOCl layers heterostructure with a super-cell of about 200 atoms (more details can be found in Methods). It is noticed that, since experimentally CrOCl still acts as a gate dielectric and hence no free carriers can be found in it, the electrons transferred (via tunneling from MoS₂ to the Cr-3*d* orbitals as suggested by DFT calculations) into the surface states of CrOCl have to be in a localized manner. Indeed, by considering the band structures of MoS₂, CrOCl, and MoS₂/CrOCl heterostructures, separately (Supplementary Figures 1-3), we found that the mechanism here in our system is more than a trivial charge transfer, but rather further followed by a combination of *e-e* interaction (which drives the charges in CrOCl surface state into insulator) and self-adjustment of band-alignments (see Supplementary Note 1). This is fundamentally different compared to conventional doping strategies for such TMDs semiconductors.

We have calculated the band structures of MoS₂/CrOCl heterostructure without electric field (Supplementary Figure 3a), as well as those of MoS₂ slabs and CrOCl slabs under opposite directions of electric fields (Supplementary Figure 3b). The detailed evolution behavior of the band edges of CrOCl and MoS₂ as a function of vertical electric field in these scenarios can be seen in Supplementary Table 1. When a negative electric field ~ 0.1 V/nm is applied (corresponding to the situation of negative bottom gate in our setup), the CBM of CrOCl is slightly pushed upwards in energy by 0.09 eV compared to the case without electric field, while the VBM of MoS₂ is dramatically increased in energy by ~ 0.5 eV such that it is only ~ 0.07 eV above the CBM of CrOCl. In the meanwhile, the electron carriers that are transferred to the surface CBM of CrOCl are expected to be frozen to form electronic crystal state driven by *e-e* interactions by virtue of the large effective mass and small carrier density, the chemical potential would be further lowered getting closer to the VBM of MoS₂ as illustrated in Fig. 1e, which thus can be easily tuned to be p-type upon further non-disruptive negative gate voltage. Therefore, it is the subtle interplay among the gate tunable band alignment, charge transfer, and *e-e* interactions that results in a chemical potential resided in the vicinity of MoS₂ VBM, and eventually leads to the effective gate tuning from n type to p-type carriers. Such a mechanism is likely to be universal and applicable to a wide range of 2D semiconducting materials [18].

Furthermore, we have included detailed additional computational results in the Supplementary Information (pages 3-9 in the Suppl. Info.), quoted below:

“Supplementary Note 1. Band structure of MoS₂/CrOCl heterostructures

In this Supplementary Note, we give details of the density functional theory (DFT) calculations as described in the main text in the section “Modelling of vdW interfacial coupling induced p-type doping in few-layered MoS₂”. Notice that Fig. 1d in the main text is a focused interface calculated from a 3-layered MoS₂ + 3-layered CrOCl heterostructure, and the holes located on the MoS₂ side of the interface are compensated by electrons deep in MoS₂ so that the entire MoS₂

multi-layer remains charge neutral, and almost all the doped electron carriers are transferred to CrOCl.

Calculation methods including the single particle and interaction pictures can be found in the Methods part in the main text. Here, we discuss some additional detailed results. In Supplementary Figure 1b and d, we show the band structures of MoS₂ and CrOCl, respectively. Their lattice structures are given in Supplementary Figure 1a and c. We consider two competing magnetic states of CrOCl: (I) the interlayer antiferromagnetic (AFM) and intralayer ferromagnetic (FM) state as shown in Supplementary Figure 1e, and interlayer and intralayer AFM state as shown in Supplementary Figure 1f. It turns out that the magnetic ground state of CrOCl in CrOCl-MoS₂ heterostructure is the type (II) state. We adopt type (II) magnetic order in all of our calculations.

In Supplementary Figure 2a-b we show the side view and top view of the lattice structure of MoS₂-CrOCl heterostructure. Supplementary Figure 3 a shows the electronic band structures of MoS₂-CrOCl heterostructure under zero electric field. The contributions from MoS₂ and CrOCl orbitals are marked by blue and red colors, respectively. We see that the MoS₂ bands have little hybridization with those from CrOCl. The conduction band minimum (CBM) of the heterostructure is contributed by MoS₂ conduction bands and the valence band maximum (VBM) is contributed by CrOCl valence bands.

We have calculated the band structures of the MoS₂/CrOCl heterostructure, from which the band alignment can be treated more accurately. We find that, in the absence of vertical electric fields, the heterostructure supercell calculation indicates that the CBM of CrOCl is 0.65 eV lower than the CBM of MoS₂, and is 0.44 eV above the valence band maximum (VBM) of MoS₂, as schematically shown in Supplementary Figure 4b. This is qualitatively consistent with those obtained from work-function calculations. Moreover, we find that the effective mass around CBM of CrOCl is $m^* = 2.88 m_e$ (m_e is the bare electron mass). If a small amount of electron carriers ($\sim 10^{12}$ cm⁻² as deduced from the applied gate voltage) are transferred to the surface of CrOCl, the corresponding Wigner-Seitz radius is as large as 122 (assuming a dielectric constant $\epsilon = 5$), far above the threshold value to form a Wigner crystal. This implies that the small amount of electron carriers transferred to CrOCl would not conduct, rather they tend to spontaneously crystallize driven by long-range e - e interactions and form an insulating electronic crystal.[1] This explains why the system remains insulating even under nominal electron doping.

We have also performed first principles DFT calculations for 10-layer slab of MoS₂ and 5-layer slab of CrOCl with electric fields (\mathbf{E}) ranging from - 0.2 V/nm to 0 V/nm (negative electric fields correspond to negative bottom gate voltages in our experimental configurations). The evolution of the CBM and VBM of CrOCl and MoS₂ with respect to vacuum level are given in Supplementary Table I. When \mathbf{E} is negative, the CBM of both CrOCl and MoS₂ increase with respect to the vacuum level, corresponding to the case of negative gate voltage in our experimental

set up. Despite of the increase of CrOCl CBM, the VBM of MoS₂ increases even faster under negative electric field. For example, when $E = -0.1$ V/nm, the CBM of CrOCl has been pushed up by 0.09 eV compared to that with zero electric field, while the VBM of MoS₂ is dramatically increased in energy by 0.5 eV (compared to the zero-field case), which is only 0.07 eV below CrOCl CBM, as shown in Supplementary Figure 3b (also see schematic illustration in Supplementary Figure 4c). As a result, if the system is initially electron doped, the electron carriers would be transferred to the CBM of CrOCl, and the chemical potential is only 0.07 eV above the VBM of MoS₂. Furthermore, as discussed above, the electron carriers transferred to the surface of CrOCl occupying its CBM are expected to be frozen to form an electronic crystal due to the large effective mass and small transferred carrier density, which would further pull down the chemical potential, thus can be easily hole doped upon further application of negative gate voltage.

The above calculations with finite electric fields are based slab models, which can capture the qualitative features of band alignment under vertical electric fields. A more comprehensive and accurate study requires direct first principles DFT calculations based on CrOCl/MoS₂ heterostructures under finite electric fields. More detailed results about this mechanism can be found in our forthcoming theory paper [2].

Supplementary Figure 1. Band structures and magnetic configurations of MoS₂ and CrOCl. Crystal structures of (a) MoS₂ and (c) CrOCl. The energy bands of (b) bulk MoS₂ and (d) bulk CrOCl. In order to take into account of the subtle *e-e* interactions, spin configurations of the CrOCl crystal have to be calculated. We here considered two magnetic configurations: (I) an interlayer antiferromagnetic and intralayer ferromagnetic state as shown in (e); and (II) an intralayer and interlayer antiferromagnetic state as shown in (f).

Supplementary Figure 2. Unit cell of the MoS₂-CrOCl heterostructure. (a) and (b) indicate the side and top views of the heterostructure of three layers of MoS₂ on top of three layers of CrOCl, as described in detail in the Methods section in the main text.

Supplementary Figure 3. Calculated band structure of MoS₂-CrOCl heterostructures. (a) show the electronic band structures of MoS₂-CrOCl heterostructure under zero electric field. The contributions from MoS₂ and CrOCl are marked by blue and red colours, respectively. (b) Calculated band edge of CBM for CrOCl and VBM for MoS₂, as a function of electrical fields. Here, negative electric fields correspond to negative bottom gate voltages in our experimental configurations.

Supplementary Table 1. Evolution of CBM and VBM of MoS₂ and CrOCl under different electric field E (in units of V/nm). Vacuum energy is set to zero. The band-edge energies are in units of eV.

E (V/nm)	-0.2	-0.15	-0.1	-0.05	0
MoS ₂ CBM	-4.193	-4.233	-4.248	-4.282	-4.34
CrOCl CBM	-4.651	-4.676	-4.718	-4.748	-4.805
MoS ₂ VBM	-4.228	-4.518	-4.781	-5.056	-5.293
CrOCl VBM	-6.187	-6.378	-6.567	-6.746	-6.926

Supplementary Figure 4. Schematic band alignments of MoS₂-CrOCl heterostructures. (a) shows the initial state of a n-type MoS₂. While in (b), when the few-layered MoS₂ is interfaced with CrOCl without any external electrical applied, the interfacial charge transfer takes place, leading to band re-alignment. Notice that, unlike a trivial charge transfer, here in our system *e-e* interaction has to be taken into account. And an insulator is formed in the surface state of CrOCl after the charge transfer, since the CrOCl surface state is not conducting. When a finite negative electrical field (positive bottom gate voltage in the experimental configuration) of $E = -0.1$ V/nm is applied in (c), the new Fermi level is pulled down to a much lower position as compared to that in (a), which is showing p-type semiconductor characteristics. Notice that, although the whole bands are shifting up in (c), the relative position of the MoS₂ VBM and CrOCl CBM are getting closer. Along with further increasing the amplitude of negative electrical field, the system will end up in two different band alignments, both of p-type as indicated in (d)-(e). (d) shows that the CBM of CrOCl is lower than the Fermi level, while (e) shows a CBM higher than the Fermi level of MoS₂, respectively. The difference between (d) and (e) depend on the detailed self-balance of total charges and nominal doping in each constituent layers, and both scenarios can be possible even though the general behavior of p-type MoS₂ is the same. The number index of 1 to 4 (or 4') are guides for the evolutions of the doping states and band-alignments in (a)-(e)."

Comment 3. *What other candidate materials can we consider to play the same role as that of CrOCl? It is important to elaborate or at least comment on this issue.*

Response:

We appreciate the great point by the referee very much.

Indeed, we have found other candidate materials other than CrOCl, which can also play the role of p-doping the TMDs. We have added the relative comments in the revised manuscript in Page 7, highlighted in blue, and also quoted below:

“We did find that other replacement such as few-layered $\text{Cr}_2\text{Ge}_2\text{Te}_6$ can effectively dope monolayered MoS_2 into a p-type FET, as shown in Supplementary Figure 22. In this case, even the on-state threshold voltage V_{th} can be tuned to different values compared to that in TMD/CrOCl devices.”

A new Supplementary Figure 22 is added in the Suppl. Info. in this new submission, also presented below. And, in the meantime, our theoretician collaborators are now working on high-throughput computation,^[1] in order to predict a library of such candidate materials in a systematic manner.

“**Supplementary Figure 22. Field effect curves of a typical $\text{MoS}_2/\text{Cr}_2\text{Ge}_2\text{Te}_6$ p-FET constructed through monolayer MoS_2 .** (a) Photo image of the $\text{MoS}_2/\text{Cr}_2\text{Ge}_2\text{Te}_6$ p-FET. (b) Structure illustration of the device. Field effect property of $\text{MoS}_2/\text{Cr}_2\text{Ge}_2\text{Te}_6$ (c) and MoS_2 (d).”

¹ Li M., and Liu J., Manuscript in preparation, (2023).

Comment 4. The authors claimed a hole mobility as high as $400 \text{ cm}^2/\text{V}\cdot\text{s}$, as far as I know, that is the highest value reported. How is this value calculated? How about the temperature dependence? Recently, it was reported that n-type MoS_2 can manifest an ON state current exceeding $100 \text{ }\mu\text{A}/\mu\text{m}$ under a bias voltage of a few hundred mV. I am interested to know the comparison of the performances of the p-type FETs in this work with the state-of-the-art results for n-type FETs based on MoS_2 .

Response:

We first answer the first question in this comment. “How is the hole mobility calculated”.

The hole mobility is calculated by using the standard model:

$$\mu = \frac{L}{W \cdot C_i \cdot V_{sd}} \cdot \frac{dI_{sd}}{dV_g}$$

$$\frac{1}{C_i} = \frac{1}{C_{\text{SiO}_2}} + \frac{1}{C_{\text{CrOCl}}}$$

$$C_{\text{SiO}_2} = (\epsilon_0 \cdot \epsilon_{\text{SiO}_2})/d_{\text{SiO}_2}$$

$$C_{\text{CrOCl}} = (\epsilon_0 \cdot \epsilon_{\text{CrOCl}})/d_{\text{CrOCl}}$$

where $L = 6 \text{ }\mu\text{m}$, $W = 1.5 \text{ }\mu\text{m}$, $\epsilon_{\text{SiO}_2} = 3.9$, $\epsilon_{\text{CrOCl}} = 3$, $d_{\text{SiO}_2} = 300 \text{ nm}$, $d_{\text{CrOCl}} = 8 \text{ nm}$, and $\frac{dI_{sd}}{dV_g} = 1.4 \times 10^{-7}$.

As to the temperature dependence, we measured the output curves of a typical $\text{MoS}_2/\text{CrOCl}$ p-FET, as shown in Figure R1. It is seen that, at low temperatures, the devices exhibit a contact barrier, and the low bias differential resistance is significantly higher compared to the values at room temperature.

Figure R1. Output characteristics of $\text{MoS}_2/\text{CrOCl}$ FETs measured at room temperature (a) and 1.5 K (b). Inset of (a) shows an optical micrograph of $\text{MoS}_2/\text{CrOCl}$ FETs.

Figure R2. (Cited as Supplementary Figure 25 in the updated Suppl. Info.) Transfer characteristics of a typical MoS₂/CrOCl p-FET at different temperatures measured at $V_{ds} = 1$ V (a) and $V_{ds} = 0.1$ V (c), with corresponding extracted field effect mobility (extracted at $V_{bg} = -10$ V) as a function of temperature shown in (b) and (d), respectively.

In the revised manuscript, we have added comments in the main text on the temperature dependence of the hole mobility in typical MoS₂/CrOCl devices (see Figure R2), in Page 7, highlighted in blue. We quote it below:

“Taking CrOCl-coupled few layer MoS₂ and MoSe₂ for example, we found a significantly suppression of the hole mobility upon cooling below 200 K (Supplementary Figure 25). It suggests that, although the I - V curves are rather linear (Fig. 4f) in the V_{ds} range of ± 0.1 V at room temperature, there is still a tiny contact barrier which is detrimental for low temperature performances”.

More comments are also added in Supplementary Note 2, to discuss the issue of temperature dependences of hole mobility.

Figure R3. (Cited as Supplementary Figure 24 in the updated Suppl. Info.) Transfer curves and Output characteristics of a typical MoS₂/CrOCl p-FET and a typical MoSe₂/CrOCl p-FET. In order to optimize the electrical properties of the TMD/CrOCl FETs, metal electrodes with typical thickness of Ti/Au ~ 5/50 nm were fabricated by EBL and e-beam evaporator, followed by an annealing under inert atmosphere (320 °C, 5 h, Ar : H₂ = 30 : 4 sccm) .

For the last question in this comment, indeed, as being pointed out by the referee, recent reports show that n-type MoS₂ can manifest an ON state current exceeding 100 μA/μm under a bias voltage of a few hundreds of mV. In order to make a comparison of the performances of the p-type FETs in this work with the state-of-the-art results for n-type FETs based on MoS₂, we have made fine tunings of the sample fabrication process, and found that a post annealing of the p-FET device

can yield much higher ON state current, with the maximum value reaching ~ 0.3 mA (a few tens of $\mu\text{A}/\mu\text{m}$) at V_{ds} of 2 V. The performance is still inferior to the best value found in n-type TMD FETs, but is rather high for p-type TMD FETs, as shown in Fig. R3. The relative comments are added in the Supplementary Note 2, which we cite below:

“In addition, we also found that, for the TMD/CrOCl FETs, metal electrodes with typical thickness of Ti/Au $\sim 5/50$ nm were fabricated by EBL and e-beam evaporator, followed by an annealing under inert atmosphere (320 °C, 5 hours, Ar : H₂ = 30 : 4 sccm) can significantly improve the electrical performances. For example, as shown in Supplementary Figure 24, both MoS₂/CrOCl and MoSe₂/CrOCl p-FETs can reach an ON-state current up to ~ 0.3 mA (a few tens of $\mu\text{A}/\mu\text{m}$) at V_{ds} of 2 V.”

Comment 5. *The fact that stand-alone devices can be stacked up to 14 layers might already be the limit. Can the authors comment on the possibility that such 3D interconnected circuits can be fabricated in a wafer-scale in the future?*

Response:

We thank the referee for pointing this out.

This comment is overlapping with her/his Comment 1. We would like to refer to our answers in that thread (response to Comment 1 of Referee#1), and we now will briefly describe this issue again here.

For the possibility of such 3D interconnected circuits fabricated in a wafer-scale in the future, we tend to believe that it surely faces a series of grand challenges, but is promising. As shown in our additional data in Supplementary Note 3, and Supplementary Figures 27-31 therein, we have successfully demonstrated large scale growth of CrOCl thin layers, which might be a starting point of our future work toward large area vertical heterostructures of TMD/CrOCl by CVD methods.

Moreover, we have given a cartoon illustration in the new submission, which depicts the vision of future skyscrapers of 3D interconnected circuits fabricated using the technique reported in this work, shown in Fig. R4 (cited as Figure 4).

Figure R4. (Cited as Figure 4h-n in the updated main text) An outlook of future 3D skyscrapers of 2D VIP-FETs, based on the technology described in this work.

Comment 5. *Is there any chance that the authors could demonstrate more complicated structures using the same technique of 3D integration of 2D semiconductors, such as SRAM and ring oscillators? Adding schematics and different 3D architectures to an outlook figure will make it more impactful.*

Response:

We thank the referee for pointing this out.

Yes, we have made our efforts in demonstrating more structures using the same technique of 3D integration of 2D semiconductors.

Typical vertical 4T-SRAM constructed using 2 sets of vertical CFETs with 6 vdW layers, and 1 single stack of vertical FETs with 14 vdW layers, have been demonstrated, as shown in Fig. R5 (cited as Supplementary Figure 26 in the updated Suppl. Info.).

Schematics and different 3D architectures are also added in the revised submission.

Figure R5. (Supplementary Figure 26 in the updated Suppl. Info.) 3D integrated 4T-SRAM. (a) Schematics of 4T-SRAM constructed using 2 sets of vertical CFETs with 6 vdW layers. (b) Schematics of 4T-SRAM constructed using 1 single stack of vertical FETs with 14 vdW layers. (c) and (d) are false-color SEM images of real device using the design in (a) and (b), respectively. (e) The input wave form of the SRAM. (f) and (g) are output wave form of the SRAM for typical devices shown in (c) and (d), respectively.

We would like to sincerely thank the very helpful comments given by Referee #1. The revisions according to her/his suggestions have made our manuscript of much improved quality. Her/his support in publication in Nature will be greatly appreciated.

Reply to referee #2

General Comment. *In the manuscript, "Toward Skyscrapers of 2D Semiconducting Complementary Logics" by Y. Guo et al., the authors have explored a new charge transfer p-doping technique for the fabrication of p-type 2D transistors, which ultimately yields vertically integrated NAND gates. The manuscript is suggestive of a new approach to designing vertically-stacked 2D complementary logic transistors. Despite the promising foundation of the study, the reviewer concludes that the manuscript lacks sufficient novelty and merit to justify publication in Nature. Detailed comments are provided below.*

Response:

We appreciate the comment of Referee #2.

We first would like to emphasize the innovative part in our manuscript.

It is known that, if there is one Achilles' heel of Si technology, it would be its inability to scale up in the z -dimension. This is because carrier creation in a Si MOS-FET is achieved through ion implantation onto the surface of a single-crystal Si wafer -- it has not been possible to grow/transfer another layer (or multiple) of single-crystal Si on top of the doped ones.

In this sense, van der Waals semiconductors demonstrate a strong capability of z -dimensional stacking, theoretically allowing for an unlimited number of layers, which makes them one of the most promising candidates for future 3D integration of CMOS logic. However, being studied for over two decades, 2D semiconductors are still at a status of 'being at the verge of, but not yet quite there' for a successful 3D vertical integration.

Our work for the first time solved the major challenge of controllable and stable p-doping that endows the easily switchable polarity of a van der Waals (vdW) semiconducting FET from n-type to p-type. This doping strategy, as will be discussed in more detail in the coming responses, is a result of charge transfer from TMDs to CrOCl, followed by subtle e - e interactions in the surface state of CrOCl, which is fundamentally different from conventional doping methods.

As stated above, we have no doubt of the novelty of our work, and the key merit of our study is that we devise a new route toward up-going scaling of CMOS logic, which shed lights for a robust and universal route in tailoring the carrier type in high mobility 2D semiconductors – a key toward skyscrapers of 2D complementary logic gates in their future 3D integrations.

We admit that there were some weaknesses of the performances of our p-FETs in the previous submission. Nevertheless, during the past months, we have significantly improved the electrical performances (more than 20 figures in the Suppl. Info. have been updated/added), and have performed systematic analysis/simulations in addition to the previous manuscript. These weaknesses have been corrected/strengthened up in the revision.

We will answer in a point-to-point manner to the questions raised by the Referee #2. We hope that the new revision, together with the additional analysis according to the comments, will be satisfactory.

Comment 1. *The presented charge transfer p-type doping method is not a novel contribution. Previous literature [1,2,3] has documented numerous such doping schemes that employ the interface of a 2D-TMD with a high work function material, achieving comparable, if not superior, doping concentrations. Certain p-2D-FET transistors crafted through such methods have exhibited higher performance metrics than the one presented in this manuscript.*

Response:

We thank the comments given by Referee #2 here.
However, we beg to differ.

The 3 literatures provided by the Referee#2 are definitely interesting, and is important for the community in the pursuit towards high performance p-FETs based on 2D-TMD materials.

However, the physical mechanism of the p-doping method in our system is more than a trivial charge transfer, but rather further followed by a combination of $e-e$ interaction (which drives the charges in CrOCl surface state into insulator) and self-adjustment of band-alignments (see Supplementary Note 1 in the revised submission).

Our doping method kept the device without notable hysteresis, and yield world-record hole mobility (over $400 \text{ cm}^2/\text{V/s}$) at room temperature.

And, furthermore, in this revision, we have made fine tunings of the sample fabrication process, and found that a post annealing of the p-FET device can yield much higher ON state current, with the maximum value reaching $\sim 0.3 \text{ mA}$ (a few tens of $\mu\text{A}/\mu\text{m}$) at V_{ds} of 2 V. The performance is still inferior to the best value found in n-type TMD FETs, but is rather high for p-type TMD FETs, as shown in Fig. R6. The relative comments are added in the Supplementary Note 2, which we cite below:

“In addition, we also found that, for the TMD/CrOCl FETs, metal electrodes with typical thickness of Ti/Au \sim 5/50 nm were fabricated by EBL and e-beam evaporator, followed by an annealing under inert atmosphere (320 °C, 5 hours, Ar : H₂ = 30 : 4 sccm) can significantly improve the electrical performances. For example, as shown in Supplementary Figure 24, both MoS₂/CrOCl and MoSe₂/CrOCl p-FETs can reach an ON-state current up to \sim 0.3 mA (\sim 60 μ A/ μ m) at V_{ds} of 2 V.”

Figure R6. (Cited as Supplementary Figure 24 in the updated Suppl. Info.) Transfer curves and Output characteristics of a typical MoS₂/CrOCl p-FET and a typical MoSe₂/CrOCl p-FET. In order to optimize the electrical properties of the TMD/CrOCl FETs, metal electrodes with typical thickness of Ti/Au \sim 5/50 nm were fabricated by EBL and e-beam evaporator, followed by an annealing under inert atmosphere (320 °C, 5 h, Ar : H₂ = 30 : 4 sccm) .

Figure R7. (a) Field effect curves of p-type MoSe₂ (red curves, with a statistics of 12 devices) and p-type MoS₂ (green curves, with a statistics of 10 devices) induced by coupling of the CrOCl substrates, respectively. $V_{ds} = 0.1$ V is used in the measurement. (b) The device fabricated for transfer length method (TLM) measurement. (c) Transfer curves of typical device in (b), measured for different channel length.

As shown in Fig. R7, statistics of both p-type MoSe₂ and p-type MoS₂, and TLM test for MoSe₂ suggest much improved electrical performance of the p-FETs device fabricated in our work. The relative data are updated in the revised Fig. 4e-g in the main text.

In the revised manuscript, we have also cited positively the 3 papers (indexed as Refs. 38-40 in our main text) mentioned by the Referee#2, quoted below:

“The interfacial-coupling induced p-doping and the therefore enabled VIP-FETs are key techniques invented in this study, and are conceptually suitable to the up-going scaling of future 2D semiconducting CMOS circuits. To visualize the envisioned picture, we compare side-by-side the 4T-SRAM devices based on the planar-FET, CFET, and VIP-FET architectures, shown in the SEM images in Fig. 4h-j, with their cartoon illustrations in Fig. 4k-m, respectively. Clearly, the footprint sizes of them are sequentially decreasing from 4 unit areas to 1 unit area, with a z-dimensional accumulation from 3 vdW to 14 vdW layers, respectively. We thus can expect future

skyscrapers of 2D VIP-FETs in Fig. 4n. And this makes the most distinct feature of our technique, although there are very interesting alternative ways of p-doping TMDs [38–40]”.

As a summarization, we believe that our doping strategy provide new insights and added-value in addition to the 3 papers recommended by the referee. Both routes are interesting, but the key point of our method is that it not only gives rise to high-performance p-FETs, but also allows further 3D vertical integration of 2D CMOS logic.

We hope the Referee#2 can be convinced this way, and her/his support in publishing our work is very much appreciated.

Comment 2. *Moreover, the p-type doping scheme explored in this study is not selective, thereby doping the entire source-channel-drain region. This is disadvantageous to electrostatic channel control as doping strength escalates. It is speculated that the weak electrostatic gate control at reduced channel thicknesses, with more effective doping, could be why the manuscript does not report the p-2D-FET performance with thinner layers (<4 channel layers). It is crucial to note that the primary advantage of 2D-TMDs for FET design is their superb gate electrostatic control facilitated by their atomic body thickness. Hence, this process must be displayed for relevant thinner body thicknesses, where 2D-FETs surpass Si (and/or Ge) counterparts.*

Response:

We appreciate that the Referee#2 pointed out for the discussion on the issue of selective doping.

The terminology “selective doping” in semiconductors refers to the intentional introduction of specific impurities or dopants into localized regions of a semiconductor material while leaving other areas undoped.

In fact, our technique is naturally a selective doping method, as is clearly shown in Fig. 2a in our main text, when half of the MoS₂ flake is interfacially-coupled to CrOCl, the semiconducting channel is selectively doped into p-type on the left half ONLY, leaving the right half undoped.

We now answer to the comment “*It is speculated that the weak electrostatic gate control at reduced channel thicknesses, with more effective doping, could be why the manuscript does not report the p-2D-FET performance with thinner layers (<4 channel layers). It is crucial to note that the primary advantage of 2D-TMDs for FET design is their superb gate electrostatic control facilitated by their atomic body thickness*”.

In fact, we found that **monolayered** TMD can be p-doped using our doping-scheme. For example, few-layered $\text{Cr}_2\text{Ge}_2\text{Te}_6$ can effectively dope monolayered MoS_2 into a p-type FET, as shown in Supplementary Figure 22, also presented below in Fig. R8.

Therefore, the above concern of Referee#2 is addressed.

Figure R8. (Cited as Supplementary Figure 22). Field effect curves of a typical $\text{MoS}_2/\text{Cr}_2\text{Ge}_2\text{Te}_6$ p-FET constructed through monolayer MoS_2 . (a) Photo image of the $\text{MoS}_2/\text{Cr}_2\text{Ge}_2\text{Te}_6$ p-FET. (b) Structure illustration of the device. Field effect property of $\text{MoS}_2/\text{Cr}_2\text{Ge}_2\text{Te}_6$ (c) and MoS_2 (d).

Second, indeed as being pointed out by Referee#2, we did observe that when thickness of MoS_2 is lower than a certain value, the p-dope does not take effects. This might be due to the fact that thinner TMDs have larger band gaps which do not fulfill the required Fermi level down-shift as calculated by our DFT results, as illustrated in Table R1, and Fig. R9. We also have to emphasize that the doping-mechanism in this work is a cooperative effect of gate tunable band alignment, charge transfer, and $e-e$ interactions, which is essentially different from conventional p-doping strategies for semiconducting TMDs. It is worthwhile noting that similar mechanism already leads

to robust high-temperature, low-field quantum Hall effect as well as correlated insulator states in CrOCl/graphene heterostructures.^{2,3}

Here in this work, the thinnest device which holds the validity of p-doped behavior is shown in Fig. R10 (cited as Supplementary Figure 21 in the updated Suppl. Info.), with the thickness t of the MoS₂ flake being about 3.2 nm.

Table R1 (Supplementary Table 1. in the revised Suppl. Info.) Evolution of CBM and VBM of MoS₂ and CrOCl under different electric field E (in units of V/nm). Vacuum energy is set to zero. The band-edge energies are in units of eV.

E (V/nm)	-0.2	-0.15	-0.1	-0.05	0
MoS ₂ CBM	-4.193	-4.233	-4.248	-4.282	-4.34
CrOCl CBM	-4.651	-4.676	-4.718	-4.748	-4.805
MoS ₂ VBM	-4.228	-4.518	-4.781	-5.056	-5.293
CrOCl VBM	-6.187	-6.378	-6.567	-6.746	-6.926

² Wang, Y. et al. Quantum Hall phase in graphene engineered by interfacial charge coupling. Nature Nanotechnology 17, 1272–1279 (2022).

³ Lu, X. et al. Synergistic correlated states and nontrivial topology in coupled graphene-insulator heterostructures. Nature Communications 14, 5550 (2023).

Figure R9. (Supplementary Figure 4 in the revised Suppl. Info.) Schematic band alignments of MoS₂-CrOCl heterostructures. (a) shows the initial state of a n-type MoS₂. While in (b), when the few-layered MoS₂ is interfaced with CrOCl without any external electrical applied, the interfacial charge transfer takes place, leading to band re-alignment. Notice that, unlike a trivial charge transfer, here in our system *e-e* interaction has to be taken into account. And an insulator is formed in the surface state of CrOCl after the charge transfer, since the CrOCl surface state is not conducting. When a finite negative electrical field (positive bottom gate voltage in the experimental configuration) of $E = -0.1$ V/nm is applied in (c), the new Fermi level is pulled down to a much lower position as compared to that in (a), which is showing p-type semiconductor characteristics. Notice that, although the whole bands are shifting up in (c), the relative position of the MoS₂ VBM and CrOCl CBM are getting closer. Along with further increasing the amplitude of negative electrical field, the system will end up in two different band alignments, both of p-type as indicated in (d)-(e). (d) shows that the CBM of CrOCl is lower than the Fermi level, while (e) shows a CBM higher than the Fermi level of MoS₂, respectively. The difference between (d) and (e) depend on the detailed self-balance of total charges and nominal doping in each constituent layers, and both scenarios can be possible even though the general behavior of p-type MoS₂ is the same. The number index of 1 to 4 (or 4') are guides for the evolutions of the doping states and band-alignments in (a)-(e).”

Figure R10. (Cited as Supplementary Figure 21). Field effect curve of a MoSe₂/CrOCl p-FET with $t_{\text{MoSe}_2} \sim 3.2$ nm measured at $V_{ds} = 0.1$ V. (a) Optical photo of the tested device. (b) AFM height profile of the MoSe₂ layer in the device. (c) Field effect curve of the device measured at $V_{ds} = 0.1$ V.

In addition, we found that the ON-state threshold voltage V_{th} can be tuned into different values by selection of gate dielectric materials (HfO_2 , SiO_2 , etc.) (shown in Fig. R11), as well as the selection of different interfacial-coupling layer ($CrOCl$, $Cr_2Ge_2Te_6$ (shown in Fig. R8), and etc.), making the future logic applications using these p-FETs devised in this work more applicable.

Figure R11. (Cited as Supplementary Figure 23 in the revised Suppl. Info) Field effect curves of $MoS_2/CrOCl$ p-FETs with Al_2O_3 (a) and HfO_2 (b) served as dielectric layer, respectively.

A paragraph in page 27 in the revised Suppl. Info. has also been added:

“Interestingly, when we substitute the h-BN dielectric with Al_2O_3 or HfO_2 for these TMD/ $CrOCl$ FETs, the general p-type doping behavior do not change, but their V_{th} are effectively tuned toward the positive direction of gate voltages (Supplementary Figure 23), yielding the so-called depletion mode, instead of enhancement mode p-FET in most of the cases such as that shown in Fig. 2a the main text. The selection of gate dielectric materials, as well as the selection of different interfacial-coupling layer ($CrOCl$, $Cr_2Ge_2Te_6$, and etc.) make the selective doping possible in future logic applications using these p-FETs devised in this work”.

As a summarization, we hope our new results of a monolayered MoS₂ being p-doped will convince Referee#2 that our doping scheme is still working for thinner body thicknesses, down to the one atomic layer limit. And in the meantime, we have proposed theoretical explanations for the reason why MoS₂/CrOCl did not work for p-doping when thinner than 4 layers.

With all the added analysis in the main text (Page 7) and Suppl. Info. (Page 27-33), we believe we have fully addressed the concerns raised by Referee#2 in this comment. Her/his support of publications will be greatly appreciated.

Comment 3. *In addition to the claimed novel p-type doping scheme, the paper illustrates the creation of stacked complementary transistor logic. However, this demonstration does not address the most significant challenges in vertical (3D) integration of 2D transistor logic. Primarily, the transistors developed in this paper are transfer-based, rather than directly grown, which is critical for a functioning 3D integration process flow. Furthermore, n- and p-2D-FETs lack a consistent gate topology (either top/bottom-gated; while top-gate preferred) crucial for equivalent, matched transistor performance.*

Response:

We appreciate the concern raised by Referee#2.

Indeed, our demo devices are still transfer-based. Directly growth will be the holy-grail, but out of the scope of our manuscript so far. It will definitely be our future investigations.

Nevertheless, we already started considering chemical vapor deposition (CVD) growth of large scale CrOCl, as a starting point. We hope in near future, as inspired by Referee#2, we will be able to submit new independent manuscript, reporting the results of wafer-scale p-FETs based on the doping scheme reported in our current fundamental-research paper.

We have added in Supplementary Note 3 some of the CVD progress of our future work, quoted as below:

“Supplementary Note 3. Possibility of large-scale production of CrOCl thin films

One of the challenges for future applications based on the VIP-FETs described in this work is the scaling up of these devices. Apparently, large scale production of CrOCl thin layers can be a starting point. In this supplementary note, we discuss the possibility of chemical vapor deposition (CVD) growth of CrOCl thin films.

2D CrOCl nanoflakes were synthesized by an ambient pressure CVD method. The growth was conducted in a 1-m length, 1-inch outer diameter quartz tube heated by a one-zone furnace (MTI, OTF-1200X-III-C). The reaction principle for synthesizing CrOCl involves the hydrolysis of chromium trichloride. The specific reaction equation is as follows: $\text{CrCl}_3 + \text{H}_2\text{O} \rightarrow \text{CrOCl} + 2\text{HCl}$.

According to this reaction principle, we have devised the following experimental procedure: using chromium trichloride powder as the source of chromium and utilizing air as the source of water. In the initial stages, we employ compressed air as the source of air. Different quantities of CrCl₃ powder (Alfa, 99.99%) was used as the solid precursor which was placed under a stacked mica substrate in a quartz boat. Prior to the growth process, the furnace was purged by the flowing of 300 sccm high-purity Ar gas for 5 min. Then, 50-200 sccm Ar and 1-10 sccm Air was introduced into the CVD system. The furnace was heated to 600-800 °C and growth time was set at 10 min. However, regardless of temperature and gas flow rate adjustments, the resulting outcome remains chromium oxide nanoflakes, shown in Supplementary Figure 27. The results indicate that the water content flow still exceeds our expectations.

In order to reduce the water content to a level conducive for the growth of CrOCl during the experimental process, compressed air was no longer utilized. Instead, the air inlet was slightly opened to allow a minimal amount of atmospheric air to enter the reaction system. Furthermore, to achieve thinner nanoflake thickness, a spatial confinement approach for growth was employed as depicted in Supplementary Figure 28.

The specific experimental procedure was as follows: 50 mg of CrCl₃ powder was positioned before a stacked mica substrate in a quartz boat. Prior to the growth process, the furnace was purged by flowing 300 sccm of high-purity Ar gas for 5 minutes. Subsequently, 200 sccm of Ar was introduced into the CVD system with an incomplete seal of the quartz tube to permit a small amount of air to enter. The furnace was then heated to 780°C for 78 minutes, with the growth time set at 10 minutes. Once the heating process was concluded, the furnace was allowed to naturally cool down. The CrOCl nanoflakes would be obtained between the two mica pieces. An optical image of the synthesized CrOCl nanoflakes is depicted in Supplementary Figure 29.

The nanoflakes exhibited a rhombic morphology. To characterize the phase of the nanoflakes, Raman and XRD analyses were conducted. As shown in Supplementary Figure 30 (a), the Raman peaks at 209.5 cm⁻¹, 418.5 cm⁻¹, and 460.5 cm⁻¹ corresponds to the A_g mode of CrOCl. The XRD patterns are shown in Supplementary Figure 30 (b). The sample demonstrated a strong preferred orientation, with the intense peaks in the pattern corresponding to the (001) diffraction plane of CrOCl. In summary, CrOCl nanoflakes were successfully synthesized.

It is thus believed that wafer-scale growth of CrOCl thin films, followed by either further transfer or growth steps of TMDs (and by repeatedly doing so) could lead to larger scale or higher number of stacked layers of VIP-FET arrays, which will be our future studies.

Supplementary Figure 27. Optical images of the obtained chromium oxide (Cr₂O₃).

Supplementary Figure 28. Schematic illustration of the CVD setup for growing CrOCl nanoflakes.

Supplementary Figure 29. Optical images of the obtained CrOCl nanoflakes.

Supplementary Figure 30. Raman and XRD of the obtained CrOCl nanoflakes.

Supplementary Figure 31. Optical images of the obtained CrOCl nanoflakes with different coverage rates. It is believed that wafer-scale growth of CrOCl thin films, followed by either further transfer or growth steps of TMDs (and by repeatedly doing so) could lead to larger scale or higher number of stacked layers of VIP-FET arrays, which will be our future studies.”

For the comment “*Furthermore, n- and p-2D-FETs lack a consistent gate topology (either top/bottom-gated; while top-gate preferred) crucial for equivalent, matched transistor performance*”. I think we failed in delivering some key messages in our previous submission. We feel sorry to have made confusions to the Referee #2.

In fact, gate topology is rather a ‘free-design’ in our devices. For convenience, we used vertical 6 vdW layer CFT with p- and n-channel sharing a common gate. But this can be totally rearranged, there is no problem in this matter.

Comment 4. *Although the FET mobility and ON/OFF ratio are commendable, the ON-current is significantly low (~1-5 μ A) for practical applications. This deficiency results from poor contacts, a large Schottky barrier height, and a lack of expansive doping schemes.*

Response:

We thank the comments given by the referee.

This comment is overlapping with Comment 1 given by Referee#2. We would like to emphasize again that the performances of our p-FETs have been much improved, and the maximum ON-current can now reach up to 0.3 mA.

As already describe in our response to Comment 1, we would like to rephrase it here: in this revision, we have made fine tunings of the sample fabrication process, and found that a post annealing of the p-FET device can yield much higher ON state current, with the maximum value reaching ~0.3 mA (a few tens of μ A/ μ m) at V_{ds} of 2 V. The performance is still inferior to the best value found in n-type TMD FETs, but is rather high for p-type TMD FETs, as shown in Fig. R6.

Additionally, in order to systematically characterize the contact barrier in the newly obtained results, we further carried out temperature dependence measurements. And it is seen that, while the device at high bias ($V_{ds} = 1$ V) show metallic behavior with increasing hole mobility upon cooling (Supplementary Figure 25a-b), its hole mobility at low bias ($V_{ds} = 0.1$ V) significantly decreases at temperatures below 200 K (Supplementary Figure 25c-d). It suggests that, although the I - V curves are rather linear (Fig. 4f) in the V_{ds} range of ± 0.1 V at room temperature, there is still a tiny contact barrier which is detrimental for low temperature performances, shown in Fig. R12.

Figure R12. (Cited as Supplementary Figure 25 in the updated Suppl. Info.) Transfer characteristics of a typical MoS₂/CrOCl p-FET at different temperatures measured at $V_{ds} = 1$ V (a)

and $V_{ds} = 0.1$ V (c), with corresponding extracted field effect mobility (extracted at $V_{bg} = -10$ V) as a function of temperature shown in (b) and (d), respectively.

In the revised manuscript, we have added comments in the main text on the temperature dependence of the hole mobility in typical MoS₂/CrOCl devices, in Page 7, highlighted in blue. We quote it below:

“Taking CrOCl-coupled few layer MoS₂ and MoSe₂ for example, we found a significantly suppression of the hole mobility upon cooling below 200 K (Supplementary Figure 25). It suggests that, although the I - V curves are rather linear (Fig. 4f) in the V_{ds} range of ± 0.1 V at room temperature, there is still a tiny contact barrier which is detrimental for low temperature performances”.

At this stage, we believe that the concerns of the Referee#2 is no longer present, since the performance of p-FETs have been improved in the revision. We hope the additional analysis on temperature dependence could also make our results convincing to Referee#2.

Comment 5. *Lastly, the demonstrated NAND gate operates with input voltages ranging from -4V to 4V, but generates an output voltage of only 0V to 5V. This discrepancy impedes circuit design as one stage cannot power the subsequent stage.*

Response:

We thank the comments given by Referee #2.

This mismatch problem has been corrected in the revised submission, as explained in detail in Fig. R13 (Supplementary Figure 20), cited below:

Figure R13. (Cited as Supplementary Figure 20 in the updated Suppl. Info.) Performances of NAND logic based on MoS₂/CrOCl VIP-FETs. (a) and (b) are input and output waveforms of the NAND logic with 14 vdW layers under $V_{DD} = 0$, and $V_{DD} = -3$ V, respectively. Depending on the matching of V_{th} between n- and p-FETs, the output waveform may be un-matched (a) or matched (b) with the input waveform. It is preferred to have the operation waves as the one illustrated in (b), in order to power the subsequent stages in circuit designs. It is also noticed that, although we are mainly focused on the CrOCl interfaced TMDs in this work, other layered insulators (or much lower conductivity compared to TMDs at room temperature) such as Cr₂Ge₂Te₆ (shown in Supplementary Figure 22), or even TMD/CrOCl heterostructures with different gate dielectrics (Supplementary Figure 24) will give rise to rather different range of V_{th} . It says the fact that one can indeed engineer the selective doping, to some extent, of the p-type semiconducting channel using the strategy devised in this work. As also reported in the literature,[10] selective p-doping (i.e., continuously adjustable V_{th}) by tuning the number of trimethylaluminium soaking cycles could be our future studies.

Figure R14. (Cited as Figure 4 in the updated main text) Universality and versatility of TMD-based VIP-FETs.

At the end, we would like to add a few more words on the improved part of our manuscript:

- In addition to the NAND presented in the previous submission, we have also demonstrated other logic gates such as 14 vdW layer 4T-SRAM,
- and have systematically discussed the universality and versatility of the TMD/CrOCl VIP-FETs with improved electrical performances.

- CVD growth of CrOCl, parasitic capacitance of the vertically integrated complementary logic, as well as the new DFT calculation with $e-e$ interaction taken into account, are added into the revision.

All the above efforts (a significant part of them was suggested/inspired by Referee #2, which we thank her/him a lot) have made our manuscript much more solid in terms of scientific innovation and technical leap forward. A new Fig.4 can be found in the main text, which is Fig. R14 shown in this letter.

In conclusion, we sincerely appreciate the constructive comments given by Referee #2 during the review process, which has significantly improved the quality of our manuscript.

Her/his support in publication in Nature will be greatly appreciated.

Reply to referee #3

General Comment. *Vertical 3D integration of 2D semiconductors is a promising technology that could overcome the limitations of silicon-based transistors. The authors have made a significant contribution to the field of 2D semiconductor integration by demonstrating a robust and universal method for controlling the carrier polarity of 2D semiconductors. The results are impressive, with transistors achieving room-temperature hole mobilities up to $\sim 425 \text{ cm}^2 \text{ V}^{-1} \text{ s}^{-1}$, and on/off ratios exceeding 106. The devices are also air-stable for over six months, which is a significant advantage. The authors have also demonstrated the feasibility of vertically constructing inverters and NANDs with 6 and 14 vdW layers, respectively. These findings are important for the future of 3D integrated circuits, and I believe that this work will be of interest to a wide audience. However, there are still some important points that need to be clarified in order to be published in Nature. Below are some comments.*

Response:

We appreciate very much the positive comments by Referee#3, and we feel happy that she/he found our work “I believe this work can be published in high impact journals”.

In the following, we will answer in a point-to-point manner to the remaining questions raised by the Referee #3, and we hope that the new revision will be satisfactory.

Comment 1. *Metal electrodes are formed in multiple layers to create a three-dimensional device structure. The thickness of the h-BN used as the insulator layer is 27.3 nm, but the metal electrodes are formed vertically, creating unwanted parasitic capacitors, and the capacitance will be high. A clear explanation and supporting data are required for this part.*

Response:

We thank the Referee#3 for this valuable comment.

Indeed, the interlayer arranged electrodes will create unwanted parasitic capacitors. To address this issue, we performed detailed TCAD simulations which can serve as supporting data, and we added a whole new discussion part to the manuscript, which is cited as Supplementary Note 4 in the revised Suppl. Info.:

“Supplementary Note 4. TCAD Simulations of parasitic capacitances of the VIP-FETs devices

In this Supplementary Note, we investigate the impact of parasitic effects on the electrical performances of the vertically integrated multi-vdW-layer VIP-FETs devices. For simplicity, complementary field effect transistor (CFET inverter) devices with 6 vdW layers (similar to the device shown in Fig. 2c in the main text) were simulated, in order to estimate the influence of parasitic capacitance at different frequencies on the output characteristics of such an inverter using the TCAD tool Sentaurus.

Before the simulation, calibration of the model was necessary. To simplify the simulation, we assumed that both NMOS and PMOS were doped through body defects, with donors in NMOS and acceptors in PMOS. Degradation of mobility due to vertical electric fields were not taken into account, and a constant mobility model was used instead. Physical effects including the quantum potential model, non-local Schottky barrier tunneling in source and drain, and high-field velocity saturation, were considered in our analysis. As shown in Supplementary Figure 32, the simulated transfer curves closely match the experimental results for both PMOS and NMOS FETs, validating the applicability of our models.

Next, we used the parasitic capacitance extraction tool, *i.e.*, the TCAD tool Raphael, to extract parasitic capacitance between metal lines in the inverter. Generally, two scenarios can be possible in device fabrications: 1) metal lines for different channels are designed in a parallel manner; 2) metal lines for different channels are designed in an orthogonally-intersected manner. When metal lines intersect, parasitic capacitance is introduced only at the intersecting nodes between the two metal lines in upper and lower layers. Due to the characteristics of metal line distribution, this capacitance cannot be treated as a simple parallel plate capacitor. Therefore, specialized parasitic capacitance analysis tools like TCAD tool Raphael are needed. In this work, we analyzed the parasitic capacitance introduced in both parallel and intersecting configurations of metal lines, as shown in Supplementary Figure 33. In the case of orthogonally intersected metal lines (Supplementary Figure 33a), four sets of parasitic capacitances are introduced. However, only the capacitance between V_{dd} - V_{out} and V_{ss} - V_{out} affects the inverter's characteristics, while the capacitance between V_{dd} - V_{ss} and V_{out} - V_{out} has no impact on the output characteristics. For the parallelly arranged metal lines (Supplementary Figure 33b), we focused on the configuration where V_{out} and V_{dd} are overlapping through different layers.

According to the results extracted by Raphael, the two sets of capacitance values were almost identical. In the case of intersecting configuration, $C_1 = C_2 = 5.5465 \times 10^{-16}$ farads, while in parallel configuration, $C_1 = C_2 = 1.4 \times 10^{-15}$ farads. It's clear that the parallel configuration introduces larger parasitic capacitance compared to the intersecting ones.

To further analyze the impact of parasitic capacitance on the inverter characteristics, we performed device-circuit co-simulation using the TCAD tool Sentaurus. NMOS, PMOS, and parasitic capacitance were connected in a circuit netlist. We simulated the input-output characteristics of the inverter at different device sizes, metal overlap configurations, and frequencies. It can be seen that for long-channel devices (similar to the parameters in the experimentally tested devices in the

main text), the inverter operates normally below a frequency of 1 MHz (Supplementary Figure 34), with or without considering parasitic capacitance, showing identical V_{out} -Time relationships. However, above 1 MHz, the inverter gradually fails, exhibiting noticeable overshooting effects, which are generally caused by the presence of a large gate-source-drain ($C_{\text{g-s-d}}$) capacitance in the circuit, leading to an output voltage exceeding the logic high level, or falling below the logic low level. At a frequency of 1 GHz (Supplementary Figure 34), the inverter completely fails, and the overshooting-effect dominates. At this point, the presence (absence) of the parasitic effects results in a slow charging (discharging) of the inverter output voltage, and upon further increasing the frequency to, for example, 10 GHz, the impact of parasitic capacitance becomes much more pronounced. Notably, the parallel configuration of metal electrode lines exhibits larger impact of parasitic capacitance, as shown in Supplementary Figure 34.

It is seen that, for the long-channel (a few micron-meters) devices, parasitic capacitance may start to take effects at a frequency of around 1 GHz, affecting the inverter's charging and discharging characteristics. However, due to the large $C_{\text{g-s-d}}$ capacitance, significant overshooting already occurs around 1 MHz, and the impact of parasitic capacitance is not negligible at this low frequency regime.

To further validate the role of parasitic capacitance, we simulated the characteristics of small-sized (around 100 nm lateral size) devices, as shown in Supplementary Figure 35.

Due to the smaller $C_{\text{g-s-d}}$ capacitance in small-sized devices, it is observed that the functionality of the inverter prevails up to 10 MHz, an order of magnitude higher than that of the long-channeled devices. At around 100 MHz, impact of parasitic capacitance becomes significant, slowing down the transition rate of the inverter's high and low levels. Furthermore, parallelly arranged inter-layer electrodes manifest larger parasitic capacitance, resulting in a slower level transition rate as compared to the devices with orthogonally intersected inter-layer electrodes. We do observe failure of the inverters with small-size at 10 GHz, as shown in Supplementary Figure 35.

In summary, the impact of parasitic capacitance is expected to occur between the frequency range of 100 MHz and 1 GHz, leading to a slower transition rate between high and low levels of the simulated 6-vdW-layer VIP-FET inverters. Additionally, inter-layer electrodes configured in a parallel manner (Supplementary Figure 34c & 35c) yields larger parasitic capacitances, as compared to the orthogonally intersected ones (Supplementary Figure 34a & 35a). However, due to the large $C_{\text{g-s-d}}$ capacitance (capacitance from the device itself, not from the parasitic ones from inter-layer arranged electrodes) in large-sized devices, the inverter starts to fail around 1 MHz, making the impact of parasitic capacitance less noticeable. In contrast, for small-sized devices with smaller $C_{\text{g-s-d}}$, failure occurs around 1 GHz, allowing for a more pronounced observation of the impact of parasitic capacitance. These simulated results will be a guidance for future nano-electronic circuitries made from the reported VIP-FETs technique in this work.

Supplementary Figure 32. Calibration of the model for TCAD simulations. Left: simulated field effect curve (red solid line), according to the experimental data (green dashed line) for the p-FET. Right: simulated field effect curve (red solid line), according to the experimental data (green dashed line) for the n-FET. Insets are the log-scale plot of the corresponding data.

Supplementary Figure 33. Different interlayer arrangements of the electrodes for CFET inverter. The devices are made of 6 vdW layers in a vertical stack, similar to the device described in Fig. 2c in the main text. (a) and (b) are false-colored SEM images for vertical inverter with 6 vdW layers, with their interlayer electrodes constructed in an orthogonally intersected, and a parallel manner, respectively. (c) The schematic drawing of the circuit with two parasitic capacitances labeled as $C1$ and $C2$. T_P and T_N denote p-type and n-type transistors.

Supplementary Figure 34. TCAD simulations of parasitic capacitances for long channel device. Here 6 vdW layered vertical CFET inverter is considered. The lateral sizes are adopted from experimental device described in Fig. 2c in the main text. (a) and (b) are the cartoon illustration of the vertical CFET inverter with the interlayer electrodes arranged in an intersected manner, and the simulated waveforms at different frequencies from 1 kHz up to 10 GHz with/without the influence of parasitic capacitance, respectively. (c) and (d) are the cartoon illustration of the vertical CFET inverter with the interlayer electrodes parallelly constructed, and the simulated output waveforms at different frequencies with/without the influence of parasitic capacitance, respectively.

Supplementary Figure 35. TCAD simulations of parasitic capacitances for short channel device. Here 6 vdW layered vertical CFET inverter is considered. Channel length is set to be about 100 nm. (a) and (b) are the cartoon illustration of the vertical CFET inverter with the interlayer electrodes arranged in an intersected manner, and the simulated waveforms at different frequencies from 1 kHz up to 10 GHz with/without the influence of parasitic capacitance, respectively. (c) and (d) are the cartoon illustration of the vertical CFET inverter with the interlayer electrodes parallelly constructed, and the simulated output waveforms at different frequencies with/without the influence of parasitic capacitance, respectively.

”.

Comment 2. *The Raman spectra of MoS₂, MoS₂/CrOCl, and CrOCl are shown in Supplementary Figure 7. In the Raman spectrum of MoS₂/CrOCl, the author only describes the peak shift of the A_{1g} mode compared to MoS₂, but it appears that the E_{2g} mode is also shifted in the same direction and by a similar distance. The authors need to clarify this phenomenon.*

Response:

We thank the comment by Referee#3 for her/his valuable suggestions and very careful readings of our manuscript.

In the revised version, we have added the discussion of Supplementary Figure 10 on the shift of E_{2g} mode of MoS₂, as well. This phenomenon is now clarified, we quote it below:

“

Supplementary Figure 10. Raman spectra of typical MoS₂/CrOCl samples. (a) Raman spectra for three regions of a typical MoS₂/CrOCl sample (No. S-21-10a): few-layer MoS₂, few-layer CrOCl, and few-layered MoS₂/CrOCl overlapping region of the sample. (b) Same data plotted in another sample (No. S-23-11b). As shown in (a), it is seen that the two individual layers exhibit consistent Raman characteristic peaks as compared to the previous reported results[4,5], namely, an in-plane active mode E¹_{2g} at 379 cm⁻¹ and an out-of-plane mode A¹_g at 404 cm⁻¹ for the few-layer MoS₂, and A_g² mode at 411 cm⁻¹ and A_g³ mode at 452 cm⁻¹ for the few-layer CrOCl, respectively. Interestingly, compared with few-layer MoS₂, both the frequencies of the E¹_{2g} and the A¹_g modes of MoS₂ in the MoS₂/CrOCl overlapping region are slightly blue-shifted from 381 to 382.5 cm⁻¹ in (c), and from 404 to 405 cm⁻¹ in (e), respectively. In (c) and (e), the dotted circles are experimental data, while the solid lines are Lorentzian fit of the measured peaks. Green curves/points correspond to MoS₂ only, while yellow ones correspond to MoS₂ interfaced with CrOCl. It is reported that the blue shift of the A¹_g mode can be attributed to a p-doping effect of the few-layered MoS₂. [6,7] We further made a statistic of 13 MoS₂/CrOCl samples whose field effect curves are confirmed to p-type. While most of them showed blue-shifts in both the E¹_{2g} and the A¹_g modes, and we noticed that, in some rare cases, these heterostructure samples with especially thin CrOCl showed absence of blue-shifts, as indicated in (d) and (f).”

Comment 3. *The purpose and concept of this research are excellent. However, unless the area of the two-dimensional semiconductor is fundamentally large, it is difficult to replace the Si semiconductor. Authors should suggest ways to overcome these limitations.*

Response:

We appreciate the positive comments on the novelty of our work by Referee#3.

And we thank the referee a lot for raising the concern of large area fabrication. This comment is in echo with the Comment 1 from Referee#1, as well. Here, we will repeat some of our responses.

We fully agree with Referee #3 that the 3D vertical integrated 2D semiconducting device demonstrated in this work cannot compete with Si, unless they can be scaled up to wafer-scale. This is an important issue that should be discussed in our work.

In the revised manuscript, we added a paragraph in the section of **Outlook of the VIP-FETs**, to emphasize this grand challenge, quoted below:

“These challenges include the heat dissipation, large- area growth of p-type film (MoS₂/CrOCl heterostructure for example) with high uniformity suitable for industrial production, as well as the possible interlayer parasitic capacitance. We here briefly discuss the possibility of chemical vapor

deposition (CVD) growth of large scale CrOCl thin films in Supplementary Note 3 (Supplementary Figures 27-31), and we also performed Technology Computer-Aided Design (TCAD) simulations of the parasitic capacitance due to the interlayer overlapping of electrodes, shown in Supplementary Note 4 (Supplementary Figures 32-35)".

And, we also added additional data in Supplementary Note 3, and Supplementary Figures 27-31 therein, to show the attempt of large scale growth of CrOCl thin layers as a starting point of our future work toward large area TMD/CrOCl by CVD methods. We quote it below:

"Supplementary Note 3. Possibility of large-scale production of CrOCl thin films

One of the challenges for future applications based on the VIP-FETs described in this work is the scaling up of these devices. Apparently, large scale production of CrOCl thin layers can be a starting point. In this supplementary note, we discuss the possibility of chemical vapor deposition (CVD) growth of CrOCl thin films.

2D CrOCl nanoflakes were synthesized by an ambient pressure CVD method. The growth was conducted in a 1-m length, 1-inch outer diameter quartz tube heated by a one-zone furnace (MTI, OTF-1200X-III-C). The reaction principle for synthesizing CrOCl involves the hydrolysis of chromium trichloride. The specific reaction equation is as follows: $\text{CrCl}_3 + \text{H}_2\text{O} \rightarrow \text{CrOCl} + 2\text{HCl}$.

According to this reaction principle, we have devised the following experimental procedure: using chromium trichloride powder as the source of chromium and utilizing air as the source of water. In the initial stages, we employ compressed air as the source of air. Different quantities of CrCl_3 powder (Alfa, 99.99%) was used as the solid precursor which was placed under a stacked mica substrate in a quartz boat. Prior to the growth process, the furnace was purged by the flowing of 300 sccm high-purity Ar gas for 5 min. Then, 50-200 sccm Ar and 1-10 sccm Air was introduced into the CVD system. The furnace was heated to 600-800 °C and growth time was set at 10 min. However, regardless of temperature and gas flow rate adjustments, the resulting outcome remains chromium oxide nanoflakes, shown in Supplementary Figure 27. The results indicate that the water content flow still exceeds our expectations.

In order to reduce the water content to a level conducive for the growth of CrOCl during the experimental process, compressed air was no longer utilized. Instead, the air inlet was slightly opened to allow a minimal amount of atmospheric air to enter the reaction system. Furthermore, to achieve thinner nanoflake thickness, a spatial confinement approach for growth was employed as depicted in Supplementary Figure 28.

The specific experimental procedure was as follows: 50 mg of CrCl_3 powder was positioned before a stacked mica substrate in a quartz boat. Prior to the growth process, the furnace was purged by flowing 300 sccm of high-purity Ar gas for 5 minutes. Subsequently, 200 sccm of Ar was

introduced into the CVD system with an incomplete seal of the quartz tube to permit a small amount of air to enter. The furnace was then heated to 780°C for 78 minutes, with the growth time set at 10 minutes. Once the heating process was concluded, the furnace was allowed to naturally cool down. The CrOCl nanoflakes would be obtained between the two mica pieces. An optical image of the synthesized CrOCl nanoflakes is depicted in Supplementary Figure 29.

The nanoflakes exhibited a rhombic morphology. To characterize the phase of the nanoflakes, Raman and XRD analyses were conducted. As shown in Supplementary Figure 30 (a), the Raman peaks at 209.5 cm^{-1} , 418.5 cm^{-1} , and 460.5 cm^{-1} corresponds to the A_g mode of CrOCl. The XRD patterns are shown in Supplementary Figure 30 (b). The sample demonstrated a strong preferred orientation, with the intense peaks in the pattern corresponding to the (001) diffraction plane of CrOCl. In summary, CrOCl nanoflakes were successfully synthesized.

It is thus believed that wafer-scale growth of CrOCl thin films, followed by either further transfer or growth steps of TMDs (and by repeatedly doing so) could lead to larger scale or higher number of stacked layers of VIP-FET arrays, which will be our future studies.

Supplementary Figure 27. Optical images of the obtained chromium oxide (Cr_2O_3).

Supplementary Figure 28. Schematic illustration of the CVD setup for growing CrOCl nanoflakes.

Supplementary Figure 29. Optical images of the obtained CrOCl nanoflakes. (a) Optical micrograph of the obtained CrOCl nanoflakes. (b) AFM height map of the selected area in (a). (c) Height profile along the red solid line in (b).

Supplementary Figure 30. Raman (a) and XRD (b) of the obtained CrOCl nanoflakes.

Supplementary Figure 31. Optical images of the obtained CrOCl nanoflakes with different coverage rates. It is believed that wafer-scale growth of CrOCl thin films, followed by either further transfer or growth steps of TMDs (and by repeatedly doing so) could lead to larger scale or higher number of stacked layers of VIP-FET arrays, which will be our future studies.”

At the end of this Rebuttal Letter, we would like to sincerely appreciate the very constructive comments given by Referee #3. Thanks to her/his insightful comments/suggestions, our revised manuscript has been significantly improved.

Her/his support in publication in Nature will be greatly appreciated.

Reviewer Reports on the First Revision:

Referees' comments:

Referee #1 (Remarks to the Author):

I have had a close look at the authors' revisions. All my comments were well addressed, except for one that I am still not fully convinced about: more information should be provided for the "hero device" that has a high on/off ratio reaching 10^6 and a hole mobility of $\sim 425 \text{ cm}^2/\text{Vs}$. In the reply letter, the authors claim " $L = 6 \text{ nm}$, $W = 1.5 \text{ nm}$, $d\text{SiO}_2 = 300 \text{ nm}$, $d\text{CrOCl} = 8 \text{ nm}$ ", but it looks different from all the devices shown in the main text. Moreover, under a high standard, having only one "hero device" may not be enough.

P.S., Although the CFET structure is becoming an ultimate strategy for Moore's law, in my opinion, realizing monolithic planar 2D CMOS might be the first step for the 2D community, because the transfer technique is still immature. But the doping strategy offered by this paper could still be a key step towards real applications of 2D electronics.

After the authors have sufficiently addressed my criticisms, I am happy to recommend that the revised manuscript be published in Nature.

Referee #2 (Remarks to the Author):

The presented charge transfer p-type doping method is trivial, though more DFT simulation results were added, it was not much different from other transfer charge doping schemes (i.e., MoOx cap or chemical molecules (AuCl₃ or NO_x doping), which provide higher performance doping by NO_x (i.e., $I_{\text{on}} > 700 \text{ } \mu\text{A}/\mu\text{m}$, see for example, Hao-Yu Lan et al., "Wafer-scale CVD monolayer WSe₂ p-FETs with record-high $727 \mu\text{A}/\mu\text{m}$ I_{on} and $490 \text{ } \mu\text{s}/\mu\text{m}$ G_{max} via hybrid charge transfer and molecular doping," IEDM 2023).

The p-type doping scheme is not selective naturally, since it needs to be placed intentionally and this is extremely difficult for sub-50 nm channel length devices where it is only needed to dope the source and drain, and it is not compatible with the self-aligned process, which is very important for nanoscale devices for any practical implementations.

In addition, I have some questions about the DFT simulations and device performance:

1. In the Supplementary Note 1, if the holes located on the MoS₂ side are compensated by electrons deep in the MoS₂ as claimed, how did the hole doping happen? ...since MoS₂ transferred electrons to the CrOCl.

2. Why is the magnetic ground state of CrOCl the type II state?

3. What was the mechanism of the annealing process (5 hrs 320 °C) making the device performance better?

4. Since the TLM measurements were performed, why was there no result of extracted contact resistance? Contact resistance is an important parameter that reflects the doping concentration.

5. This paper claimed a very high hole mobility (over 400 cm²/Vs). The method used to extract it should be clarified.

6. Since in an ultra-scaled region, the gate pitch or contact pitch will be < 100 nm. In such dimensions, will the CrOCl still be an effective doping functional layer? and why?

7. One of the major advantages of 2D TMDs' arise from their ultra-thin channel thickness, which can be thinner than 1 nm. However, the authors claimed that the proposed doping method only worked for channel thickness thicker than 3.2 nm, which restricts its capability to deliver the desirable performance of 2D TMDs. So, what is the practical utility of the proposed method?

8. It can be observed from Figure R8 that the monolayer TMD device exhibited a worse performance, i.e., $I_{on} < 1 \mu A$ under $V_{ds} = 1V$. This suggests that some kind of scattering hindered the electron transport, which could be induced by the doping layer. Can the authors confirm that this degradation is due to something else?

9. The DFT simulation results of monolayer MoS₂ doping with CrOCl should be provided to illustrate why p-doping does not happen in monolayer MoS₂.

Referee #3 (Remarks to the Author):

The authors have responded appropriately to the reviewer's comments and the manuscript has been well revised. The manuscript can be published in the present form.

Author Rebuttals to First Revision:

Reply to referee #1:

Comment 1. I have had a close look at the authors' revisions. All my comments were well addressed, except for one that I am still not fully convinced about: more information should be provided for the "hero device" that has a high on/off ratio reaching 10^6 and a hole mobility of $\sim 425 \text{ cm}^2/\text{Vs}$. In the reply letter, the authors claim " $L = 6 \text{ nm}$, $W = 1.5 \text{ nm}$, $d\text{SiO}_2 = 300 \text{ nm}$, $d\text{CrOCl} = 8 \text{ nm}$ ", but it looks different from all the devices shown in the main text. Moreover, under a high standard, having only one "hero device" may not be enough.

P.S., Although the CFET structure is becoming an ultimate strategy for Moore's law, in my opinion, realizing monolithic planar 2D CMOS might be the first step for the 2D community, because the transfer technique is still immature. But the doping strategy offered by this paper could still be a key step towards real applications of 2D electronics.

After the authors have sufficiently addressed my criticisms, I am happy to recommend that the revised manuscript be published in Nature.

Response:

We are very grateful for Reviewer #1 for her/his positive comments. Her/his comments are very helpful in terms of improving our manuscript.

We fully agree that the previous presentation of the best performance of a "hero device" is to be strengthened by including more systematic data from multiple devices.

We have listed 6 different samples that are used for the estimation of hole mobility. We believe these data can address his/her concerns in the following, with related modified parts highlighted in blue color in the main text and/or Supplementary Information in the revised version of our manuscript.

Basically, the hole mobility is estimated from 2 types of devices,

Type-I device:

Sample without local bottom gate, but a global doped-Si gate through 300 nm SiO₂. That is, a configuration of MoS₂/CrOCl/SiO₂/Si⁺⁺. And the maximum hole mobility of this type of device can be obtained by fitting the maximum slope of the field effect curve, using the standard model:

$$\mu = \frac{L}{W \cdot C_i \cdot V_{sd}} \cdot \frac{dI_{sd}}{dV_g}$$

$$\frac{1}{C_i} = \frac{1}{C_{SiO_2}} + \frac{1}{C_{CrOCl}}$$

$$C_{SiO_2} = (\epsilon_0 \cdot \epsilon_{SiO_2})/d_{SiO_2}$$

$$C_{CrOCl} = (\epsilon_0 \cdot \epsilon_{CrOCl})/d_{CrOCl}$$

Where C_{SiO_2} is the gate capacitance of the 300 nm SiO_2 , and C_{CrOCl} is the capacitance induced by the dielectric $CrOCl$ layer. Taking Sample 221228_S2.1 for example, In this case, $L = 6 \mu m$, $W = 1.5 \mu m$, $\epsilon_{SiO_2} = 3.9$, $\epsilon_{CrOCl} = 3$, $d_{SiO_2} = 300 \text{ nm}$, $d_{CrOCl} = 15 \text{ nm}$, $V_{sd} = 0.1 \text{ V}$, $\frac{dI_{sd}}{dV_g} = 8.8 \times 10^{-8}$. Thus, μ is estimated to be $325 \text{ cm}^2 \text{ V}^{-1} \text{ s}^{-1}$.

Type-II device:

We also have typical devices equipped with local bottom gate. That is, a configuration of $MoS_2/CrOCl/h-BN/Au$. And the maximum hole mobility of this type of device can be obtained by fitting the maximum slope of the field effect curve, using the model:

$$\mu = \frac{L}{W \cdot C_i \cdot V_{sd}} \cdot \frac{dI_{sd}}{dV_g}$$

$$\frac{1}{C_i} = \frac{1}{C_{hBN}} + \frac{1}{C_{CrOCl}}$$

$$C_{hBN} = (\epsilon_0 \cdot \epsilon_{hBN})/d_{hBN}$$

$$C_{CrOCl} = (\epsilon_0 \cdot \epsilon_{CrOCl})/d_{CrOCl}$$

Where C_{hBN} is the gate capacitance of the h-BN, and C_{CrOCl} is the capacitance induced by the dielectric $CrOCl$ layer. Taking Sample 230711_S3a for example, in this case, $L = 2 \mu m$, $W = 5 \mu m$, $\epsilon_{BN} = 4$, $\epsilon_{CrOCl} = 3$, $d_{BN} = 45 \text{ nm}$, $d_{CrOCl} = 10 \text{ nm}$, $V_{sd} = 0.1 \text{ V}$, maximum slop in the field effect curve $\frac{dI_{sd}}{dV_g} = 3.48 \times 10^{-6}$. Thus $\mu = 229 \text{ cm}^2 \text{ V}^{-1} \text{ s}^{-1}$.

All the details of samples are listed in Figure R1 (cited as Supplementary Figure 19) and Table R1 (cited as Supplementary Table 2).

Figure R1. Field effect curves of typical samples that are used for the estimation of hole mobilities. (a)–(f) Six typical samples with either type-I or type-II configurations in the gate dielectrics, as detailed in Table R1. All data are measured at room temperature.

Table R1. Details of the information of samples for the estimation of hole mobilities.

Type-I device: MoS₂/CrOCl/SiO₂/Si⁺⁺ p-FET						
	d_{SiO_2} (nm)	d_{CrOCl} (nm)	L (μm)	W (μm)	$\frac{dI_{sd}}{dV_g}$ (A/V)	μ_h ($\text{cm}^2 \text{V}^{-1} \text{s}^{-1}$)
221228-S3	300	8	6	1.5	1.18×10^{-7}	425
221228_S2a	300	15	6	1.5	8.8×10^{-8}	325
221228_S5	300	30	6	1.5	5.35×10^{-8}	210
221228_S2b	300	30	6	1.5	7.05×10^{-8}	266
Type-II device: MoS₂/CrOCl/h-BN/Au p-FET						
	d_{BN} (nm)	d_{CrOCl} (nm)	L (μm)	W (μm)	$\frac{dI_{sd}}{dV_g}$ (A/V)	μ_h ($\text{cm}^2 \text{V}^{-1} \text{s}^{-1}$)
230711_S3a	45	10	2	5	3.48×10^{-6}	229
230711_S2	35	10	2	5	2.57×10^{-6}	140

As a summarization, an updated Supplementary Figure 19 is now added in the revised manuscript, and the Fig. 2h in the main text is also updated with multiple devices.

We appreciate very much the support of publication by Referee#1. Her/his comments/suggestions have significantly improved our manuscript.

Reply to referee #2

General Comment 1. *The presented charge transfer P-type doping method is trivial, though more DFT simulation results were added, it was not much different from other transfer charge doping schemes (i.e., MoOx cap or chemical molecules (AuCl3 or NOx doping), which provide higher performance doping by NOx (i.e., $I_{on} > 700 \mu A/\mu m$, see for example, Hao-Yu Lan et al., "Wafer-scale CVD monolayer WSe2 p-FETs with record-high 727 $\mu A/\mu m$ I_{on} and 490 $\mu s/\mu m$ G_{max} via hybrid charge transfer and molecular doping," IEDM 2023).*

Response:

While we appreciate very much the comment of Referee #2, we beg to differ from his/her opinion that “*The presented charge transfer P-type doping method is trivial, though more DFT simulation results were added, it was not much different from other transfer charge doping schemes (i.e., MoOx cap or chemical molecules (AuCl3 or NOx doping)*”.

First, we agree that the I_{on} of P-type WSe₂ in the above-mentioned IEDM paper is exceptionally good. We cited this paper as Ref. 41 in the revision, in order to make the background of our manuscript more up-to-date. However, we still have to emphasize that it is NOT the main scope of our manuscript.

In our work, the major finding is a **novel scaling route of 3D vertically integrating complementary 2D semiconductors** (14 vdW layers in the vertical-SRAM and NAND logics, for example). Which we found that the Referee#2 has been overlooking/misunderstanding.

In short, there are 2 main totally different routes for the development of 2D semiconductors, as shown in Fig. R2:

- Conventionally, many authors take Route 1 (Fig. R2a), to enhance the performance of TMD transistors;
- We are adopting Route 2 (Fig. R2b), to scale vertically, and to demonstrate 3D integration of 2D complementary logic.

Our $e-e$ interaction assisted P-doping scheme is therefore strictly suitable for 3D vertical integration of 2D complementary logics, for its stable, clean, and 3D-stackable nature. And our work is the 1st demonstration of 3D CMOS based on polarity-reversible 2D TMD-FET by vdW intercalation.

Improving/enhancing the metrics (such as I_{on}) of conventional in-plane P-type CVD WSe₂ transistors is preferable, but not really the top priority in this regard.

a Route 1, conventional path which aims at further shrinking in-plane size with improved performance

- Monolayer TMD
- Shorter Channel Length
- Higher I_{ON}
- Higher ON/OFF ratio
- ⋮

b Route 2, new path scaling up in “Z”-dimension, with vertical 3D integration of 2D complementary logics

Figure R2. Two very different routes toward future developments of 2D vdW semiconductors. (a) Route 1, as promoted by Referee#2, is aiming at better performance of 2D FET (p- or n-type), with smaller footprints. (b) Route 2, an alternative way to go “upwards” in order to vertically integrate 2D complementary logics, as is taken by many recent studies including our manuscript.

Second, we would like to emphasize that our method is not trivial. There might be some misunderstanding by Referee#2, as in her/his case of WSe_2 , it was a doping of a bipolar semiconductor, as shown in Fig. R3a. The polarity of the field effect curve was NOT completely reversed, but shifted to some extent, which is the case of conventional charge doping.

However, in our case, as shown in Fig. R3b, the polarity of the MoS_2 FET is completely reversed from N-type into P-type. Here, a Fermi level crossing from conduction band to valence band has to happen. And it requires $e-e$ interactions (Fig. R3c) in the CrOCl (detailed in Supplementary Note 1), a totally new effect that cannot be explained by trivial charge transfer.

Figure R3. Conventional doping vs non-trivial e - e interaction assisted doping schemes. (a) Trivial doping effect via charge transfer from molecular or oxides, which cause a shift of fermi level. Taking WSe₂ as an example, the system starts from an initial bipolar state, and becomes more P-doped after charge transfer. (b) Schematics of N- to P-type doping, taking MoS₂/CrOCl system as an example. (c) Schematics of the e - e interaction assisted polarity change process, *i.e.*, the nontrivial doping scheme. Numbers of the gap sizes in (c) are obtained by DFT simulations.

General Comment 2. *The P-type doping scheme is not selective naturally, since it needs to be placed intentionally and this is extremely difficult for sub-50 nm channel length devices where it is only needed to dope the source and drain, and it is not compatible with the self-aligned process, which is very important for nanoscale devices for any practical implementations.*

Response:

Although we appreciate very much the comment of Referee #2, we found him/her controversial in the past 2 rounds of review.

- At the very beginning, Referee#2 said there is NO selectivity in P-doping in our work,
- Now he/she admits that there indeed IS selectivity, but not compatible with the sub-50 nm channel.

The selectivity depends solely on whether if there is CrOCl/TMD interface. And the sub-50 nm pitch is a lithography process, not a vdW transfer issue.

In the revised manuscript, we have added a comment in the main text in Page 7 (highlighted in blue), in order to address the concern about compatibility with the sub-50 nm channel:

“Notice that to obtain ultra-scaled sub-50 nm channel lengths, it may need higher precision lithography tool. In the scenario of very small samples, manual alignment will also pose limitations. Larger-sized films would be preferable for potential practical applications”.

Comment 1. *In addition, I have some questions about the DFT simulations and device performance: 1. In the Supplementary Note 1, if the holes located on the MoS₂ side are compensated by electrons deep in the MoS₂ as claimed, how did the hole doping happen? ...since MoS₂ transferred electrons to the CrOCl.*

Response:

We thank the comments given by Referee #2 here.

Supplementary Note 1 deals mainly with Fig. 1d in the main text, and the calculations in Fig. 1d is to simulate a scenario of how the charge distributes at the interface of MoS₂(10 layers)/CrOCl(5 layer), when additional electron charges ($\sim 10^{11} \text{ cm}^{-2}$) are injected into the system. Despite the slight electron doping, no electric field has been applied to the MoS₂/CrOCl heterostructure in our DFT calculations, such that the chemical potential still stays in the gap of MoS₂. As a result of the charge transfer, our calculations indicate that the doped electron carriers all stay at the surface of

CrOCl, and MoS₂ remains charge neutral. This is why we say that the holes at the surface of MoS₂ close to CrOCl is compensated by electrons in the opposite surface.

MoS₂ becomes P-type doped only if an electric field pointing from MoS₂ to CrOCl is applied, which is illustrated by the evolution of MoS₂ valence band maxima (VBM) and CrOCl conduction band minima (CBM) as a function of electric field (Supplementary Figure 3b). Specifically, in the presence of negative gate voltage, the CBM of CrOCl would be lowered and the VBM of MoS₂ would be increased such that at some critical negative electric field ~ 0.15 V/nm, they start to overlap with each other so that the chemical potential would be located in the valence bands of MoS₂ which become hole doped. See Supplementary Figure 4 for schematic illustration of such a scenario.

In response to this question, we have added the following sentence in Supplementary Note 1:

“Here, to test the interfacial charge-transfer effects, additional electron charges ($\sim 10^{11}$ cm⁻²) are injected into the system under zero electric field. MoS₂ still remains charge neutral since no electric field has been applied, and the chemical potential remains in the gap of MoS₂.”

Comment 2. *Why is the magnetic ground state of CrOCl the type II state?*

Response:

Sorry that we did not provide more information about the magnetic ground state of CrOCl in our previous submission.

Actually, we have considered extensively the magnetic ground states of CrOCl in our previous DFT calculations in the graphene/CrOCl systems [Nature Nanotechnology **17**, 1272 (2022); and Nature Communications **14**, 5550 (2023)], in which the *e-e* interaction picture has successfully captured most of the experimental observations in the interfacial coupling between graphene and CrOCl. Here, we would like to quote the DFT simulations for the magnetic ground states of CrOCl from our previous work,^[R1] as follows:

^{R1} Supplementary Information in Wang, Y. *et al.* “Quantum Hall phase in graphene engineered by interfacial charge coupling”. *Nature Nanotechnology* **17**, 1272-1279 (2022).

[REDACTED]

“To be specific, as shown in Fig. R4, we have calculated the energies of three magnetic configurations: (I) an inter-layer AFM and intralayer FM state; (II) an intralayer AFM and interlayer FM state; and (III) a $1 \times 4 \times 1$ intralayer AFM and interlayer FM state that enlarges the primitive cell by four times along the in-plane b -axis. Previous first principles calculations show that the type (III) intralayer AFM state is the bulk ground state with the Hubbard $U = 3.2$ eV calculated from constrained random phase approximation.^[R2]”

As discussed above, there are actually multiple competing magnetic states in CrOCl, which has been proven experimentally as well^[3]. Theoretically, the calculated magnetic ground state depends on the value of Hubbard U , and also depends on the dielectric environment. Experimentally, applying a small magnetic field may induce multiple magnetic transitions in CrOCl. All the theoretical results discussed above apply to free-standing CrOCl. Our DFT+ U calculations (with $U = 3$ eV) indicate that the magnetic ground state of trilayer CrOCl becomes the type-II state after forming an interface with MoS₂. Nevertheless, the Mott insulating-like band structures, and the positions of the band edges are almost the same for the different magnetic states. This is because the charge degrees of freedom in the CrOCl Mott insulator is determined by U (\sim a few eV), while the magnetic degrees of freedom are determined by magnetic exchange J (\sim a few meV).

It is also worthwhile noting that the magnetic coupling between CrOCl and MoS₂ does not play any significant role for the physics discussed in this work. First, the gate tunable P-type behavior of MoS₂ is robust even under room temperature, which is far beyond the Neel temperature of CrOCl (~ 14 K). Second, even at temperatures below 14 K, the interfacial magnetic exchange coupling between CrOCl and MoS₂ is expected to decay exponentially with the interlayer distance, such that the spin splitting in MoS₂ due to the magnetic proximity would be exponentially small as confirmed by DFT calculations.

^{R2} S. W. Jang, D. H. Kiem, J. Lee, Y.-G. Kang, H. Yoon, and M. J. Han, “Hund's physics and the magnetic ground state of CrOX (X= Cl, Br)”, *Phys. Rev. Materials* **5**, 034409 (2021).

^{R3} P. Gu, et al., “Magnetic phase transitions and magnetoelastic coupling in a two-dimensional stripy antiferromagnet”, *Nano Letters*, **22**, 1233, (2022).

Comment 3. *What was the mechanism of the annealing process (5 hrs 320 °C) making the device performance better?*

Response:

We appreciate the concern raised by Referee#2.

The annealing process is specially working for Ti/Au electrode. We believe it is due the contact issue.

In the case of Cr/Au electrodes, annealing is NOT working, and it even degrades the performance of the resulted devices. We have listed the annealed Ti/Au electrodes, as well as non-annealed Cr/Au electrodes for typical MoS₂/CrOCl and MoSe₂/CrOCl devices studied, as summarized in Table R2 (also cited as Supplementary Table 3 in the revised Suppl. Info.):

Table R2. Improvement of electrical performances by annealing procedure. On state current I_{ON} at $V_{ds} = 0.1$ V are measured at room temperature, in samples typically of lateral size of 2-3 μm in length and 3-5 μm in width. Annealing process are described in the figure caption of Supplementary Figure 26.

		Cr/Au contacts		Ti/Au contacts	
		Without Anneal	Annealed	Without Anneal	Annealed
MoS ₂ /CrOCl	Carrier type	P	N/A	P	P
	I_{ON} at $V_{ds} = 0.1$ V	$\sim 1 \mu\text{A}$	N/A	\sim a few nA	\sim a few μA
MoSe ₂ /CrOCl	Carrier type	P	N/A	P	P
	I_{ON} at $V_{ds} = 0.1$ V	~ 300 nA	N/A	\sim a few nA	$\sim 10 \mu\text{A}$

Comment 4. *Since the TLM measurements were performed, why was there no result of extracted contact resistance? Contact resistance is an important parameter that reflects the doping concentration.*

Response:

We thank the comments given by the referee. Indeed, we can obtain contact resistances from the TLM samples, as following:

Figure R5. Contact resistance for MoSe₂/CrOCl devices. (a) Transfer curves for different channel lengths. (b) Total resistance for different channel lengths extracted at fixed hole doping density. Contact resistance is estimated to be $8.8 k\Omega \cdot \mu m$.

The above data are included in the updated revision of Supplementary Information note 2, highlighted in blue color.

Comment 5. *This paper claimed a very high hole mobility (over $400 cm^2/Vs$). The method used to extract it should be clarified.*

Response:

We thank the comments given by Referee #2. Since this comment is overlapping with the comment by Referee#1 in this round of review, we will repeat the responses below:

“We have listed 6 different samples that are used for the estimation of hole mobility. We believe these data can address his/her concerns in the following, with related modified parts highlighted in blue color in the main text and/or Supplementary Information in the revised version of our manuscript.

Basically, the hole mobility is estimated from 2 types of devices,

Type-I device:

Sample without local bottom gate, but a global doped-Si gate through 300 nm SiO₂. That is, a configuration of MoS₂/CrOCl/SiO₂/Si⁺⁺. And the maximum hole mobility of this type of device can be obtained by fitting the maximum slope of the field effect curve, using the standard model:

$$\mu = \frac{L}{W \cdot C_i \cdot V_{sd}} \cdot \frac{dI_{sd}}{dV_g}$$

$$\frac{1}{C_i} = \frac{1}{C_{SiO_2}} + \frac{1}{C_{CrOCl}}$$

$$C_{SiO_2} = (\varepsilon_0 \cdot \varepsilon_{SiO_2})/d_{SiO_2}$$

$$C_{CrOCl} = (\varepsilon_0 \cdot \varepsilon_{CrOCl})/d_{CrOCl}$$

Where C_{SiO_2} is the gate capacitance of the 300 nm SiO₂, and C_{CrOCl} is the capacitance induced by the dielectric CrOCl layer. Taking Sample 221228_S2.1 for example, In this case, $L = 6 \mu\text{m}$, $W = 1.5 \mu\text{m}$, $\varepsilon_{SiO_2} = 3.9$, $\varepsilon_{CrOCl} = 3$, $d_{SiO_2} = 300 \text{ nm}$, $d_{CrOCl} = 15 \text{ nm}$, $V_{sd} = 0.1 \text{ V}$, $\frac{dI_{sd}}{dV_g} = 8.8 \times 10^{-8}$. Thus μ is estimated to be $325 \text{ cm}^2 \text{ V}^{-1} \text{ s}^{-1}$.

Type-II device:

We also have typical devices equipped with local bottom gate. That is, a configuration of MoS₂/CrOCl/h-BN/Au. And the maximum hole mobility of this type of device can be obtained by fitting the maximum slope of the field effect curve, using the model:

$$\mu = \frac{L}{W \cdot C_i \cdot V_{sd}} \cdot \frac{dI_{sd}}{dV_g}$$

$$\frac{1}{C_i} = \frac{1}{C_{hBN}} + \frac{1}{C_{CrOCl}}$$

$$C_{hBN} = (\varepsilon_0 \cdot \varepsilon_{hBN})/d_{hBN}$$

$$C_{CrOCl} = (\varepsilon_0 \cdot \varepsilon_{CrOCl})/d_{CrOCl}$$

Where C_{hBN} is the gate capacitance of the h-BN, and C_{CrOCl} is the capacitance induced by the dielectric CrOCl layer. Taking Sample 230711_S3a for example, in this case, $L = 2 \mu\text{m}$, $W = 5 \mu\text{m}$, $\epsilon_{BN} = 4$, $\epsilon_{CrOCl} = 3$, $d_{BN} = 45 \text{ nm}$, $d_{CrOCl} = 10 \text{ nm}$, $V_{sd} = 0.1 \text{ V}$, maximum slope in the field effect curve $\frac{dI_{sd}}{dV_g} = 3.48 \times 10^{-6}$. Thus $\mu = 229 \text{ cm}^2 \text{ V}^{-1} \text{ s}^{-1}$.

All the details of samples are listed in Figure R6 (cited as Supplementary Figure 19) and Table R3 (cited as Supplementary Table 2).

Figure R6. Field effect curves of typical samples that are used for the estimation of hole mobilities. (a)–(f) Six typical samples with either type-I or type-II configurations in the gate dielectrics, as detailed in Table R1. All data are measured at room temperature.

Table R3. Details of the information of samples for the estimation of hole mobilities.

Type-I device: MoS₂/CrOCl/SiO₂/Si⁺⁺ p-FET						
	d_{SiO_2} (nm)	d_{CrOCl} (nm)	L (μm)	W (μm)	$\frac{dI_{sd}}{dV_g}$ (A/V)	μ_h ($\text{cm}^2 \text{V}^{-1} \text{s}^{-1}$)
221228-S3	300	8	6	1.5	1.3×10^{-7}	425
221228_S2a	300	15	6	1.5	8.8×10^{-8}	325
221228_S5	300	30	6	1.5	5.35×10^{-8}	210
221228_S2b	300	30	6	1.5	7.05×10^{-8}	266
Type-II device: MoS₂/CrOCl/h-BN/Au p-FET						
	d_{BN} (nm)	d_{CrOCl} (nm)	L (μm)	W (μm)	$\frac{dI_{sd}}{dV_g}$ (A/V)	μ_h ($\text{cm}^2 \text{V}^{-1} \text{s}^{-1}$)
230711_S3a	45	10	2	5	3.48×10^{-6}	229
230711_S2	35	10	2	5	2.57×10^{-6}	140

As a summarization, an updated Supplementary Figure 19 is now added in the revised manuscript, and the Fig. 2h in the main text is also updated with multiple devices.”.

Comment 6. *Since in an ultra-scaled region, the gate pitch or contact pitch will be < 100 nm. In such dimensions, will the CrOCl still be an effective doping functional layer? and why?*

Response:

In our experiment, the P-doping scheme still works for the MoS₂/CrOCl system at a channel length of 100 nm (gate pitch ~ 100 nm), as shown in Fig. R7. The reason is simply because the interfacial coupling doping is effective as long as the *e-e* interaction assisted P-doping process (Fig. R3b-c) is still valid, that is, when a clean interface of MoS₂/CrOCl is existing, even at the ultra-scaled region. Due to the limitation of our lithography tool (Zeiss Sigma300 scanning electron microscope equipped with a Raith Elphy Quantum graphic writer), shorter channel length (sub-50 nm for example) was not carried out.

Figure R7. P-type MoS₂/CrOCl devices for sub 500 nm channel lengths. P-type transfer curves of typical MoS₂/CrOCl devices for channel length of (a) 200 nm and (b) 100 nm.

Comment 7. *One of the major advantages of 2D TMDs' arise from their ultra-thin channel thickness, which can be thinner than 1 nm. However, the authors claimed that the proposed doping method only worked for channel thickness thicker than 3.2 nm, which restricts its capability to deliver the desirable performance of 2D TMDs. So, what is the practical utility of the proposed method?*

Response:

We thank the comments given by Referee #2.

We would rather not concur with the comment “...channel thickness thicker than 3.2 nm, which restricts its capability to deliver the desirable performance of 2D TMDs...”

The desirable performance of 2D TMD has not to be necessarily in a monolayer (thinner than 1 nm).

On the contrary, we tend to believe that, in terms of future potential applications of 2D semiconductors, multilayered TMD might be very promising – as they do offer advantages in specific aspects. For example, they have higher tolerance toward defects/impurities, sometimes better mobility^[R4].

^{R4} Qinqin Wang, *et al.*, “Layer-by-layer epitaxy of multi-layer MoS₂ wafers”, *National Science Review*, **9**, nwac077 (2022).

We would like to emphasize that our aim is not to utilize ‘monolayer’ 2D semiconductors, which actually is surely not the only pursuit in the studies of 2D semiconductors. Moreover, notice that even 10-layers TMD is still less than 10 nm, way thinner than any Si or Ge FET. It is advisable to refrain from becoming excessively focused on attaining a monolayer, provided that the wavefunctions/behavior remain within the confines of the two-dimensional regime.

Again, let’s get back to the main contribution of our paper: Our focus is to find a new doping route, to switch the N-type TMD into P-type, in a stable, clean, and 3D-stackable manner. It further allows them to be vertically integrated/stacked, **to realize a new paradigm of 3D vertically stacked 2D complementary logics.**

Comment 8. It can be observed from Figure R8 that the monolayer TMD device exhibited a worse performance, i.e., $I_{on} < 1\mu A$ under $V_{ds} = 1V$. This suggests that some kind of scattering hindered the electron transport, which could be induced by the doping layer. Can the authors confirm that this degradation is due to something else?

Response:

Here in Figure R8 in the previous submission, the monolayer semiconductor MoS₂ was placed on Cr₂Ge₂Te₆, a different material than CrOCl. In this case, the performance of P-type MoS₂ is indeed having a small ON current.

This poor hole transport performance might come from the Cr₂Ge₂Te₆ substrate, which is known to be vulnerable to air exposure. Since our focus is mainly on CrOCl, we therefore will not devote too much our time in analyzing the degradation of Cr₂Ge₂Te₆ itself.

The key point is to demonstrate that the switch from n- to P-doping via interfacial coupling works for some of the monolayer semiconductors, depending on the selection of substrate underneath. Again, the ultimate goal of our work is to demonstrate a vertically stackable and polarity reversible doping scheme (realized by the TMD with and without interfacial coupling to CrOCl), thus allowing the 3D integration of complementary 2D logics.

Comment 9. The DFT simulation results of monolayer MoS2 doping with CrOCl should be provided to illustrate why P-doping does not happen in monolayer MoS2.

Response:

We thank the comments given by Referee #2.

Monolayer MoS₂ has a larger gap with much lower VBM energy, and cannot be tuned to be p-doped by non-disruptive negative gate, as verified by our DFT calculations. Specifically, in Fig. R8, we present the VBM of monolayer MoS₂ and the CBM of CrOCl as a function of negative electric field (pointing from MoS₂ to CrOCl), clearly the CrOCl CBM is always far above monolayer MoS₂ VBM, thus the chemical potential would remain in the gap of monolayer MoS₂, which thus cannot be p-doped.

Figure R8. Comparison of MoS₂/CrOCl band alignments. The figures show MoS₂ VBM and CrOCl CBM calculated at different vertical electrical fields for (a) 10-layer, and (b) monolayer MoS₂, interfaced with 5-layer CrOCl, respectively.

A new figure in the Supplementary Information is included in the revision, in order to clarify this point raised by Referee #2, also quoted below in Fig. R9 (cited as Supplementary Figure 23 in the new submission). We thank her/his valuable comment very much.

In conclusion, we sincerely appreciate the very constructive comments given by Referee #2 during the review process, which has significantly improved the quality of our manuscript.

Her/his support in publication in Nature will be greatly appreciated.

Reply to referee #3

General Comment. *The authors have responded appropriately to the reviewer's comments and the manuscript has been well revised. The manuscript can be published in the present form.*

Response:

We appreciate very much the support of publication by Referee#3.

Reviewer Reports on the Second Revision:

Referees' comments:

Referee #1 (Remarks to the Author):

The revised manuscript is now suitable for publication in Nature as the authors have addressed my criticisms. However, it was noted that referee #2 raised several points for the authors to consider in the future development of their method/technique for the semiconductor industry. I agree with the common assertion that a scientific paper cannot solve all problems in cutting-edge advanced node chip manufacture. However, now, more than a decade after the first MoS₂ transistor was fabricated, individuals within the 2D community should seriously consider this.

In terms of the topic of this work, various methods have already been developed to reverse the polarity of MoS₂, WSe₂, MoTe₂, etc. However, while these methods may be interesting, physically sound, and innovative, they may not be technically suitable for practical implementation in 2D IC manufacture. Therefore, I hope this work will not fall into that category, and I encourage authors to further research this topic with an emphasis on developing actual 2D ICs rather than solely publishing.

Referee #2 (Remarks to the Author):

The reviewer acknowledges the detailed response and is fully aware of the scope of the work. However, although the 3D integration of over 10 layers is appreciated, there is nothing sufficiently novel in this work to warrant publication in Nature. Specifically, the major finding claimed by the authors regarding the 3D vertical integration of complementary 2D semiconductors lacks novelty due to the prevalence of similar demonstrations in the literature (doi: 10.1038/s41586-023-06860-5; 10.1038/s41563-023-01704-z; 10.1038/s41699-023-00371-7). Contrary to the authors' claim, the p-type doping method proposed in this study lacks uniqueness for 3D integration, as there are alternative promising approaches available for achieving 3D complementary logic (doi: 10.1109/IEDM19574.2021.9720533). Additionally, the p-type doping aspect of the study is indistinguishable from charge transfer doping, as both methods can result in polarity reversal effects, as evidenced by the available literature (doi: 10.1021/acsami.1c11328; 10.1002/adma.201505154).

Author Rebuttals to Second Revision:

Reviewer #1(Remarks to the Author):

The revised manuscript is now suitable for publication in Nature as the authors have addressed my criticisms. However, it was noted that referee #2 raised several points for the authors to consider in the future development of their method/technique for the semiconductor industry. I agree with the common assertion that a scientific paper cannot solve all problems in cutting-edge advanced node chip manufacture. However, now, more than a decade after the first MoS₂ transistor was fabricated, individuals within the 2D community should seriously consider this.

In terms of the topic of this work, various methods have already been developed to reverse the polarity of MoS₂, WSe₂, MoTe₂, etc. However, while these methods may be interesting, physically sound, and innovative, they may not be technically suitable for practical implementation in 2D IC manufacture. Therefore, I hope this work will not fall into that category, and I encourage authors to further research this topic with an emphasis on developing actual 2D ICs rather than solely publishing.

Response:

We fully agree with the referee that “a scientific paper cannot solve all problems in cutting-edge advanced node chip manufacture”. And her/his encouragement about not only staying on the stage of publishing, but rather to push the study a step forward to real application, is certainly very helpful and inspiring.

In the final revision, we have revised the whole abstract to include more up-to-date the literature, and have also stated in Page 7, the last paragraph on the left column:

“Nevertheless, we emphasize that there are yet further technical challenges that need to be addressed in the longer-term perspective for the 3D-integration of 2D semiconductors in future nanoelectronics at a large-scale application (We became aware that recent efforts on wafer-scale vdW vertical-CFETs have been reported during our submission [2, 3]).”

Indeed, we are focusing our future efforts on obtaining wafer-scale CrOCl, and are trying to scale up our doping-scheme in the near future, although there are still quite some challenges.

Overall, our manuscript has been improved significantly thanks to the very constructive comments from Referee#1. We appreciate very much the Reviewer #1 for her/his support of publication in Nature.

Reviewer #2 (Remarks to the Author):

The reviewer acknowledges the detailed response and is fully aware of the scope of the work. However, although the 3D integration of over 10 layers is appreciated, there is nothing sufficiently novel in this work to warrant publication in Nature. Specifically, the major finding claimed by the authors regarding the 3D vertical integration of complementary 2D semiconductors lacks novelty due to the prevalence of similar demonstrations in the literature (doi: 10.1038/s41586-023-06860-5; 10.1038/s41563-023-01704-z; 10.1038/s41699-023-00371-7). Contrary to the authors' claim, the p-type doping method proposed in this study lacks uniqueness for 3D integration, as there are alternative promising approaches available for achieving 3D complementary logic (doi: 10.1109/IEDM19574.2021.9720533). Additionally, the p-type doping aspect of the study is indistinguishable from charge transfer doping, as both methods can result in polarity reversal effects, as evidenced by the available literature (doi: 10.1021/acsami.1c11328; 10.1002/adma.201505154).

Response:

We appreciate very much the criticisms by Referee#2. And all her/his comments in the past rounds of reviewing are very helpful in terms of improving our manuscript.

As there is no further technical concerns in her/his comments now, we note that referee#2 pointed out some referencing issues. Indeed, there are other methods existing for both implementing polarity reversal in 2D TMDs and (very recently) for achieving 3D vertical integration of complementary 2D semiconductors. And some of the above references has already been cited (10.1002/adma.201505154) at our first submission, while some other (10.1038/s41586-023-06860-5) was published when our paper was under consideration.

In the final revision, we have systematically updated the referencing (and associated discussion). There are 10 new references added, and the abstract is re-written to accurately represent the very current state of the art. Related discussions are revised in the main text in Pages 2,3 and 7, highlighted in blue.

Again, we thank Referee#2 for her/his very constructive comments.